# Decadal changes in global surface NOx emissions from multi-constituent satellite data assimilation

Kazuyuki Miyazaki[1,5], Henk Eskes[2], Kengo Sudo[3], K. Folkert Boersma[4,2], Kevin Bowman[5], and Yugo Kanaya[1]

[1]Japan Agency for Marine-Earth Science and Technology, Yokohama 236-0001, Japan
[2]Royal Netherlands Meteorological Institute (KNMI), Wilhelminalaan 10, 3732 GK, De Bilt, The Netherlands
[3]Graduate School of Environmental Studies, Nagoya University, Nagoya, Japan
[4]Wageningen University, Meteorological and Air Quality department, Wageningen, the Netherlands
[5]Jet Propulsion Laboratory-California Institute of Technology, Pasadena, CA, USA

*Correspondence to:* K. Miyazaki (kmiyazaki@jamstec.go.jp)

**Abstract.**

Global surface emissions of nitrogen oxides ($NO_x$) over a ten-year period (2005–2014) are estimated from an assimilation of multiple satellite datasets: tropospheric $NO_2$ columns from OMI, GOME-2, and SCIAMACHY; $O_3$ profiles from TES; CO profiles from MOPITT; and $O_3$ and $HNO_3$ profiles from MLS using an ensemble Kalman filter technique. Chemical concentrations of various species and emission sources of several precursors are simultaneously optimized. This is expected to improve the emission inversion because the emission estimates are influenced by biases in the modelled tropospheric chemistry, which can be partly corrected by also optimizing the concentrations. We present detailed distributions of the estimated emission distributions for all major regions, the diurnal and seasonal variability, and the evolution of these emissions over the ten-year period. The estimated regional total emissions show a strong positive trend over India (+29 %/decade), China (+26 %/decade), and the Middle East (+20 %/decade), and a negative trend over the United States (-38 %/decade), Southern Africa (-8.2 %/decade), and western Europe (-8.8 %/decade). The negative trends in the United States and western Europe are larger during 2005–2010 relative to 2011–2014, whereas the trend in China becomes negative after 2011. The data assimilation also suggests a large uncertainty in anthropogenic and fire-related emission factors and an important underestimation of soil $NO_x$ sources in the emission inventories. Despite the large trends observed for individual regions, the global total emission is almost constant between 2005 (47.9 Tg N yr$^{-1}$) and 2014 (47.5 Tg N yr$^{-1}$).

## 1 Introduction

Nitrogen oxides ($NO_x$=NO+$NO_2$) play an important role in air quality, tropospheric chemistry, and climate. Tropospheric $NO_x$ concentrations are highly variable in both space and time, reflecting its short chemical lifetime in the atmosphere and the heterogeneous distribution of its sources and sinks. Emission sources are important in determining the amount and distribution of $NO_x$. Natural $NO_x$ sources include biogenic emissions from bacteria in soils, biomass burning, and lightning. Anthropogenic $NO_x$ sources include fossil fuel and biofuel combustion, emissions from vehicle transport, and industrial emissions. Bottom-

up inventories from different sources and regions contain large uncertainties, which result from inaccurate emission factors and activity rates for each source category. Examples include traffic rush hours, economic activity, biomass-burning activity, wintertime-heating of buildings, and rain-induced emission pulses of $NO_x$ (e.g., Velders et al., 2001; Jaeglé et al., 2005; Wang et al., 2007; Xiao et al., 2010; Streets et al., 2013; Castellanos et al., 2014; Reuter et al., 2014; Vinken et al., 2014; Oikawa et al., 2015). As a result, bottom-up inventories generally do poorly at representing the spatial and temporal variability at multiple scales (i.e., diurnal, daily, seasonal, and interannual). Large uncertainties in biomass burning emissions mainly reflect a relative lack of observations for characterizing the large spatial and temporal variations of burning conditions (Castellanos et al., 2014). The wide range in soil $NO_x$ emission estimates in previous studies reflect incomplete knowledge of the emission factors and processes driving these emissions (Oikawa et al., 2015). Recent studies (e.g., Steinkamp and Lawrence, 2011, Hudman et al., 2012, Vinken et al., 2014) suggest that soil $NO_x$ emissions are likely around 10 $Tg\,N\,yr^{-1}$, a considerable increase relative to earlier studies that assumed about 5 $Tg\,N\,yr^{-1}$ soil $NO_x$ emissions (Yienger and Levy, 1995). Large uncertainties are also in lightning $NO_x$ ($LNO_x$) source estimates. Schumann and Huntrieser (2007) provided a best estimate of 5±3 $Tg\,N\,yr^{-1}$ for the annual global $LNO_x$ source. More recently, Murray et al. (2012), Stavrakou et al. (2013), and Miyazaki et al. (2014) estimated at 6±0.5, 3.3–5.9, and 6.3±1.4 $Tg\,N\,yr^{-1}$, respectively.

Tropospheric $NO_2$ columns retrieved from satellite measurements, including the Global Ozone Monitoring Experiment (GOME), Scanning Imaging Absorption Spectrometer for Atmospheric Cartography (SCIAMACHY), GOME-2, and the Ozone Monitoring Instrument (OMI), have been used to infer $NO_x$ emissions using top-down approaches (e.g., Martin et al., 2003; Richter, 2004; Jaeglé et al., 2005; van der A et al., 2006; Zhang et al., 2007; Boersma et al., 2008a; Stavrakou et al., 2008; van der A et al., 2008; Kurokawa et al., 2009; Zhao and Wang, 2009; Lamsal et al., 2010; Lin et al., 2010; Miyazaki et al., 2012a; Gu et al., 2013; Mijling et al., 2013; Vinken et al., 2014; Ding et al., 2015; Lu et al., 2015). Long-term tropospheric $NO_2$ column records have allowed us to investigate changes in the atmospheric environment over the past decade as a result of economic growth and emission controls over major polluted regions (Castellanos and Boersma, 2012; Hilboll et al., 2013; Cui et al., 2015; Lelieveld et al., 2015; Wang et al., 2015; Duncan et al., 2016; Krotkov et al. 2016).

Advanced data assimilation techniques such as four-dimensional variational assimilation (4D-VAR) (Müller and Stavrakou, 2005; Kurokawa et al., 2009; Chai et al., 2009) and ensemble Kalman filter (EnKF) (Miyazaki et al., 2012a, 2012b, 2014, 2015) have been employed to take full advantage of the chemical transport model (CTM) and satellite retrievals in top-down emission estimates. These advanced techniques consider flow-dependent forecast error covariance and take errors from both the model and retrievals into account. These advantages are considered essential for improving long-term global emission estimates, as dominant atmospheric processes, the emission–concentration relationships, and observational sampling and errors must be incorporated into the analysis. These advanced methodologies can readily assimilate multiple-species. The additional observations of $O_3$ and CO constrain surface $NO_x$ emissions through their indirect impact on $NO_2$ concentrations through tropospheric chemistry. These species directly influence OH concentrations, which control the $NO_x$ variability and lifetime, and indirectly the accuracy of the emission estimates. Chemically consistent, multi–constituent assimilation is an advance over conventional approaches, which assume $NO_2$ observations are uniquely controlled by $NO_x$ emissions.

Various sources of error in current chemical transport models (CTMs) impact the simulated $NO_x$ lifetime and the accuracy of $NO_x$ emission inversions (Lin et al., 2012a; Miyazaki et al., 2012a; Stavrakou et al., 2013). Stavrakou et al. (2013) showed the strong effect of chemical $NO_x$ loss uncertainties on top-down $NO_x$ source inversions. OH is the main radical responsible for the removal of atmospheric pollution and for determining the lifetime of many chemicals including $NO_x$ (Levy, 1971; Logan et al., 1981; Thompson, 1992), but its concentrations in CTMs are considered to have large uncertainties (Naik et al., 2013; Miyazaki et al., 2015; Patra et al., 2015). Meanwhile, representations of $LNO_x$ sources are essential for realistic representations of tropospheric $NO_2$ columns, but current parameterizations contain large uncertainties (Martin et al., 2007; Schumann and Huntrieser, 2007; Miyazaki et al., 2014). Errors in representing these natural sources of $NO_2$ can directly propagate into surface $NO_x$ emissions estimates.

Increasing attention has been paid to combining observations of multiple-species to improve the analysis of tropospheric chemistry, including for $NO_x$ emission estimates. Measurements of species other than $NO_2$ (e.g., $O_3$ and $HNO_3$) could improve the representation of $NO_x$ in models through their chemical interactions with $NO_x$ (e.g., Hamer et al., 2015). Advanced data assimilation techniques such as 4D-VAR and EnKF propagate observational information from a limited number of observed species to a wide range of chemical components. Miyazaki et al. (2012b, 2014, 2015) and Miyazaki and Eskes (2013) demonstrated that the assimilation of multiple-species observations, taking their complex chemical interactions into account using an EnKF technique, can provide comprehensive constraints on both concentration and emissions, and this approach has the potential to improve emission inversions by accounting for confounding factors in the relationship between $NO_x$ emissions and $NO_2$ concentrations. Because of the simultaneous assimilation of multiple-species data with optimisation of both the concentrations and emission fields, the global distribution of OH was modified considerably, decreasing the OH gradient between NH and SH (Miyazaki et al., 2015). The changes in OH are the important chemical pathway for propagating observational information between various species and for modulating the chemical lifetimes among these species.

In this study, we estimate global surface $NO_x$ emissions between 2005 and 2014 using the assimilation of multiple-species data from OMI $NO_2$, GOME-2 $NO_2$, SCIAMACHY $NO_2$, Tropospheric Emission Spectrometer (TES) $O_3$, Measurement of Pollution in the Troposphere (MOPITT) CO, and Microwave Limb Sounder (MLS) $O_3$ and $HNO_3$ retrievals using an EnKF technique. We attempted to optimize the diurnal variations in surface $NO_x$ emissions, while updating daily, seasonal, and interannual emission variations, based on a combination of three $NO_2$ retrievals obtained at different overpass times. The assimilation of multiple chemical data sets with different vertical sensitivity profiles provides comprehensive constraints on the global $NO_x$ emissions while improving the representations of the entire chemical system affecting tropospheric $NO_2$ column variations, including $LNO_x$ sources. Based on the EnKF estimations, this study presents detailed distributions of the surface $NO_x$ emissions for all major regions, the diurnal and seasonal variability, and the development over the ten-year period.

## 2   Methodology

The data assimilation system is constructed based on the global CTM MIROC-Chem (Watanabe et al. 2011) and on a variance of the EnKF technique. The basic framework is similar to the system used to produce tropospheric chemistry reanalysis data

 in our previous study (Miyazaki et al., 2015); however, some updates to the data assimilation framework have been made and the calculation has been extended to cover the ten years from 2005 to 2014, as described below.

## 2.1 MIROC-Chem model and a priori emissions

The original forecast model used in our previous study (CHASER ; Sudo et al. 2002) is replaced by the newer MIROC-Chem model (Watanabe et al., 2011). MIROC-Chem represents the chemistry part of the MIROC-ESM Earth system model. It considers detailed photochemistry in the troposphere and stratosphere by simulating tracer transport, wet and dry deposition, and emissions, and calculates the concentrations of 92 chemical species and 262 chemical reactions (58 photolytic, 183 kinetic, and 21 heterogeneous reactions). Its tropospheric chemistry was developed based on the CHASER model, with many updates to

chemical reactions and emissions, considering the fundamental chemical cycle of $Ox$-$NO_x$-$HOx$-$CH_4$-$CO$ along with oxidation of NMVOCs (ethane, ethane, propane, propene, butane, acetone, methanol, isoprene, and terpenes) to properly represent ozone chemistry in the troposphere. Its stratospheric chemistry was developed based on the CCSR/NIES stratospheric chemistry model (Akiyoshi et al., 2004), which calculates chlorine and bromine containing compounds, CFCs, HFCs, OCS, $N_2O$, and the formation of PSCs and associated heterogeneous reactions on their surfaces.

MIROC-Chem has a T42 horizontal resolution (approximately $2.8° \times 2.8°$) and uses the hybrid terrain-following pressure vertical coordinate system with 32 vertical levels from the surface to 4.4 hPa. It is coupled to the atmospheric general circulation model MIROC-AGCM version 4 (Watanabe et al., 2011). The radiative transfer scheme considers absorption within 37 bands, scattering by gases, aerosols, and clouds, and the effect of surface albedo. Detailed radiation calculations are used for photolysis calculation. The MIROC-AGCM fields were nudged toward the 6-hourly ERA-Interim (Dee et al., 2011) at every model time

step to reproduce past meteorological fields and to simulate short-term (i.e., less than 6 hours) meteorological variability and sub-grid scale transport effects.

The forecast model update from CHASER to MIROC-Chem improved the simulated profiles of various tropospheric species (not shown). The inclusion of stratospheric chemistry in MIROC-Chem allowed us to provide reasonable estimates of a priori profiles and their ensemble spread in the stratosphere. Since TES $O_3$ and MOPITT CO retrievals in the troposphere, together

with MLS retrievals, have sensitivity to the lower stratospheric concentration to some degree, the improved representation of background error covariance in the stratosphere, as estimated from ensemble model simulations, meant that satellite retrievals are more effectively assimilated into the updated system throughout the troposphere and stratosphere through the use of observation operator (c.f., Sec. 2.3).

The a priori values for surface emissions of $NO_x$ and CO were obtained from bottom-up emission inventories. Annual

total anthropogenic $NO_x$ and CO emissions were obtained from the Emission Database for Global Atmospheric Research (EDGAR) version 4.2 (EC JRC/PBL, 2012) for 2005–2008. Emissions from biomass burning were based on the monthly Global Fire Emissions Database (GFED) version 3.1 (van der Werf et al., 2010) for 2005–2011. Emissions from soils were based on monthly mean Global Emissions Inventory Activity (GEIA) (Yienger and Levy, 1995). To cover data limitations during 2005–2014, EDGAR emissions for 2008 were used in the calculations for 2009–2014, and GFED emissions averaged

over 2005–2011 were used in the 2012–2014 calculation. The global total a priori $NO_x$ emissions averaged over the 2005-2014 period from anthropogenic sources, biomass burning, and soils are 28.7, 4.3, and 5.4 $Tg\,N\,yr^{-1}$, respectively. The total aircraft $NO_x$ emission is 0.55 $Tg\,N\,yr^{-1}$, which is obtained from the EDGAR inventory.

Following the settings of Lotos-Euros (Schaap et al., 2008) and Boersma et al. (2008b), we applied anthropogenic-type diurnal variations for total emissions with maxima in morning and in evening with a factor of about 1.5 (black dotted line in Fig. 1, for which the daily mean hourly emission value is 1) in Europe, eastern China, South Korea, Japan, India, and North America; biomass burning-type variations with a rapid increase in morning and maximal emissions in the mid-day with a maximum factor of about 4 in North and central Africa, southeast Asia, and northern and central South America; and soil-type diurnal variations with maximal emissions in afternoon with a factor of about 1.2 in Australia, Sahara, western China, and Mongolia.

$LNO_x$ sources in MIROC-Chem were calculated in conjunction with the convection scheme of MIROC-AGCM. The global distribution of the flash rate was parameterised for convective clouds based on the relationship between lightning activity and cloud top height (Price and Rind, 1992). The vertical profiles of the $LNO_x$ sources are determined on the basis of the C-shaped profile given by Pickering et al. (1998). The mean yearly global flash rate obtained for 2005–2014 was 42.4 $flashes\,s^{-1}$, which is close to climatological estimates of 46 $flashes\,s^{-1}$ derived from Lightning Imaging Sensor (LIS) and Optical Transient Detector (OTD) measurements (Cecil et al., 2014). The $LNO_x$ sources were optimized in the data assimilation runs, following the method of Miyazaki et al. (2014).

## 2.2 Emission estimates from EnKF data assimilation

Data assimilation is based on an ensemble square root filter (SRF) EnKF approach (i.e., a local ensemble transform Kalman filter; LETKF; Hunt et al., 2007). As in other EnKF approaches, the background error covariance is estimated from ensemble model forecasts based on the assumption that background ensemble perturbations sample the forecast errors. Using the covariance matrices of observation error and background error, the data assimilation determines the relative weights given to the observation and the background, and then transforms a background ensemble into an analysis ensemble. Unlike standard EnKF analyses, the LETKF analysis is performed locally in space and time, which reduces sampling errors caused by limited ensemble size. Furthermore, the analysis is performed independently for different grid points, which reduces the computational cost through parallel computations. More details on the data assimilation technique are given in Miyazaki e al. (2015).

The emission estimation is based on a state augmentation technique, which was employed in our previous studies (Miyazaki et al., 2012a; 2012b; 2013; 2014; 2015). In this approach, the background error correlations, estimated from the ensemble model simulations at each analysis step, determine the relationship between the concentrations and emissions of related species for each grid point. This approach allows us to reflect temporal and geographical variations in transport and chemical reactions in the emission estimates. The state vector in this study is optimized following Miyazaki et al. (2015), which includes several emission sources (surface emissions of $NO_x$ and CO, and $LNO_x$ sources) as well as the concentrations of 35 chemical species. In order to improve the filter performance, the covariance among non- or weakly related variables in the state vector is set to zero, as in Miyazaki et al (2012b) and Miyazaki et al (2015). The emissions in the state vector are represented by scaling factors

for each surface grid cell for the total $NO_x$ and CO emissions, and for each production rate profile of the $LNO_x$ sources. For surface $NO_x$ emissions, only the combined total emission is optimized in data assimilation. This is to reduce the degree of freedom in the analysis and to avoid the difficulty associated with estimating spatiotemporal variations in background errors for each category source separately.

In the MIROC-Chem simulations, an emission diurnal variability function ($Et$ ($t = 1, ..., 24$)) was applied following the approach of Miyazaki et al. (2012a). Its application generally improved the model simulation performance; however, because $Et$ was constructed based on simple assumptions, and because it does not change with season and location within an area of the same dominant category, its application can cause large uncertainties in simulated $NO_2$ variations. Multiple satellite $NO_2$ retrievals obtained at different overpass times have a potential to constrain diurnal emission variability (e.g., Lin et al., 2010),

although differences between the different $NO_2$ retrievals and errors in model processes could introduce artificial corrections (see also Section 5.2). Note that the retrievals from different instruments used are all based on the same retrieval method (DOMINO v2, TM4NO2A v2) and largely consistent ancillary data, which limits the discrepancies between the data sets to large degree (Boersma et al., 2008) (see Section 2.3.1). We also acknowledge that differences between the surface reflectivity and cloud data used may lead to some structural uncertainty between the morning and afternoon sensors, although numerous

validation studies pointed out that the three $NO_2$ column retrievals agree well with independent reference data (e.g., Irie et al., 2011; Ma et al., 2013).

We attempt to optimize $Et$ using data assimilation of OMI, SCIAMACHY, and GOME-2 retrievals, with local equator overpass time of 13:45, 10:00 and 9:30, in order to improve the representation of diurnal emission variability. In our approach, a correction factor for emission diurnal variability ($Etc$) and an emission scaling factor ($Es$) for surface $NO_x$ emissions are

simultaneously optimized in the analysis step using multiple $NO_2$ retrievals, by adding them to the state vector together with other variables such as predicted concentrations. The background error correlation between $Es$ and $Etc$ is not considered; the two emission parameters are independently optimized using measurements from instruments with different overpass times. As in Miyazaki et al. (2012a), we apply covariance inflation to the emission factors to prevent covariance underestimation caused by the application of a persistent forecast model, by inflating the spread to a minimum predefined value (i.e., 30 % of the

initial standard deviation) at each analysis step for both $Es$ and $Etc$. The initial error is set to 40 % for both $Es$ and $Etc$. For concentrations, multiplication factors (5 %) are applied to prevent an underestimation of background error covariance. The emission factors are analyzed and updated at every analysis step (i.e., two hours). Because of the lack of any applicable model, a persistent forecast model is used for the emission factors. When there is no observational information available in the analysis step, previously analyzed emission factors are used in the next forecast step.

Figure 1 depicts a schematic diagram of the emission correction scheme for anthropogenic emissions. First, we obtain optimal values of $Es$ and $Etc$ from the data assimilation analysis. Second, $Es$ is applied to scale up/down daily total emissions while maintaining the a priori diurnal variability shape (black solid line). Third, optimized $Etc$ is applied to modify the diurnal variability shape (red line). Considering the overpass time of the satellite retrievals and the typical daytime lifetime of $NO_x$ (i.e. 2–3 hours), a square-wave with amplitude of $Etc$ and a wavelength of six hours was applied. This assumes that GOME-2 and

SCIAMACHY measurements constrain emissions in the 07:30–10:30 window, and OMI measurements constrain the 10:30-

13:30 window. Consequently, an analysis of the emission diurnal variability function is obtained as $Et^a = Et^b \times Es - Etc$ for 07:30–10:30, and $Et^a = Et^b \times Es + Etc$ for 10:30–13:30), where $a$ and $b$ represent the analysis and background states, respectively. $Etc$ is set to zero (i.e., $Et^a = Et^b \times Es$) from 13:30 to 07:30. The optimized emission factors are used as initial conditions in the next forecast step of ensemble model simulations.

## 2.3 Measurements used in the assimilation

Trace gas concentrations were obtained from OMI, SCIAMACHY, and GOME-2 satellite measurements of $NO_2$, from TES of $O_3$, from MOPITT measurements of CO, and from MLS of $O_3$ and $HNO_3$. The retrieved concentration and observation error information were obtained for each retrieval, where the observation error included contributions from smoothing errors, model parameter errors, forward model errors, geophysical noise, and instrument errors. These combined errors, together with a representativeness error for super observations (Miyazaki et al., 2012a), were considered in the observation error matrix ($\mathbf{R}$) for data assimilation.

For the assimilation of the satellite retrievals, observation operators (H) were developed, consisting of the spatial interpolation operator ($S$), a priori profile in the satellite retrievals ($\boldsymbol{x}_{\mathrm{apriori}}$), and an averaging kernel ($\mathbf{A}$). This operator mapped the model fields ($\boldsymbol{x}_i^{\mathrm{b}}$) into retrieval space ($\boldsymbol{y}_i^{\mathrm{b}}$), as follows:

$$\boldsymbol{y}_i^{\mathrm{b}} = H(\boldsymbol{x}_i^{\mathrm{b}}) = \boldsymbol{x}_{\mathrm{apriori}} + \boldsymbol{A}(S(\boldsymbol{x}_i^{\mathrm{b}}) - \boldsymbol{x}_{\mathrm{apriori}}), \tag{1}$$

where $i$ indicates the ensemble member. The use of the averaging kernel $A$ removes the dependence of the analysis or of the relative model retrieval comparison ($(\boldsymbol{y}_i^{\mathrm{b}} - \boldsymbol{y}^{\mathrm{o}})/\boldsymbol{y}_i^{\mathrm{b}}$) on the retrieval a priori profile (Eskes and Boersma, 2003; Jones et al., 2003).

We employed the super-observation approach to produce representative data with a horizontal resolution of MIROC-Chem (T42) for OMI, SCIAMACHY, GOME-2, and MOPITT observations. Super observations were generated by averaging all data located within a super observation grid cell, following the approach of Miyazaki et al. (2012a). Super observation measurement error was estimated by considering an error correlation of 15 % among the data, although there is no evidence for this value. Representativeness error was introduced when the super-observation grid was not fully covered by observation pixels. The super-observation approach generally provided more representative data with reduced random error and resulted in more stable analysis increments than did the individual observations (Miyazaki et al., 2012a). Another popular approach in data assimilation is to apply data thinning. However, individual observations are much more noisy than super observations, and the representativity error is large. Note that, in our previous studies (Miyazaki et al., 2012a, 2012b, 2013, 2014, 2015), the super observation was produced with a resolution of $2.5° \times 2.5°$, which was similar but not equivalent to the model grid size (T42). In this study, the super observation was set to be equivalent to the model grid size (T42), which generally led to larger adjustments in the estimated emissions over industrial areas, and resulted in better data assimilation performance for most cases (e.g., reduced OmF).

### 2.3.1 Tropospheric NO$_2$ columns from OMI, SCIAMACHY, and GOME-2

The tropospheric NO$_2$ column retrievals used are from the version-2 DOMINO data product for OMI (Boersma et al., 2011) and version 2.3 TM4NO2A data products for SCIAMACHY and GOME-2 (Boersma et al., 2004) obtained through the TEMIS website (www.temis.nl). The ground pixel size of the OMI retrievals is 13–24 km with daily global coverage. Since December 2009, approximately half of the pixels have been compromised by the so-called row anomaly, which reduced the daily coverage of the instrument. GOME-2 retrievals have 80 km $\times$ 40 km ground pixel size with a global coverage within 1.5 days. SCIAMACHY retrievals have 60 km $\times$ 30 km ground pixel size with a global coverage once every 6 days. OMI measurements were assimilated throughout the analysis period during 2005–2014. In contrast, because of the data limitations, SCIAMACHY retrievals were assimilated before February 2012, and the GOME-2 measurements were assimilated after January 2007. Low-quality data were excluded before assimilation following the recommendations of the products' specification document (Boersma et al., 2011). We employed clear-sky data for surface NO$_x$ emission estimations and both clear-sky data and cloud-scene data for LNO$_x$ estimations, following the method of Miyazaki et al. (2014). The analysis increments in the assimilation of the NO$_2$ retrievals were limited to adjusting only the surface emissions of NO$_x$, LNO$_x$ sources, and concentrations of NO$_y$ species using the estimated inter-species error correlations.

Boersma et al. (2011) summarized the general error characteristics of tropospheric NO$_2$ retrievals. More recently, Maasakkers (2013) presented the possibility for improving the tropospheric NO$_2$ column retrievals algorithm; for example, in the a priori profiles, the effective surface pressure calculation, and in the cloud retrieval. Maasakkers (2013) presented an improved error parameterization for the tropospheric NO$_2$ column, which reduced errors in high tropospheric columns by up to 41 % and in the mean global error by 13 %. Following this result, we modified the version-2 DOMINO and version 2.3 TM4NO2A data products (Boersma et al., 2004; 2011) used in data assimilation; we reduced retrieval errors of individual NO$_2$ retrievals by 30 % over polluted areas (for columns $> 1.1 \times 10^{15}$ molec cm$^{-2}$) before producing super observation for all the NO$_2$ retrievals. The assimilation of NO$_2$ retrievals with reduced error increased the effective use of observational information (i.e., larger emission adjustments) and improved the chi-square statistics (not shown). The obtained super observation error is typically about 20–50 %, 30–60 %, and 25–50 % of the NO$_2$ columns over polluted areas for OMI, SCIAMACHY, and GOME-2 retrievals, respectively (Fig. S1). The differences between the instruments mainly reflect the differences in coverage and pixel size.

### 2.3.2 TES O$_3$

The Tropospheric Emission Spectrometer (TES) is a Fourier Transform Spectrometer (FTS) that measures spectrally-resolved outgoing longwave radiation of the Earth's surface and atmosphere. The TES O$_3$ data used are version 6 level 2 nadir data obtained from the global survey mode (Herman and Kulawik, 2013). This data set consists of 16 daily orbits with a spatial resolution of 5–8 km along the orbit track, with an equator crossing time of 13:40 and 02:29 local mean solar time. Retrievals of atmospheric parameters and their error characterization are based upon optimal estimation (Worden et al., 2004; Bowman et al., 2006; Kulawik et al., 2006) which provide the diagnostics (a priori, averaging kernels, and error covariances) needed to construct the observation operator. The standard quality flags were used to exclude low-quality data. The data assimilation of

the TES $O_3$ retrievals was performed based on the logarithm of the mixing ratio following the retrieval product specification (Bowman et al., 2006).

### 2.3.3 MLS $O_3$ and $HNO_3$

The MLS data used are the version 4.2 $O_3$ and $HNO_3$ level 2 products (Livesey et al., 2011). We excluded low quality data, following the recommendations of Livesey et al. (2011). We used data for pressures of less than 215 hPa for $O_3$ and 150 hPa for $HNO_3$. The accuracy and precision of the measurement error, described in Livesey et al. (2011), were included as the diagonal element of the observation error covariance matrix.

### 2.3.4 MOPITT CO

The MOPITT CO data used are version 6 level 2 TIR products (Deeter et al., 2013). The MOPITT instrument is mainly sensitive to free-tropospheric CO, especially in the middle troposphere, with degrees of freedom for signals (DOFs) typically much larger than 0.5. Owing to data quality problems, we excluded data poleward of 65° and night-time data. Data at 700 hPa were used for constraining surface CO emissions.

## 2.4 Measurements used in the validation

We use vertical $NO_2$ profiles observed from in-situ and aircraft measurements to validate the simulated $NO_2$ distributions. The model simulation and assimilation fields were interpolated to the time and location of each measurement, and then compared with the measurements.

### 2.4.1 DANDELIONS

Vertical $NO_2$ profiles were measured using the Netherlands National Institute for Public Health and the Environment (RIVM) NO2 lidar during the Dutch Aerosol and Nitrogen Dioxide Experiments for Validation of OMI and SCIAMACHY (DANDE-LIONS) campaign in September 2006 (Volten et al., 2009). The lidar data have a spatial representation of 2 km in the viewing direction and approximately 12 km in the direction of the wind, which is much finer than the model resolution (approximately 2.8°). The model grid points used for the interpolation around Cabauw are located in Belgium, northeastern Netherlands, western Germany, and on the North Sea. Boundary layer conditions are different among the grid points, especially between land and ocean. To avoid a possibly large error of representativeness in the validation, particularly under the different boundary layer condition, the profiles obtained in the morning (before 12:00 p.m.) were used because the differences between land and sea mixing layer depths are then still relatively small, following Miyazaki et al. (2012a).

### 2.4.2 INTEX-B

During the Intercontinental Chemical Transport Experiment Phase B (INTEX-B) campaign, vertical $NO_2$ profiles were obtained using the UC Berkeley Laser-Induced Fluorescence (TD-LIF) instrument on a DC-8 over the Gulf of Mexico in March

2006 (Singh et al., 2009). We removed data collected over highly polluted areas over Mexico City and Houston from the comparison to avoid a serious spatial representativeness error, as applied in Miyazaki et al. (2015).

### 2.4.3 ARCTAS

The Arctic Research of the Composition of the Troposphere from Aircraft and Satellites (ARCTAS) campaign (Jacob et al., 2010) was conducted over Alaska (between 60–90°N) in April 2008 (ARCTAS-A) and over western Canada (between 50–70°N) in June–July 2008 (ARCTAS-B). Since the data assimilation impact is limited in polar regions, the profile data obtained during ARCTAS-B were used in the comparison. Note that Browne et al. (2011) investigated that the observed $NO_2$ concentrations could be too high in the upper troposphere.

### 2.4.4 DC3

The Deep Convective Clouds and Chemistry (DC3) experiment field campaign was conducted over northeastern Colorado, western Texas to central Oklahoma, and northern Alabama during May and June 2012 (Barth et al., 2015). The observations obtained from the DC-8 by the UC Berkeley measurement were used in the validation.

### 2.4.5 SEAC⁴RS

The Studies of Emissions and Atmospheric Composition, Clouds and Climate C oupling by Regional Surveys (SEAC⁴RS) aircraft campaign was conducted over the southeast US in August–September 2013 (Travis et al., 2016). The observations obtained from the DC-8 by the UC Berkeley measurement were used in the validation.

## 3 Simulated and retrieved tropospheric $NO_2$ columns

Tropospheric $NO_2$ columns obtained from data assimilation and model simulation (without any assimilation) are compared with satellite observations. For these comparisons, concentrations were interpolated for the retrieval pixels to the overpass time of the satellite, while applying the averaging kernel of each retrieval, and both the retrieved and simulated concentrations were mapped on the horizontal grid of the super observation (i.e., T42).

### 3.1 Global distribution

Figure 2 compares global distributions of annual mean tropospheric $NO_2$ columns obtained from the three satellite retrievals (OMI for 2005-2014, SCIAMACHY for 2005–2011, and GOME-2 for 2007–2014), the MIROC-chem simulation, and the data assimilation. The three satellite measurements commonly reveal high tropospheric $NO_2$ concentrations over large industrial regions: eastern China, Europe, and the United States. High concentrations are also found over the Southern and Central Africa, India, Middle East, Japan, South Korea, and Southeast Asia. Tropospheric $NO_2$ concentrations are generally lower in OMI retrievals compared to GOME-2 and SCIAMACHY retrievals over polluted areas, reflecting the diurnal cycle of emissions and chemistry, with faster chemical loss of $NO_2$ at noon compared to early morning (e.g. Boersma et al., 2009). All of the retrievals

are produced using the same retrieval approach (Boersma et al., 2011). Therefore, the differences in overpass time and also in pixel size could be the main cause of the differences between the three different satellite retrievals, although the use of super observations for all the sensors reduces the influence of different pixel sizes.

The MIROC model reproduces the general features of observed tropospheric $NO_2$, with a global spatial correlation of 0.86–0.94 for the annual mean concentration during the ten-year period between 2005–2014 (Fig. 2 and Table 1). However, the simulated regional mean tropospheric $NO_2$ columns are generally too low over most industrial areas and major biomass-burning areas and too high over remote areas. In the global mean, the model is negatively biased relative to the three retrievals (i.e., -0.04−-0.18×$10^{15}$ molec cm$^{-2}$ compared with the three retrievals). Data assimilation improves agreements with the satellite retrievals for most industrial and biomass-burning areas mainly because of the optimized surface $NO_x$ emissions, with great reductions in the ten-year global mean negative bias (i.e., -0.02−+0.03×$10^{15}$ molec cm$^{-2}$) (Table 1). Improvements can also be found in the improved spatial correlation (from 0.86–0.94 to 0.95–0.98) and the reduced global root mean square error (RMSE: reduced by about 40, 30, and 50 % compared with OMI, SCIAMACHY, and GOME-2, respectively). The annual mean analysis–observation differences show similar spatial distributions between SCIAMACHY and GOME-2 (r=0.93) and differed somewhat between OMI and other sensors (r=0.55–0.60).

## 3.2 Regional distribution

The regional mean tropospheric $NO_2$ columns are compared in Table 2. The data assimilation reduced the ten-year mean negative bias of the model by 40–62 % over China and 48–50 % over the United States compared to the three retrieval. The data assimilation also reduced the almost constant negative bias over Australia by 20–76 %, over India by 57–60 %, and over Southern Africa by 35–64 %. The error reduction over China and southern Africa is generally smaller for the SCIAMACHY and GOME-2 retrievals compared with the OMI retrievals.

Improvements are also found over biomass burning areas. The ten-year mean negative model bias over Southeast Asia is reduced by 57–77 %, which is mainly attributed to the positive adjustments in the biomass burning season (i.e., in boreal winter-spring). The persistent negative biases throughout the year over central and North Africa are also reduced, with a ten-year mean reduction of 66-80 % and 78–86 %, respectively. These improvements over the tropical regions are mostly commonly found in comparisons with the three retrievals. Considering the short lifetime and rapid diurnal variation of biomass burning activity at low latitudes, these improvements suggest that the assimilation of multiple-species and multiple $NO_2$ measurements effectively corrected the temporal changes in the tropospheric $NO_2$ column between the different overpass times.

Despite the general improvement by data assimilation, disagreements remain between the simulated and observed $NO_2$ concentrations over polluted regions, such as Europe, Southern Africa, and China. The inadequacies of the improvements can be partly attributed to the small number of observations and large observation errors for highly polluted cases. The quality and abundance of the retrievals varies largely with season and area (Fig. S1), reflecting observation conditions (e.g., clouds, aerosols, and surface albedo), which have great impacts on the magnitude of data assimilation improvement. For instance, over Europe in winter, the number of observations is relatively small, and the observation error is relatively large. The remaining errors may also result from model errors such as too short lifetime of $NO_x$ through processes such as the $NO_2$+OH reactions

and the reactive uptake of $NO_2$ and $N_2O_5$ by aerosols (e.g., Lin et al., 2012b; Stavrakou et al. 2013). This will further be discussed in Section 5.3.

## 3.3 Seasonal and interannual variation

The underestimation in the simulated concentrations is most obvious in winter over most of the industrial regions, such as China, Europe, the United States, and Southern Africa. Data assimilation greatly reduced the wintertime low bias by 50-70 % over China, by about 50–90 % over the United States, and by 50–70 % over Southern Africa, as summarized in Table 2. Over Europe, the model's negative bias is reduced by about 10–80 % in summer, but the negative bias compared with the OMI retrievals mostly remains in winter (c.f., Section 5.3). Despite the persistent wintertime bias over Europe, the improved temporal correlation (from 0.64–0.89 in the model simulation to 0.90–0.95 in the data assimilation) confirms improved seasonality and year-to-year variation. Over India, the $NO_2$ columns in the model simulation do not reveal clear seasonal variation, whereas a significant seasonal variation is introduced by data assimilation, reflecting the observed high concentration in boreal winter-spring. The temporal correlation is largely improved over India (from -0.47–0.06 in the model simulation to 0.76–0.95 in the data assimilation).

The observed concentrations reveal large year-to-year variations over the industrial regions, which are generally underestimated in the model simulation (Fig. 3). Over China, the difference between the model simulation and the observations becomes significant after 2010, suggesting a larger underestimation in the a priori inventories in that time period, relative to the period before 2010. The observed concentrations reveal positive trends over China, with an exceptional decrease in 2009, followed by a rapid increase in 2010, and a decrease in 2014, as found by Cui et al. (2015), Duncan et al. (2016), and Krotkov et al. (2016). The data assimilation better captures the observed variations, as indicated by the better agreement in the linear trend (+40 %/decade in the OMI observation, +13 %/decade in the model simulation, and +28 %/decade in the data assimilation) and by the improved temporal correlation (from 0.85–0.94 to 0.95–0.99). Over the United States, the data assimilation removes most of the model's negative bias in 2005–2007 and reproduces the observed downward trend for the ten-year periods. These improved agreements suggest that the a posteriori emissions from data assimilation capture the actual anthropogenic emission variability.

The seasonal and year-to-year variations over Southeast Asia, and North and Central Africa are associated with changes in the biomass burning activity. Data assimilation improves the temporal variability, as confirmed by the improved temporal correlations (by 0.10–0.14 over North Africa, by 0.03–0.04 over Central Africa, and by 0.15–0.21 over Southeast Asia). Over Southeast Asia, the negative bias in the biomass burning season is largely removed by data assimilation. The systematic adjustments for North and Central Africa throughout the year suggests that the a priori emissions reasonably represent the seasonality of biomass burning activity, but emission factors might be underestimated in the a priori setting, as discussed in Section 4.

## 3.4 Vertical profiles

Figure 4 compares the vertical profiles with the aircraft observations during the INTEX-B, ARCTAS, DC3, and SEAC$^4$RS campaigns and with the ground-based lidar observations obtained during the DANDELIONS campaign. For all the profiles, the observed $NO_2$ concentrations are high in the boundary layer and decrease with height above the boundary layer in the troposphere. Both the model simulation and data assimilation reproduced these observed general features.

For the ARCTAS profile, the data assimilation has only a small effect on the lower and middle tropospheric $NO_2$ profiles, because of the large observational error of the $NO_2$ measurements at high latitudes. In contrast, the data assimilation mostly removed the model negative bias in the upper troposphere and lower stratosphere, mainly because of the MLS $O_3$ and $HNO_3$ data assimilation and through the use of the inter-species correlation that was determined using background error covariances estimated from ensemble model simulations (c.f., Section 2.2). An estimated inter-species correlation is demonstrated in Miyazaki et al. (2012b) in Fig. 3, which shows a strong positive correlation between the concentrations of $NO_2$ with those of $O_3$ and $HNO_3$, reflecting complex tropospheric chemical processes. The data assimilation widely influences the $NO_x$ and $NO_y$ species in both analysis and forecast steps. This improvement cannot be achieved using the $NO_2$ measurements only.

Compared with the INTEX-B and DC3 profiles, both the model and assimilation are too low in the middle/upper troposphere, whereas in the lower troposphere these are too high compared with the DC3 profile and too low compared with the INTEX-B profile. Compared with the SEAC$^4$RS profile, both the model and assimilation are too high in the lower troposphere. Because of the coarse model resolution (approximately 2.8°), the model has difficulty in representing the spatial footprint of the measurement, and this could cause large differences near the surface for comparisons at urban sites. The near-surface concentration will be sensitive to the model resolution owing to fine-scale emission distribution and transport, as well as non-linear chemical processes, as discussed in Valin et al (2011) and Miyazaki et al (2012a). The coarse model resolution may also make the improvements by data assimilation obscure.

During the DANDELIONS and aircraft campaigns, large variations in individual measurements along the flights were observed. Therefore we evaluate the variability as well as mean profiles using scatter plots. The right four panels in Fig. 4 show the scatter plots for an INTEX-B profile on March 9 in 2006 as an example and for the DANDELIONS measurements. For the INTEX-B profile, the data assimilation improves the agreement (i.e., the correlation and slope) with the observations in the lower and middle troposphere, except within the boundary layer (i.e., below 900 hPa). The correlation (from 0.324 to 0.455) and the slope (from 0.26 to 0.53) increased in the lower troposphere (900–750 hPa) by data assimilation. The improvements are also found for higher levels (750–600 hPa) and for other flights (not shown). The assimilation does not obviously change the model profile in the upper troposphere (600–300 hPa); the remaining negative bias could be attributed to errors in the model, such as in the chemical loss, $NO_y$ species partitioning, and atmospheric transport. For the DANDELIONS profiles, the data assimilation improves the agreement in the lower troposphere (e.g., the correlation and slope are increased from 0.14 to 0.46 and from 0.11 to 0.90, respectively, for 150–500 m), except near the surface (i.e., below 150 m).

## 4  Estimated surface $NO_x$ emissions

The a posteriori emissions were compared against the a priori emissions for the 2005–2014 period and against an independent emission inventory from EDGAR-HTAP v2 (Janssens-Maenhout et al., 2015) for the years 2008 and 2010. EDGAR-HTAP v2 was produced using nationally reported emissions combined with regional scientific inventories from the European Monitoring and Evaluation Programme (EMEP), Environmental Protection Agency (EPA), Greenhouse Gas-Air Pollution Interactions and Synergies (GAINS), and Regional Emission Inventory in Asia (REAS). For the comparison against EDGAR-HTAP v2, emissions from biomass burning and soils were obtained based on GFED version 3.1 and GEIA inventories; they were used in the a priori emissions.

### 4.1  Top-down vs. a priori global surface $NO_x$ emissions

The global distributions of the estimated emission sources are depicted in Fig. 5. As summarized in Table 3, the ten-year mean global total surface $NO_x$ emissions after data assimilation is 48.4 Tg N yr$^{-1}$, which is about 26 % higher than the a priori emissions (38.4 Tg N yr$^{-1}$). The positive analysis increment in global total emissions is attributable to an approximate +21 % increment in the Northern Hemisphere (NH, 20–90°N), a +35 % increment in the tropics (20°S-20°N), and a 42 % increment in the Southern Hemisphere (SH, 20–90°S). Strong positive increments are found over China (+39 %), the United States (+10 %), India (+22 %), and Southern Africa (+50 %). There are also positive increments in emissions over the biomass burning areas of Central Africa (+53 %) and Southeast Asia (+39 %). The a posteriori regional total emissions are clearly closer to the EDGAR-HTAP v2 emissions than the a priori emissions over China, the United States, and India. Since the same biomass burning and soil emission inventories are used in producing the total a priori and EDGAR-HTAP v2 emission data sets in this study, the emissions are similar between the two data sets over biomass burning and remote areas.

Fig. 6 depicts the global distribution of the linear trend during the ten-year period. The trend is negative over most of the United States, Europe, some parts of eastern China, South Korea, Japan, central and Southern Africa, Northern South America, with strong negative trends over the eastern United States, some parts of Europe (e.g., Northwest Europe, Po valley, and northern Spain), and Japan. Strong positive trends are found over China, India, Middle East, around Sao Paulo in Brazil, and around Jakarta in Indonesia.

Data assimilation reveals significant temporal variations (Fig. 7), including seasonal (Fig. 8) and interannual (Fig. 9) variations, in the emissions over major polluted regions. In northern mid-latitudes, the emissions are strongly enhanced in summer, and the timing of the summertime peak from data assimilation is earlier by 1–2 months over North America, Europe, and China (Fig. 8), as similarly found in our previous study (Miyazaki and Eskes, 2013). Applying the ratio of different emission categories within the a priori emissions for each grid point to the estimated emissions after data assimilation (only the total emission is optimized in our estimates), global total $NO_x$ emissions from soils are 7.9 Tg N yr$^{-1}$ for the a posteriori emissions in contrast to 5.4 Tg N yr$^{-1}$ yr$^{-1}$ for the a priori emissions. In line with recent studies by Hudman et al. (2012) and Vinken et al. (2014), our results suggest that the a priori emissions underestimate those by soils and misrepresent the seasonality.

Over biomass-burning areas, the time of the peak emissions does not change for most cases, suggesting that the a priori emissions describe the seasonality reasonably, but the systematic adjustment indicates large uncertainties in emission factors and biomass burnt estimates used in the inventories. The weak year-to-year variations in the a priori emissions are partly attributable to the use of climatology after 2011 (c.f., Sec. 2.1).

Despite the large year-to-year variations over many regions (c.f., Figs. 6 and 7), the global total emission is almost constant between 2005 ($47.9\,\mathrm{Tg\,N\,yr^{-1}}$) and 2014 ($47.5\,\mathrm{Tg\,N\,yr^{-1}}$), with a maximum in 2012 ($50.9\,\mathrm{Tg\,N\,yr^{-1}}$) and a minimum in 2008 ($46.7\,\mathrm{Tg\,N\,yr^{-1}}$). Over the ten-year period, the large emission increases over China, India, and the Middle East mostly compensate for the large emission decreases over the United State, western Europe, and Japan.

## 4.2   Top-down vs. a priori regional surface $NO_x$ emissions and their trends

**4.2.1   East Asia**

Data assimilation adjusts the total annual emissions from 4.47 to $6.21\,\mathrm{Tg\,N\,yr^{-1}}$ over China for the 2005–2014 period (Table 3), whereas the a posteriori emissions show good agreement with the EDGAR-HTAP v2 emissions ($6.19\,\mathrm{Tg\,N\,yr^{-1}}$ in the a posteriori emissions and $6.25\,\mathrm{Tg\,N\,yr^{-1}}$ in the EDGAR-HTAP v2 for 2008 and 2010). Our a priori inventory is too low over China, by about 40 %. The seasonal variation is largely corrected by data assimilation (Fig. 8), exhibiting maximum emissions

in January and June.

At the grid scale, the estimated emissions are higher than the a priori emissions over northern and eastern China, such as Beijing (+58 % at the nearest grid point), Tianjin (+97 %), Nanjing (+30 %), and around Guangzhou (+78 %), whereas they are lower around Chengdu and Chongqing (Fig. 10). In terms of the regional mean, the EDGAR-HTAP v2 is closer to the a posteriori emissions for China. However, there are disagreements at grid-scale around large cities, such as Shanghai (the a

posteriori minus EDGAR-HTAP v2 is -25 %), Guangzhou (+46 %), and Chongqing (-19 %), and also in South Korea around Seoul (+37 %) and in Japan around Tokyo (+13 %).

Our estimate of 12.5 TgN for July 2007 over East Asia (80-150°E, 10–50°N) is slightly larger than that of 11.0 TgN estimated using OMI observations (Zhao and Wang, 2009). The 6.6 TgN (8.0 TgN) estimated for July 2008 (January 2009) over east China (103.75–123.75°E, 19–45°N) from OMI and GOME-2 observations by Lin and McElroy (2010) is slightly smaller

than (larger than) our estimates of 7.4 TgN (7.4 TgN). We emphasize that the estimated emissions are strongly constrained by the assimilation of non-$NO_2$ measurements in our estimates. The estimated emissions for July 2008 over east China for the above-mentioned case from a $NO_2$-only assimilation (8.2 TgN) is 11 % larger than the estimate using multiple-species (7.4 TgN). The importance of multiple-species assimilation is further discussed in Sect. 5.1.

The estimated emission for China does not follow a simple linear increase, but rather increasing from 2005 to 2011 with a

slightly negative trend afterwards, as shown by Fig 9 and Fig. 11. The ten-year linear trend slope is estimated at +26 %/decade (Table 4). The difference in the estimated emission trend between the two time periods (2005–2010 and 2011–2014) are most commonly found across the country, which can be attributed to the competing influences of economic growth and emission controls (Cui et al., 2015). The temporal strong decrease in the estimated emissions in 2008 summer (Fig. 7) could be associated

with the Beijing Olympic games, as suggested by Mijling et al. (2009), Witte et al. (2009), and Worden et al. (2012). The trend for 2005–2010 over China is estimated at +3.0 %/year in our estimate, which is slightly smaller than the +4.0 %/year estimate using OMI measurements by Gu et al. (2013). The increase from 2008 to 2010 for China is larger in the a posteriori emissions ($+0.73 \, \mathrm{Tg \, N \, yr^{-1}}$) than in EDGAR-HTAP v2 ($+0.49 \, \mathrm{Tg \, N \, yr^{-1}}$).

5 As shown by Fig. 12, strong positive trends are found over large cities such as Wuhan (+42 %/decade), Nanjing (+35 %/decade), Tianjin (+35 %/decade), Chengdu (+56 %/decade), and over eastern China. A larger relative positive trend occurs over western China, especially over northwestern China (around 88–110°E, 37–48°N) where the rate of increase reaches +50–+110 %/decade at grid scale. Despite the general large positive trend for the ten-year period, the three largest cities in China show a net reduction or a small increase during 2005–2014; Beijing (-0.6 %/decade), Shanghai (-6.2 %/decade), and

10 Guangzhou (+4.5 %/decade), as commonly found in the observed $NO_2$ concentrations (Wang et al., 2015). In East Asia, the estimated emissions also show strong negative trends over major cities in Japan and South Korea; Tokyo (-48 %/decade), Osaka (-38 %/decade), and Seoul (-11 %/decade).

### 4.2.2 Europe

The total emissions for Europe are about 5 % higher in the a posteriori than in the a priori emissions (Table 3), which is

15 attributed to positive increments over some parts of western Europe, such as Belgium (+67 %), western Germany (+23 %), northern Italy (+62 %), and Istanbul (+40.3 %) (Fig. 10). The a posteriori emissions for Europe are higher than the EDGAR-HTAP v2 inventory by 17 % for 2008 and 2010, and the differences are large at the grid scale around London (+27 %), Belgium (+87 %), western Germany (+84 %), Paris (+27 %), Madrid (+55 %), northern Italy (+90 %), and Istanbul (+56 %). Both the a priori and EDGAR-HTAP v2 emission inventories show maximum emissions in summer (i.e., July), whereas the timing of

20 peak emission becomes earlier by 1 month after data assimilation (Fig. 8). The estimated seasonal amplitude is larger over Eastern Europe than over Western Europe by about 40 %, which suggests the possibility of more active summertime emissions from soil in Eastern Europe, as consistently revealed by Vinken et al. (2014).

 The estimated emissions for Europe show a slightly negative trend during 2005–2014, with a sharp decrease from 2009 to 2010 (Fig. 9). The estimated linear decrease for the ten-year period is small (-0.1 %/decade) for Europe (10°W–30°E, 35–

25 60°N), but is much larger (-8.8 %/decade) over Western Europe (10°W–17°E, 36–54°N), as summarized in Table 4. At the grid scale (Fig. 12), strong negative trends occur over large cities in Western Europe; Paris (-10 %/decade), northwestern France (-57 %/decade), London (-11 %/decade), Belgium (-24 %/decade), Athens (-22 %/decade), and over a region with many power plants in northern Spain (-45 %/decade) and Po valley (-52 %/decade). These variations are considered to be the result of the global economic recession and emission controls, as pointed out by Castellanos and Boersma (2012). The negative trends are

30 stronger during 2005–2010 than during 2011-2014 over some parts of western and southern Europe such as over northern Spain, northern Italy, and western Germany (Fig. 12). Strong negative emission trends over these regions were similarly found by Curier et al. (2014) for 2005–2010. Zhou et al. (2012) revealed that $NO_x$ emissions from Spanish Power plants have been strongly reduced for the 2004-2009 period because of emission abatement strategies, which is consistent with our estimates.

### 4.2.3 North America

The ten-year mean a posteriori emissions are higher than both the a priori ($5.73\,\mathrm{Tg\,N\,yr^{-1}}$ v.s $5.23\,\mathrm{Tg\,N\,yr^{-1}}$ for 2005–2014) and EDGAR-HTAP v2 ($5.26\,\mathrm{Tg\,N\,yr^{-1}}$ v.s $4.84\,\mathrm{Tg\,N\,yr^{-1}}$ for 2008 and 2010) emissions over the United States (Table 3). Positive increments are found over most remote areas and around the Southeast United States (e.g., +23 % near Atlanta) and most of the Western United States (e.g., +26 % near Denver), whereas negative increments are found around large cities such as New York (-28 %), Toronto (-17 %), Montreal (-19%), Houston (-19 %), and Los Angeles (-5 %) (Fig. 10). Despite the small adjustment for the ten-year mean regional total emissions, the data assimilation analysis increments for the regional total emission are strongly positive during 2005-2008, producing a long-term negative trend (Fig. 7). The timing of maximum emissions becomes earlier by 2 months (from July to May) due to data assimilation (Fig. 8). The summertime peak enhancement is obvious over remote regions such as high temperature agricultural land over the East South Central and the Southwestern United States, which suggests that the a priori emissions underestimates emissions from soil, as suggested by Oikawa et al. (2015) for the western Unites States. The estimated emissions are larger than the EDGAR-HTAP v2 emissions around large cities such as New York (+24 %), Chicago (+12 %), Denver (+35 %), Houston (+17 %), San Francisco (+74 %), and Los Angeles (+68 %) but are smaller over remote areas in the eastern and central United States for 2008 and 2010 (Fig. 10). The $0.73\,\mathrm{Tg\,N}$ estimated over the United States (130–70°W, 25–50°N) from ICARTT observations between 1 July and 15 August in 2004 (Hudman et al., 2007) is close to our estimates of 0.82 TgN for 1 July to 15 August in 2005. The 0.465 TgN estimated over the eastern United States (102–64°W, 22–50°N) from the OMI observations for March 2006 (Boersma et al., 2008a) is slightly smaller than our estimate of 0.502 TgN.

The a posteriori regional emissions for the United States show a strong negative trend during 2005–2014 (-29.4%/decade) (Table 4). The estimated trend for 2005–2012 (-32 %) in this study is close to that reported by Tong et al. (2015) using OMI measurements (-35 %). The ten-year linear trend is strongly negative over large cities such as New York (-48 %/decade), Boston (-42 %/decade), Chicago (-52 %/decade), Atlanta (-47 %/decade), Dallas (-19 %/decade), Houston (-25 %/decade), Denver (-16 %/decade), and Los Angeles (-46 %/decade) (Fig. 11). Lu et al. (2015) estimated that total OMI-derived $NO_x$ emissions over selected urban areas decreased by 49 % from 2005 to 2014, reflecting the success of $NO_x$ control programs for both mobile sources and power plants, with greater reductions before 2010 than after 2010. These variations are similarly found in our estimates (Fig. 12). Both the a posteriori and EDGAR-HTAP v2 emissions consistently reveal a decrease in the regional emissions for the United States from 2008 to 2010 (-0.34 and $-0.51\,\mathrm{Tg\,N\,yr^{-1}}$, respectively).

### 4.2.4 India

The ten-year total emissions from India are 22 % higher in the a posteriori emissions than in the a priori emissions (Table 3). The positive adjustment for the country's total emissions is large in spring, resulting in a Mar–June/July–September ratio of about $1.55\pm0.1$ (Fig. 8), which could be associated with the seasonality in open biomass burning (Venkataraman et al., 2006). The seasonal variation is mostly absent in the a priori and EDGAR-HTAP v2 inventories. The positive increment is large around large cities such as Lucknow (+110 %), Patna (+25 %), Mumbai (+50 %), Hyderabad (+16 %), and Madras (+21 %)

(Fig. 10). In contrast, the country's total emissions are about 10 % smaller in the a posteriori emissions than in the EDGAR-HTAP v2, with large negative biases (i.e., the a posteriori is smaller) around Delhi (-49 %) and southern India (-20−-70 %) and large positive biases over Lucknow (+68 %), Gwalior (+45 %), Raipur (+41 %), Mumbai (+12 %), and Hyderabad (+14 %) at grid scale (Fig. 10). These results suggest both EDGARv4 and EDGAR-HTAP v2 inventories largely underestimate emissions over some parts of India such as around Lucknow, Raipur, Mumbai, and also in Thailand around Bangkok (+26 % compared with the a priori emissions and +118 % compared with the EDGAR-HTAP v2 emissions) and Chiang Mai (+54 % and +66 %, respectively).

The a posteriori emissions for India increased continuously over the ten-year period, with a linear trend of +29 %/decade (Fig. 9). The positive trend is large across the country, with particularly strong increases around Lucknow (+29 %/decade), Kolkata (+47 %/decade), Raipur (+67 %/decade), and Madras (+40 %/decade) (Fig. 12). The positive emissions trend could be associated with increased thermal power plants in India, as pointed out by Lu and Streets (2013). In 2014, the regional total emissions for India (i.e., 3.46 $\mathrm{Tg\,N\,yr^{-1}}$) are comparable to (about 83 % of) the European-total emissions (i.e., 4.15 $\mathrm{Tg\,N\,yr^{-1}}$) and about 67 % of the United States-total emission (i.e., 5.17 $\mathrm{Tg\,N\,yr^{-1}}$). In contrast, tropospheric $NO_2$ columns over India are much lower compared to those in northern midlatitude polluted areas, as a result of the high values of temperature, photolysis rates, and specific humidity, leading to shorter $NO_2$ lifetimes throughout the year (Beirle at al., 2011).

### 4.2.5 Southern Africa

A large adjustment in $NO_x$ emissions is apparent in the Highveld region of Southern Africa with a factor of about 1.5 (table 3). The positive adjustment is relatively large in the austral summer (Fig. 8). The emissions from Southern Africa show a slight negative trend (-8 %/decade), with a temporary increase in 2006–2007, followed by a rapid decrease in 2009, and almost constant emissions afterwards (Fig. 9). The difference in emissions between 2008 and 2010 is small in EDGAR-HTAP v2 (+0.01 $\mathrm{Tg\,N\,yr^{-1}}$), whereas the a posteriori emissions show a negative trend (-0.09 $\mathrm{Tg\,N\,yr^{-1}}$ (2010-2008)) (Table 3). The ten-year linear trend reaches about -40 %/decade at grid scale over highly polluted areas. Duncan et al. (2016) highlighted a complex mixture of different emissions sources over Southern Africa. The various emission sources may have experienced different variations, and high resolution emission analysis is required to understand the detailed spatial variation in these emissions and to obtain unbiased emission estimates (Valin et al., 2011).

### 4.2.6 North and central Africa

Over North Africa, the ten-year mean emission increased by 40 % due to data assimilation from 2.07 to 2.90 $\mathrm{Tg\,N\,yr^{-1}}$ (Table 3). The positive increment is large from boreal winter to summer, producing the second maximum in July that is absent in the a priori emission (Fig. 8). The enhanced emissions for July and August are found throughout the 2005–2014 period and can mainly be attributed to emissions from the Sahel and Nigeria. This large positive increment may indicate an underestimation of soil $NO_x$ emissions in the a priori inventory. The short summer dry season in Nigeria may also lead to enhanced biomass burning emissions. The data assimilation largely corrects the spatial distribution during the peak season in January, with larger positive adjustments over the western (by about +60–+120 % at grid scale around 5W°–15W°) rather than

the eastern parts of North Africa (Fig. 5). The data assimilation also introduced a distinct year-to-year variation, reflecting the observed concentration variations associated with changes in biomass burning activity. The estimated emissions are high in 2005, 2006, 2008, and 2009, and low in 2010 (Fig. 7), which could be associated with drought events related to atmospheric variations such as ENSO (Janicot et al., 1996).

Over central Africa, the ten-year mean a posteriori emissions are larger than the a priori emissions by about 53 % (2.57 $\mathrm{Tg\,N\,yr}^{-1}$ v.s 1.68 $\mathrm{Tg\,N\,yr}^{-1}$) (Table 3). Large positive increments are found in the Congo region, with about +50–+150 % increases for the ten-year mean emissions at the grid scale (Fig. 5). The relative adjustment for the regional total emissions during the biomass burning season is +30–+40 % over central Africa and about +40 % over North Africa. These numbers may indicate a possible underestimation of the magnitude of fire-related emission factors in GFED v3. Although variation in the

seasonal emissions is different between North Africa and Central Africa (almost in opposite phase, reflecting the transition of the Intertropical Convergence Zone (ITCZ)), the year-to-year variation revealed by data assimilation is similar between the two regions. The temporal correlation of the annual total emission between North Africa and central Africa for the 2005-2011 period (when the GFED emissions are available) is estimated at 0.90 for the a posteriori emissions, and 0.01 for the a priori emissions. This result may suggest that year-to-year emission variations over the two regions are controlled in the same manner

by long-lasting atmospheric variations (e.g., ENSO), for which the a priori emissions have large uncertainties.

### 4.2.7   Southeast Asia

Over Southeast Asia, the data assimilation increases the annual mean emission by 45 % from 0.47 to 0.68 $\mathrm{Tg\,N\,yr}^{-1}$ (Table 3), with a large increase in boreal winter and spring (Fig. 8). The regional emission increment is positive over peninsular Malaysia (+20–+40 % for the ten-year mean emission), Borneo Island (+60–+100 %), and central and northern Thailand (+50–+80 %)

(Fig. 10). Because of the large adjustment in boreal winter and spring, the peak-to-peak seasonal variation for southeast Asia is enhanced by 20 % by data assimilation (Fig. 8). The a priori inventories reveal enhanced emissions in 2005, 2007, and 2010, reflecting year-to-year changes in biomass burning emissions, whereas data assimilation further increased them by up to 30 % (Fig. 7). The relative adjustment in other years (i.e. years with weaker biomass-burning activity) is even higher during the boreal winter and spring (with a factor of more than 2), which can largely be attributed to large positive increments over central

and northern Thailand. The Southeast Asia emissions can be characterized as a combination of various sources. Using the ratio between different emission categories in the a priori emission inventories at each grid point, the regional total emissions from anthropogenic sources, biomass burning, and soils are estimated at 0.51, 0.11, and 0.06 $\mathrm{Tg\,N\,yr}^{-1}$, respectively, which is 47, 32, and 58 % higher than the a priori emissions.

### 4.2.8   South America

Over South America, the ten-year mean regional total emissions are comparable between the a priori and a posteriori emissions, whereas the spatial distribution is largely corrected, with large positive increments over eastern Brazil (+50–+110 % at grid scale) and Peru (+90–+140 %) and negative increments over the central Amazon (up to -30 %) (Fig. 5). The seasonal variation of the regional total emission for South America is largely corrected by data assimilation (Fig. 8). A large decrease (by -30

%) occurs in the biomass burning season in August-September in all of the years, which might be the result of an overestimation of emissions by forest (i.e., deforestation) fires in dry conditions in the emissions inventory, as similarly investigated by Castellanos et al. (2014) using GFED v3. This is in contrast to the increased emissions over central Africa in the biomass burning season (c.f., Section 4.2.6). In contrast to the negative increments in the biomass-burning season, the emissions in the biomass burning off-season are increased by 30–60 % by data assimilation. Consequently, data assimilation decreased the seasonal amplitudes by 40 %. The year-to-year variations are similar between the a priori and a posteriori emissions (Fig. 7). As an exception, a large decrease in 2010 (with a 50 % decrease from 6.9 Tg N to 3.5 Tg N in August by data assimilation) suggests large uncertainty in fire-related emission factors in the major fire year (Bloom et al., 2015).

### 4.2.9 Other remote regions

The data assimilation may capture signals related to soil emissions, for which the inventories may have large uncertainties. For instance, the regional mean emissions over Australia are higher by about 40 %, with a large increase in boreal spring-early summer. The emissions are also higher over the central Eurasian continent, including eastern Europe and western China, and over the Sahel (Fig. 5), as was similarly found by Vinken et al. (2014). The global total $NO_x$ emissions by soils for the ten-year period are estimated at 7.9 Tg N $yr^{-1}$ $yr^{-1}$, in contrast to 5.4 Tg N $yr^{-1}$ $yr^{-1}$ for the a priori emissions. The results indicate large underestimates in the soil emission inventories over these regions. For instance, the nonlinear relationships between soil $NO_x$ emissions and time since fertilization, soil temperature, and soil moisture, are not properly considered in current inventories, as pointed out by Oikawa et al. (2015) for agricultural regions. Note that our estimate of 7.9 Tg N $yr^{-1}$ is smaller than other recent estimates (8.9 Tg N $yr^{-1}$ in Jaeglé et al. (2005), 8.6 Tg N $yr^{-1}$ in Steinkamp and Lawrence (2011), 10.7 Tg N $yr^{-1}$ in Hudman et al. (2012), and $12.9 \pm 3.9$ Tg N $yr^{-1}$ in Vinken et al. (2014)), which could partly be attributed to the assumed emission ratio between different categories for each model grid point, which is based on the a priori inventories and was not modified by the data assimilation in this study.

Among major industrialized areas, the Middle East has experienced a rapid increase in $NO_2$ levels (Lelieveld et al., 2015). Our estimates reveal a linear trend of +20 %/decade in $NO_x$ emissions and a 45 % positive adjustment from the a priori emissions for the Middle East (32–65°E, 12–40°N) during the ten-year period. Strong positive trends are found over major cities, such as Kuwait (+47 %/decade), Cairo (+29 %/decade), and Tehran (+37 %/decade). In contrast, the trend in the estimated emission over Dubai is negative (-6 %/decade). The rate of increase becomes larger after 2010 for many areas (Fig. 11), as found in observed $NO_2$ levels (Lelieveld et al., 2015). Lelieveld et al. (2015) suggested that a combination of air quality control and political factors has drastically altered the emission landscape of $NO_x$ in the Middle East.

Over the oceans, the data assimilation decreases the ten-year mean global total emissions from ships. In contrast, at the regional scale, data assimilation increments are positive over the oceans around Europe (Fig. 12), and a positive trend during 2005–2010 is introduced by data assimilation (Fig. 11, note that the estimated positive trend is more pronounced during 2005–2008, as commonly found by Boersma et al. (2015)). The overall negative increment as well as the positive increment around Europe may indicate an overestimate and an underestimation around Europe of ship emissions in the a priori inventories and errors in modelled chemical processes in the exhaust plumes (Vinken et al., 2011), which occur at fine scales relative to the

model grid. The overall negative increment can also be influenced by possible negative bias in $NO_2$ retrievals. Boersma et al. (2008a) showed negative bias over the ocean in $NO_2$ retrievals in version-1 DOMINO $NO_2$ retrievals, and the negative bias could not be fully removed in the version-2 DOMINO $NO_2$ retrievals (Boersma et al., 2011a).

## 5   Discussion

### 5.1   Importance of assimilating multiple trace gases

The differences between our $NO_x$ emissions estimates and previous studies, as discussed in Section 4, may be attributed to differences in the assimilated data, forecast model, and data assimilation approach. In particular, the use of non-$NO_2$ measurements is expected to improve emission estimates in our approach, as these affect the $NO_x$ chemistry and reduce model errors unrelated to surface emissions.

Table 5 compares the estimated emissions between the multiple-species data assimilation and a $NO_2$-only data assimilation. The estimated emissions differ in many regions if non-$NO_2$ data assimilation is considered because the ratio of predicted $NO_x$ emission and $NO_2$ column has been modified by non-$NO_2$ observations. The assimilation of non-$NO_2$ measurements leads to changes of up to about 70 % in the regional monthly-mean emissions. The estimated ten-year total regional emissions for South America and Australia are about 10 % lower in the multiple-species assimilation than in the $NO_2$-only assimilation. The RMSE

between the two estimates for the monthly total regional emissions is 15.5 % for central Africa, 16.5 % for Australia, and about 5–8 % for major polluted regions during the ten-year period. The estimated monthly mean emissions are mostly smaller in the multiple-species assimilation than in the $NO_2$-only assimilation, especially over the tropical and southern subtropical regions such as South America, central Africa, and Australia, suggesting that $NO_2$-only data assimilation tends to overcorrect the emissions from the a priori. The monthly total global emissions decrease by up to 6 TgN (in boreal summer) if non-$NO_2$ data

assimilation is considered.

We conducted Observing System Experiments (OSEs), and confirmed that the assimilation of individual data sets results in a strong influence on the estimated emissions. For instance, in January 2008, the TES $O_3$ assimilation led to substantial changes in the regional emissions over India (3.50 TgN in the $NO_2$-only assimilation and 3.15 TgN in the $NO_2$ and TES $O_3$ assimilation, in contrast to 3.16 TgN in the multi-species assimilation and 2.45 TgN in the a priori emissions), whereas other

non-$NO_2$ measurements (i.e., MOPITT and MLS) have less impact. Similar important contributions of TES $O_3$ measurements are found for South America in January 2008 (1.09 TgN in the $NO_2$-only assimilation and 0.91 TgN in the $NO_2$ and TES $O_3$ assimilation, in contrast to 0.90 TgN in the multi-species assimilation and 0.46 TgN in the a priori emissions). These changes in $NO_x$ emissions are associated with negative adjustments of $O_3$ by the TES assimilation over South America throughout the troposphere and positive adjustments of $O_3$ over India in the middle troposphere, and their influence on $NO_x$-OH-$O_3$ chemical

reactions and the $LNO_x$ source optimization, as discussed below.

The ten-year linear trend is also different over most industrial areas (Table 4). For instance, the positive trend for India is 34.3 %/decade in the $NO_2$-only assimilation, which is larger than the 29.2 %/decade in the multiple-species assimilation. For the United States, the negative trend is larger in the multiple-species assimilation (-29.4 %/decade) than in the $NO_2$-only

assimilation (-23.9 %/decade). These results confirm that the assimilation of measurements for species other than $NO_2$ provides additional constraints on the $NO_x$ emissions over both anthropogenic and biomass burning regions.

The improved representation of $NO_x$ emissions is confirmed by the better agreement of simulated $O_3$ concentrations with independent ozonesonde observations using $NO_x$ emissions from multiple-species assimilation than those using $NO_x$ emissions
from $NO_2$-only data assimilation, which was also demonstrated by Miyazaki and Eskes (2013). After 2010, TES $O_3$ retrievals were not assimilated because of the lack of standard observations. Even so, the optimized surface $NO_x$ emissions from the multiple-species assimilation improved agreements with TES $O_3$ ver. 6 special observations during 2011–2014 for most locations (Table S1). These results indicate that multiple-species measurements provide important information for improving surface $NO_x$ source estimations and improve the chemical consistency including the relation between concentrations and the
estimated emissions. Note that the emissions of $O_3$ precursors other than $NO_x$, such as VOCs, and various model processes in atmospheric transport and chemistry influence the model performance. The impact of using the optimized $NO_x$ emissions may vary with models (e.g., given different forecast errors of $NO_2$ and $O_3$). The optimization of additional precursors emissions and the improvement of the forecast model could be important for improving $O_3$ simulations, as discussed in our previous studies (Miyazaki et al., 2012b; 2015).

$LNO_x$ sources are important for a realistic representation of tropospheric $NO_2$ columns, which are optimized from data assimilation in our framework. Using the multiple-species data assimilation, the ten-year mean global $LNO_x$ source amount was estimated at $5.8\,\mathrm{Tg\,N\,yr^{-1}}$, in contrast to $5.3\,\mathrm{Tg\,N\,yr^{-1}}$ estimated from the model simulation and $6.3\pm1.4\,\mathrm{Tg\,N\,yr^{-1}}$ in our previous data assimilation estimate (Miyazaki et al., 2014). The data assimilation increments for $LNO_x$ sources are large and mostly positive in the middle and upper troposphere in the NH and the TR, in which non-$NO_2$ measurements with
different vertical sensitivities provided important constraints. Through its influence on simulated tropospheric $NO_2$ columns, for instance, the inclusion of the $LNO_x$ source optimisation altered the surface $NO_x$ emission estimates over eastern China by up to 12% in summer. Moreover, surface CO emissions increased by 10 % in the NH by the assimilation of MOPITT CO measurements in our system. Both optimised $LNO_x$ sources and CO emissions reveal enhanced seasonal and interannual variations over many regions after data assimilation, providing important constraints on long-term estimates of surface $NO_x$
emissions, through their influence on OH and thus the $NO_x$ chemical lifetime.

Figure 13 shows changes in OH concentrations ($\Delta$OH) in the lower troposphere in the boreal summer (averaged over June–August) due to data assimilation. The multiple-species assimilation changes the global OH distribution, increasing OH globally. As summarised in Table 6, the regional impact is large (greater than +20 %) in tropical regions such as over the Middle East, Southeast Asia, and Central and North Africa, and over industrial areas (greater than +10 %), such as over China,
the United States, and India. These changes in OH concentrations are influenced by changes in $NO_x$ emissions through the assimilation of $NO_2$ measurements, but the assimilation of non-$NO_2$ measurements is also important. Fig. S2 demonstrates that the assimilation of non-$NO_2$ measurements acts to decrease the OH concentration in the lower and middle troposphere for most regions in June 2008. The TES assimilation mostly reduces the $O_3$ concentration in the tropics, which leads to a decrease of OH concentrations. In contrast, the TES assimilation acts to increase the OH concentration in the NH extratropics in the lower
and middle troposphere. The assimilation of MOPITT CO acts to decrease the OH concentration in the NH, because of the

increased surface CO emissions. The ten-year mean NH/SH OH ratio is estimated at 1.19 in the multiple-species assimilation, in contrast to 1.27 in the MIROC model simulation and 1.22 in the $NO_2$-only assimilation, which is closer to 0.97$\pm$0.12 estimated based with the help of methyl chloroform observations (a proxy for OH concentrations) by Patra et al. (2014).

To elucidate the changes in the $NO_x$ chemical lifetime, Table 6 compares the lower tropospheric OH concentration and the ratio of the regional mean surface $NO_x$ emissions and lower tropospheric $NO_2$ concentrations (averaged from the surface to 790 hPa) between the multiple-species data assimilation and the model simulation ($\Delta NO_x$-emi/$NO_2$) in the boreal summer. The multiple-species assimilation leads to an increase in the OH concentration in the troposphere. Meanwhile, the increased ratio of $NO_x$ to $NO_2$ (i.e. increased fraction of NO) in the multiple-species assimilation compared to the model simulation indicates that the $HO_2 + NO \rightarrow NO_2 + OH$ reaction, which is the source of OH, is enhanced in the multiple-species assimilation. It is also found that the assimilation of non-$NO_2$ measurements suppress these changes for most regions in both the OH concentration (c.f., Fig. S2) and the $NO_x$-emi/$NO_2$ ratio. For instance, the ten-year mean ratio over Central Africa is increased by 16.5 % in the multiple-species assimilation, in contrast to the 19.3 % increase in the $NO_2$-only assimilation.

These results suggest that $NO_x$ chemical lifetime is decreased because of increased OH concentrations (through the $NO_2$+OH reaction, which acts as the main sink of $NO_x$) in the multiple-species data assimilation (and also in the $NO_2$-only assimilation) than in the model simulation over most industrial and biomass burning areas. It is also suggested for many regions $NO_x$ chemical lifetime is longer in the multiple-species assimilation than in the $NO_2$-only assimilation, because of decreased OH concentrations by the assimilation of non-NO2 measurements. These changes, together with the increased $LNO_x$ sources, could explain the smaller $NO_x$ emissions in the multiple-species assimilation than in the $NO_2$-only assimilation in many cases (c.f., Table 5). These results demonstrate the utility of the multiple-species assimilation to constrain the tropospheric chemistry (i.e., chemical regime) controlling $NO_x$ variations and to improve surface $NO_x$ emission inversions.

## 5.2 Impact of assimilating $NO_2$ observations from multiple instruments

Unlike most previous studies that used $NO_2$ retrievals from a single sensor, we assimilated multiple $NO_2$ measurements to constrain surface $NO_x$ emissions. When assimilating OMI retrievals only, the larger discrepancies with respect to the SCIA-MACHY and GOME-2 retrievals for some regions may be attributed to errors in the simulated diurnal $NO_2$ variations, since both emission factors and tropospheric concentrations of $NO_x$ are constrained only in the early afternoon in this case. When assimilating multiple $NO_2$ measurements, the application of the correction factor ($Etc$) for the emission diurnal variability function ($Et$) modified the shape of the diurnal emission variability (Fig. 1), which improved the agreement with multiple $NO_2$ retrievals in both the morning and afternoon for many cases. The global RMSE for monthly mean tropospheric $NO_2$ column is reduced by 8 % compared to the OMI retrievals and by 13 % compared to the SCIAMACHY in January 2005 by assimilating multiple $NO_2$ measurements with applying $Etc$, compared to the case with the OMI retrievals only. The estimated monthly regional emissions constrained by the three retrievals deceased by 18 % over Europe and by 9 % over Australia in January 2005 compared to those from the OMI retrievals only.

As shown in Fig. 14 and Table 7, the estimated $Etc$ is negative for most industrial regions such as Europe and North America, and over biomass burning areas, such as southeast Asia. The large adjustments ($Etc$=-0.3$-$-0.4, for which the daily mean hourly

emission value is 1) for the industrialized areas suggest that a positive adjustment to the assumed diurnal emission variability is required between 7:30–10:30 (and then a negative adjustment for emissions between 10:30–13:30), probably due to larger underestimations of emissions (e.g., morning traffic rush). Large negative values of $Etc$ are also found over northern China including Inner Mongolia, northern India, and the Middle East, where various emission sources (not only mobile sources with

morning peaks) could be important. For instance, over Inner Mongolia, the estimated emissions show a positive trend over the past decade (around 110°E, 41°N in Fig 12), which could be associated with increased emissions by power plants and industries without morning peaks. These results suggest a larger negative bias in simulated tropospheric $NO_2$ column in the morning than in the afternoon, associated with errors in the chemical lifetime and atmospheric transports (e.g., boundary layer development) and also associated with biases between the different $NO_2$ retrievals. Thus, the model errors could artificially

affect the diurnal emission variability. The optimized $Etc$ for biomass-burning and soil emission dominant regions are mostly slightly negative, which may suggest that the applied diurnal emission variability with an afternoon maximum (see Section 2.1) was inappropriate for some regions. In contrast, they are positive for most of the ocean. These results suggest the need to not only correct diurnal $NO_2$ variations, but also account for the differences in the sampling and bias between OMI and other instruments as well as the influences of model errors. Future geostationary satellite missions such as Sentinel-4, GEMS,

and TEMPO will be able to provide dramatically more systematic constraints on diurnal emission variability and observational information.

### 5.3    Possible error sources

Biases in satellite retrievals and modeling affect the magnitude of estimated emissions. Miyazaki et al. (2012a) demonstrated that possible biases (up to 40 %) in the $NO_2$ retrieval alter regional $NO_x$ emissions by 5–45 %. The emission estimates may

also be sensitive to measurement biases for species other than $NO_2$. For example, a bias correction for the positive bias in the TES $O_3$ profiles altered monthly $NO_x$ emissions by 1–11 % at the regional scale (Miyazaki and Eskes, 2013). Discontinuities in the assimilated measurements (e.g., lack of most TES retrievals after 2010, OMI row anomaly since January 2009, and the limited data coverage of SCIAMACHY (before February 2012) and GOME-2 (after January 2007)) may also affect long-term emission estimates.

Estimated emissions are sensitive to the choice of forecast model and its resolution. Our analysis using a different forecast model (CHASER versus MIROC-Chem) showed up to 20 % difference in monthly $NO_x$ emissions at the regional scale. Meanwhile, negative biases remain in tropospheric $NO_2$ columns over industrial regions, such as China, Europe, the United States, and Southern Africa, using either model and data assimilation. The inadequacies of the improvements in simulated tropospheric $NO_2$ columns could be related to model biases in the $NO_x$ chemical lifetime (e.g., Stavarakou et al., 2013) and

may also be partly attributed to the small number of observations and large observation errors for highly polluted cases (Fig. S1). Over polluted areas, observation errors increase almost linearly with the retrieved concentrations for most cases, and large observation errors may lead to the insufficient improvements by data assimilation for highly polluted cases. The remaining error may indicate a possible bias in the estimated emissions.

For example, over Europe, the increased wintertime negative bias against OMI retrievals (in contrast to the reduced bias against SCIAMACHY retrievals) in 2009 and 2010 could also be associated with difficulties in correcting the diurnal emission variation. For that time period over northern Europe, the number of OMI observations used for data assimilation is greatly reduced and observation errors are significantly increased, whereas those of SCIAMACHY vary differently (Fig. S1). More observational data (e.g., from ground-based measurements) may be required to further constrain surface $NO_x$ emissions for cloudy and snow-covered conditions and for high latitudes. Meanwhile, the diurnal variability correction scheme may need to be refined to further improve the agreement with various overpass time measurements.

Meanwhile, coarse resolution models are known to have negative biases in $NO_2$ over large sources (Valin et al., 2011). The emissions estimated at the T42 resolution in this study could potentially be overestimated over polluted areas, whereas the contrast between rural and urban areas could be underestimated. A high-resolution forecast model is important to accurately simulate nonlinear effects in $NO_2$ loss rate, while also providing insights into individual emission sources, such as power plants (e.g., de Foy et al., 2015).

Although the assimilation of multiple-species data influences the representation of the entire chemical system (Miyazaki et al., 2012b, 2015), the influence of model and observation errors remains a concern. In the multiple-species data assimilation, model performance is critical for the correct propagation of observational information between chemical species and to improve the emission estimation, whereas biases in any of the measurement data sets (including non-$NO_2$ measurements) may seriously degrade the emission estimation (Miyazaki et al., 2013). Improvements in the model, data assimilation scheme, and retrieved observations are essential to reduce the uncertainty on the emission estimates from the multiple-species data assimilation.

## 5.4   Trends in $NO_2$ concentrations and $NO_x$ emissions

We emphasize that the observed concentration variations do not necessarily correlate linearly with surface emissions, as similarly investigated by other inversion studies (e.g., Lamsal et al., 2011; Castellanos et al., 2012; Turner et al., 2012; Vinken et al., 2014). As summarised in Table 4, linear trends are significantly different between the observed concentrations and estimated emissions. The positive trend is larger in the observed $NO_2$ concentration (+39.6 %/decade) than in the emission estimates (+26.0 %/decade) for China, whereas the negative trend is larger in the emission estimates (-29.4 %/decade) than in the observed $NO_2$ concentration (-6.3 %/decade) for the United States. The relation between observed $NO_2$ concentration and estimated $NO_x$ emissions varies seasonally, as similarly expressed by Zhang et al. (2007), and the differences can be much larger at the grid scale. The results indicate that an accurate estimation of the long-term emission trends requires an emission-concentration relationship that explicitly accounts for tropospheric chemistry and non-$NO_2$ concentrations afforded by advanced data assimilation techniques (see Section 5.1). These year-to-year variations in the observed $NO_2$ concentrations have previously been reported by Duncan et al. (2016) and Krotkov et al. (2016).

These results also suggest that the tropospheric chemical regime may have changed over the ten-year period. For instance, over Europe, the linear trend is positive for the observed $NO_2$ concentration (+13.6 %/decade for all of Europe and +7.5 %/decade for western Europe in OMI) and is negative for the emission estimates (-0.1 %/decade and -8.8 %/decade, respectively). This suggests that $NO_2$ may have become longer-lived or has become a larger fraction of $NO_x$ over Europe over the

past decade. In fact, the lower tropospheric OH concentrations show slight negative trends (by up to -5 %/decade) over most of Western Europe over the past decade (figure not shown). Another possible explanation is that a shift in $NO_2$:$NO_x$ emission ratios related to the increasing share of European diesel cars could have occurred. Further efforts are required to explain the long-term variations of the tropospheric chemical regime and its causal mechanisms. Note that the linear trend in the observed concentration is different between the instruments over Europe (c.f., Fig. 3).

## 6  Conclusions

Global surface nitrogen oxides ($NO_x$) emissions are estimated for the ten-year period between 2005–2014 from the assimilation of multiple satellite datasets: tropospheric $NO_2$ columns from OMI, GOME-2, and SCIAMACHY; $O_3$ profiles from TES; CO profiles from MOPITT; and $O_3$ and $HNO_3$ profiles from MLS. The daily emission inversion is performed based on the ensemble Kalman filter data assimilation, which simultaneously optimises chemical concentrations of various species and emission sources of several precursors. Within the simultaneous emission and concentration optimisation framework, the analysis increment directly produced via chemical concentrations plays an important role in reducing model–observation mismatches arising from model errors unrelated to emissions, which can be expected to improve emission inversion. The assimilation of measurements for species other than $NO_2$ provides additional constraints on the $NO_x$ emissions over both anthropogenic and biomass burning regions, leading to changes in the regional monthly-mean emissions of up to 70 %. The impact of non-$NO_2$ measurements varied largely with season, year, and region. In addition to daily emission factors, the diurnal emission variability function was optimised using multiple $NO_2$ retrievals, obtained in the morning (SCIAMACHY and GOME-2) and afternoon (OMI). The emission correction largely improved the agreement with observed tropospheric $NO_2$ columns, at both the seasonal and interannual time scales.

The ten-year mean global total surface $NO_x$ emissions after data assimilation is 48.4 $Tg\,N\,yr^{-1}$, which is 26 % higher than a priori emissions based on bottom-up inventories. The optimised ten-year mean emissions are higher over most industrialised areas. The data assimilation corrected the timing and strength of emissions from biomass burning, such as over central Africa (the ten-year mean regional emission is 1.68 $Tg\,N\,yr^{-1}$ in the a priori emissions and 2.57 $Tg\,N\,yr^{-1}$ in the a posteriori emissions), North Africa (2.07 $Tg\,N\,yr^{-1}$ v.s 2.90 $Tg\,N\,yr^{-1}$), Southeast Asia (0.47 $Tg\,N\,yr^{-1}$ v.s 0.68 $Tg\,N\,yr^{-1}$), and South America (1.00 $Tg\,N\,yr^{-1}$ v.s 1.04 $Tg\,N\,yr^{-1}$), suggesting a large uncertainty in fire-related emission factors in the emission inventories. At northern mid-latitudes and over Australia, the emissions are largely enhanced during summer, suggesting an important underestimation of soil sources in the a priori inventory. Using the emission ratio between different categories in the a priori emission inventories, the global total soil $NO_x$ emission for the 2005–2014 period is estimated at 7.9 $Tg\,N\,yr^{-1}\,yr^{-1}$, which is much higher than the a priori estimate of 5.4 $Tg\,N\,yr^{-1}\,yr^{-1}$. This soil $NO_x$ emission estimate may nevertheless be conservative, because the ratio between the source categories is kept fixed in our approach.

The estimated regional total emissions show strong positive trends over India (+29 %/decade), China (+26 %/decade), and the Middle East (+20 %/decade), and negative trends over the United States (-29.4 %/decade), Southern Africa (-8.2 %/decade), and western Europe (-8.8 %/decade). At the grid scale, strong positive trends are found over large cities in China (e.g., Wuhan

(+42 %/decade), Chengdu (+56 %/decade), northwestern China (+50–+110 %/decade)), India (e.g., Kolkata (+47 %/decade), Raipur (+67 %/decade), Madras (+40 %/decade)), the Middle East (e.g., Kuwait (+47 %/decade), Tehran (+37 %/decade)), and Brazil (Sao Paulo (+40 %/decade)), whereas large negative trends are found in Europe (e.g., northern Spain (-45 %/decade), Po Valley (-52 %/decade)), the United States (e.g., New York (-48 %/decade), Boston (-42 %/decade), Chicago (-52 %/decade),

Atlanta (-47 %/decade), Los Angeles (-46 %/decade)), and Japan (e.g., Tokyo (-48 %/decade), Osaka (-38 %/decade)). The yearly mean emissions for China reveal a large positive trend from 2005 to 2011, subsequently decreasing through 2014. For the United States and some parts of Europe, the negative trends are larger during 2005–2010 than 2011–2014. These changes are more variable as a result of the global economic recession and emission controls. Despite the large year-to-year variations over many regions, the global total emission is almost constant between 2005 (47.9 TgN) and 2014 (47.5 TgN).

The estimated emissions have great potential to contribute to better understanding of precursor variability influences on observed air quality (e.g., tropospheric $O_3$) variations and associated climate impacts. The obtained emission data is also crucial to evaluate bottom-up inventories. The consistent data set comprising emissions and concentrations of various species, which were obtained from our simultaneous data assimilation framework, provides comprehensive information on atmospheric environmental variations, associated with both human and natural activity. Meanwhile, our results suggested that more observa-

tional constraints would be required to improve the global emission estimates. Observational information from future satellite missions such as TROPOMI and sensors on board geostationary satellites (Sentinel-4, GEMS, and TEMPO) in conjunction with exploitation of existing sounders, e.g., IASI and CrIS, can be expected to add constraints on more detailed spatiotemporal variability in surface $NO_x$ emissions and its impact on air quality (Bowman, 2013).

*Acknowledgements.* We acknowledge the free use of tropospheric $NO_2$ column data from the SCIAMACHY, GOME-2, and OMI sensors

from www.temis.nl. We also acknowledge the use of data products from the NASA AURA and EOS Terra satellite missions. We would also like to thank the editor and two anonymous reviewers for their valuable comments. This work was supported through JSPS KAKENHI grant numbers 15K05296 and 26220101 and Coordination Funds for Promoting AeroSpace Utilization by MEXT, JAPAN.

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

**Table 1.** Comparisons of tropospheric $NO_2$ columns between data assimilation and satellite retrievals: OMI for the period 2005–2014, SCIAMACHY for the period 2005–2011, and GOME-2 for the period 2007–2014. Shown are the global spatial correlation (S-Corr), the mean bias (BIAS: the data assimilation minus the satellite retrievals) and the root-mean-square error (RMSE) in $10^{15}\,\mathrm{molec\,cm^{-2}}$. The model simulation results (without data assimilation) are also shown in brackets.

|  | OMI | SCIAMACHY | GOME-2 |
| --- | --- | --- | --- |
| S-Corr | 0.98 | 0.95 | 0.95 |
|  | (0.94) | (0.86) | (0.87) |
| BIAS | +0.00 | +0.03 | −0.02 |
|  | (−0.08) | (−0.04) | (−0.18) |
| RMSE | 0.23 | 0.52 | 0.46 |
|  | (0.38) | (0.75) | (0.91) |

**Table 2.** The monthly mean bias and temporal correlation of regional mean tropospheric $NO_2$ columns: the data assimilation minus the satellite retrievals from OMI for the period 2005–2014, SCIAMACHY for the period 2005–2011, and GOME-2 for the period 2007–2014 in $10^{15}\,\text{molec}\,\text{cm}^{-2}$. The results of the model simulation (without data assimilation) are also shown in brackets.

| | Bias | | | Temporal correlation | | |
|---|---|---|---|---|---|---|
| | OMI | SCIAMACHY | GOME-2 | OMI | SCIAMACHY | GOME-2 |
| China | -0.75 | -2.60 | -1.68 | 0.99 | 0.96 | 0.95 |
| | (-1.98) | (-4.33) | (-3.46) | (0.94) | (0.91) | (0.85) |
| Europe | -0.45 | -0.11 | -0.23 | 0.95 | 0.93 | 0.90 |
| | (-0.45) | (-0.06) | (-0.23) | (0.89) | (0.64) | (0.70) |
| USA | -0.16 | -0.11 | -0.14 | 0.95 | 0.88 | 0.77 |
| | (-0.31) | (-0.22) | (-0.27) | (0.83) | (0.56) | (0.54) |
| S-America | 0.00 | 0.06 | -0.16 | 0.98 | 0.96 | 0.88 |
| | (-0.12) | (-0.10) | (-0.44) | (0.83) | (0.84) | (0.79) |
| N-Africa | -0.06 | -0.03 | -0.07 | 0.98 | 0.94 | 0.82 |
| | (-0.27) | (-0.22) | (-0.33) | (0.84) | (0.82) | (0.72) |
| C-Africa | -0.07 | -0.14 | -0.14 | 0.99 | 0.98 | 0.94 |
| | (-0.35) | (-0.41) | (-0.48) | (0.96) | (0.94) | (0.90) |
| S-Africa | -0.28 | -1.04 | -1.15 | 0.98 | 0.90 | 0.90 |
| | (-0.78) | (-1.60) | (-1.83) | (0.92) | (0.84) | (0.76) |
| SE-Asia | -0.20 | -0.05 | -0.14 | 0.98 | 0.93 | 0.88 |
| | (-0.38) | (-0.22) | (-0.39) | (0.83) | (0.74) | (0.67) |
| Australia | 0.03 | 0.05 | -0.05 | 0.96 | 0.92 | 0.87 |
| | (-0.11) | (-0.06) | (-0.21) | (0.81) | (0.73) | (0.64) |
| India | -0.12 | -0.02 | -0.01 | 0.95 | 0.92 | 0.76 |
| | (-0.28) | (-0.15) | (-0.25) | (0.06) | (-0.47) | (-0.40) |

**Table 3.** The regional ten-year mean $NO_x$ emissions (in $Tg\,N\,yr^{-1}$) obtained from the a priori emissions, a posteriori emissions, and the relative difference between these two emissions (in %) for the period 2005-2014 (left columns). The results are also shown for EDGAR-HTAP v2 emissions (as a reference) averaged over the years 2008 and 2010, the a posteriori emissions (the same results as in the left columns, but averaged over the years 2008 and 2010), and the relative difference between these two estimates (in %) (central columns), and for their difference from 2008 to 2010 (right columns). The results are also shown for the Northern Hemisphere (NH, 20–90°N), the tropics (TR, 20°S–20°N), the Southern Hemisphere (SH, 90–20°S), and the globe (GL, 90°S–90°N).

| | 2005–2014 average | | | 2008 and 2010 average | | | 2010 minus 2008 | |
|---|---|---|---|---|---|---|---|---|
| | A priori | A posteriori | Difference (%) | EDGAR-HTAP v2 | A posteriori | Difference (%) | EDGAR-HTAP v2 | A posteriori |
| China | 4.47 | 6.21 | 38.9 | 6.25 | 6.19 | -0.9 | 0.49 | 0.73 |
| Europe | 4.07 | 4.23 | 3.9 | 3.36 | 3.92 | 16.7 | -0.07 | 0.26 |
| USA | 5.23 | 5.73 | 9.6 | 4.84 | 5.26 | 8.7 | -0.51 | -0.34 |
| S-America | 1.00 | 1.04 | 4.0 | 1.14 | 1.12 | -1.8 | 0.89 | 0.14 |
| N-Africa | 2.07 | 2.90 | 40.1 | 2.01 | 2.96 | 47.3 | -0.09 | -0.82 |
| C-Africa | 1.68 | 2.57 | 53.0 | 1.70 | 2.68 | 57.6 | 0.09 | -0.62 |
| S-Africa | 0.46 | 0.60 | 50.0 | 0.37 | 0.72 | 94.6 | 0.01 | -0.09 |
| SE-Asia | 0.47 | 0.68 | 44.7 | 0.41 | 0.65 | 58.5 | 0.11 | 0.03 |
| Australia | 1.07 | 1.49 | 39.3 | 0.85 | 1.37 | 61.2 | -0.09 | -0.31 |
| India | 2.60 | 3.18 | 22.3 | 3.37 | 3.00 | 11.0 | 0.24 | 0.09 |
| NH | 24.90 | 30.06 | 20.7 | 26.40 | 28.95 | 9.7 | 0.32 | 1.38 |
| TR | 10.89 | 14.66 | 34.6 | 11.15 | 14.55 | 30.5 | 1.23 | -1.84 |
| SH | 2.60 | 3.69 | 41.9 | 2.16 | 3.69 | 70.8 | -0.08 | -0.27 |
| GL | 38.38 | 48.41 | 26.1 | 39.71 | 47.19 | 18.8 | 1.48 | -0.74 |

**Table 4.** Linear trend (in % per decade) of the regional a posteriori $NO_x$ emissions from the multiple-species assimilation (left column) and $NO_2$-only assimilation (central column), and of the regional mean tropospheric $NO_2$ columns from OMI (right column) for the period 2005-2014.

| | $NO_x$ emission (multiple-species) | $NO_x$ emission ($NO_2$-only) | OMI $NO_2$ |
|---|---|---|---|
| China | 26.0 | 27.3 | 39.6 |
| Europe | -0.1 | -1.4 | 13.6 |
| W-Europe | -8.8 | -10.0 | 7.5 |
| USA | -29.4 | -23.9 | -6.3 |
| S-America | -12.2 | -0.4 | 2.8 |
| N-Africa | -13.6 | -3.3 | 3.4 |
| C-Africa | -4.2 | 6.7 | 7.1 |
| S-Africa | -8.2 | 0.9 | 2.2 |
| SE-Asia | -0.3 | 13.0 | 13.0 |
| Australia | 1.3 | 10.2 | -1.3 |
| India | 29.2 | 34.3 | 25.0 |

**Table 5.** Difference between the a posteriori emissions from the multiple-species assimilation and $NO_2$-only assimilation. Relative difference for the regional ten-year mean emissions (left column), RMSE for the monthly regional emissions (central column), and range of relative difference for the monthly regional emissions (right column) are shown.

| | Mean diff. [%] | RMSE [%] | Range [%] |
|---|---|---|---|
| China | 1.3 | 5.2 | -10.8 – +16.2 |
| Europe | 1.9 | 6.4 | -18.1 – +16.0 |
| USA | -0.9 | 6.1 | -20.2 – +13.2 |
| S-America | -10.0 | 15.3 | -67.2 – +19.8 |
| N-Africa | -4.6 | 8.4 | -38.8 – +11.0 |
| C-Africa | -7.1 | 15.5 | -42.0 – +16.6 |
| S-Africa | -1.6 | 10.1 | -26.2 – +18.7 |
| SE-Asia | 4.9 | 11.2 | -59.8 – +34.2 |
| Australia | -10.1 | 16.5 | -69.1 – +14.2 |
| India | 2.2 | 8.1 | -23.6 – +22.4 |
| NH | -0.6 | 4.4 | -16.4 – +9.9 |
| TR | -3.6 | 7.5 | -26.6 – +9.6 |
| SH | -8.7 | 13.2 | -42.1 – +12.4 |
| GL | -2.2 | 4.8 | -17.3 – +-6.8 |

**Table 6.** Regional and ten-year mean difference in lower tropospheric OH concentration averaged below 790 hPa ($\Delta$ OH) and the ratio of surface $NO_x$ emission and lower tropospheric $NO_2$ concentration averaged below 790 hPa ($\Delta NO_x$-emi/$NO_2$) between the data assimilation run and the model simulation in the boreal summer (averaged over June–August) over the 2005–2014 period.

| | $\Delta$OH (%) | $\Delta NO_x$-emi/$NO_2$ (%) |
|---|---|---|
| China | 14.3 | 9.1 |
| Europe | 4.9 | 3.8 |
| USA | 10.7 | 0.9 |
| S-America | 21.5 | -14.8 |
| N-Africa | 20.3 | 28.5 |
| C-Africa | 21.1 | 16.5 |
| S-Africa | 20.0 | -2.1 |
| SE-Asia | 14.1 | 7.2 |
| Australia | 23.5 | 3.0 |
| India | 10.8 | -0.7 |
| NH | 9.7 (4.3) | 9.1 (-5.9) |
| TR | 13.1 (2.2) | 16.5 (-0.9) |
| SH | 23.8 (2.0) | 11.3 (-12.9) |
| GL | 12.2 (7.4) | 9.3 (2.0) |

**Table 7.** Regional and ten-year mean correction factor for the emission diurnal variability ($Etc$) for 2005–2014.

| | $Etc$ |
|---|---|
| China | -0.44 |
| Europe | -0.37 |
| USA | -0.33 |
| S-America | -0.06 |
| N-Africa | -0.03 |
| C-Africa | -0.08 |
| S-Africa | -0.20 |
| SE-Asia | -0.04 |
| Australia | -0.03 |
| India | -0.26 |

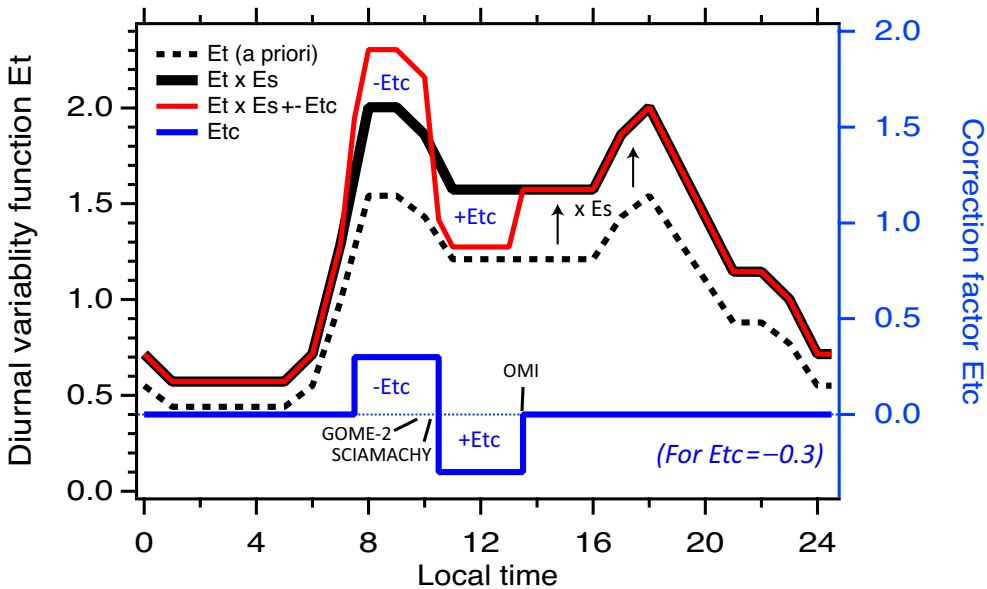

**Figure 1.** Schematic diagram of the correction scheme for the emission diurnal variation for a case with $Etc = -0.3$. The black dotted time represents the a priori emission diurnal variability function ($Et$) for anthropogenic emissions. The black solid line represents the a posteriori emission variation after applying the daily emission scaling factor ($Et \times Es$). The blue line represents the correction factor for the emission diurnal variability ($Etc$). The red line represents the a posteriori emission variation after applying the daily emission scaling factor and the correction factor for the emission diurnal variability ($Et \times Es - Etc$ for 07:30–10:30, and $Et \times Es + Etc$ for 10:30–13:30).

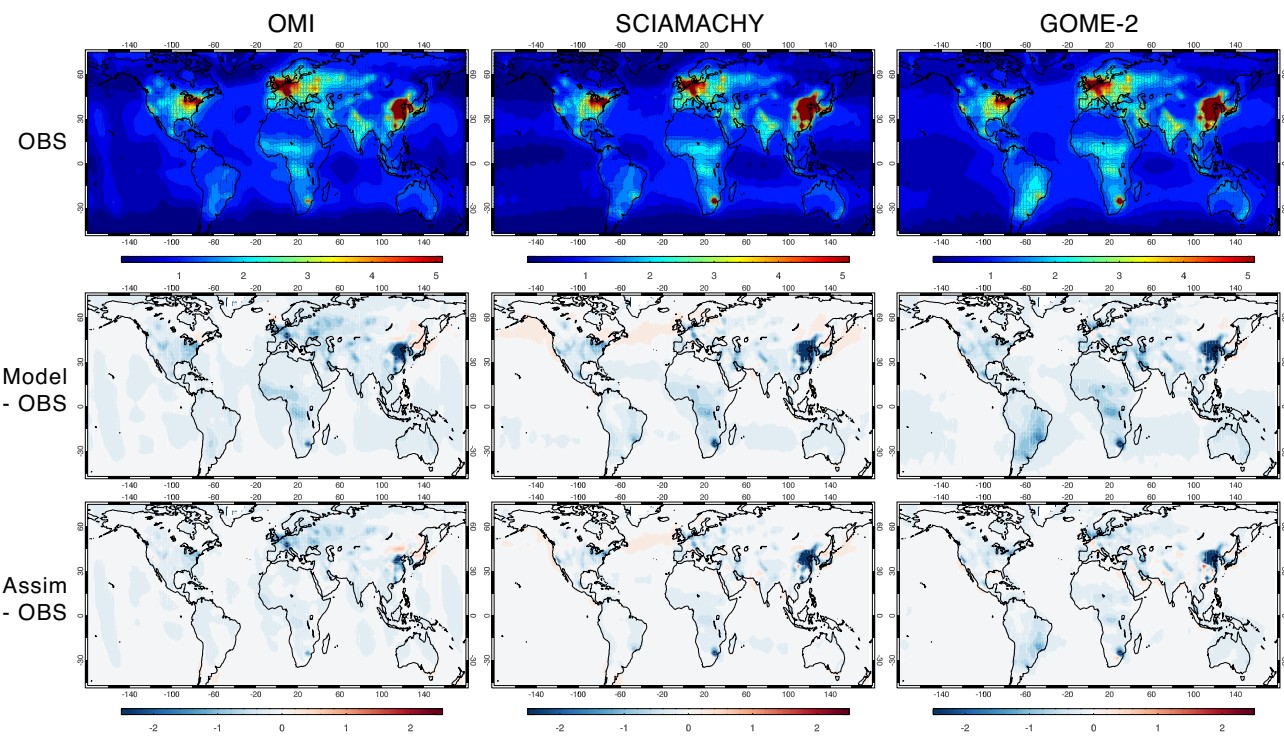

**Figure 2.** Global distributions of the tropospheric $NO_2$ columns (in $10^{15}\,\mathrm{molec\,cm^{-2}}$). The results are shown for OMI (left columns, sampling time $\approx$13:00 hrs) for 2005–2014, SCIAMACHY (middle columns, 10:00 hrs) for 2005–2011, and GOME-2 (right columns, 09:30 hrs) for 2007–2014. Upper rows show the tropospheric $NO_2$ columns obtained from the satellite retrievals (OBS); centre shows the difference between the model simulation and the satellite retrievals (Model-OBS); and lower rows show the difference between the data assimilation and the satellite retrievals (Assim-OBS).

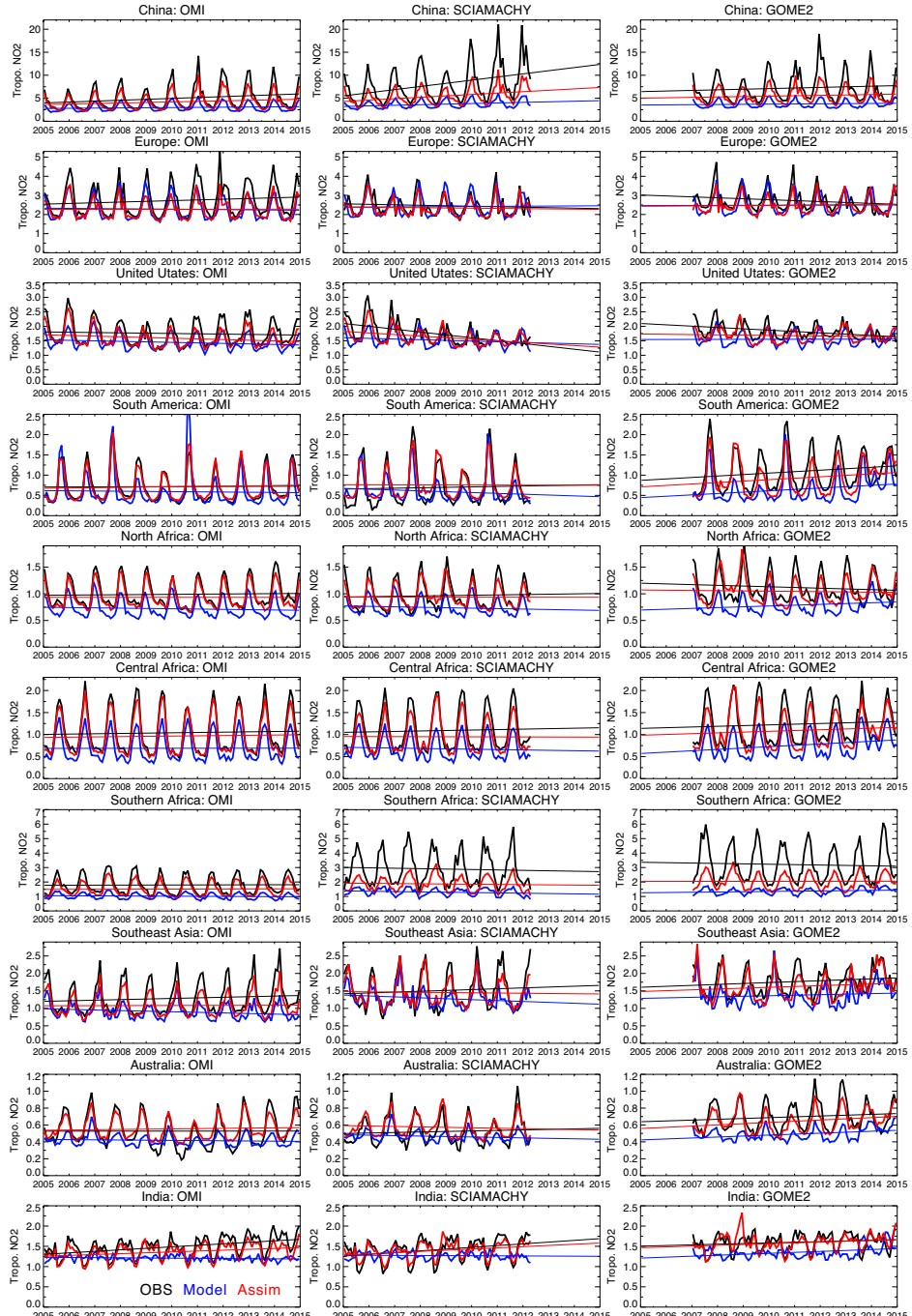

**Figure 3.** Time series of regional monthly mean tropospheric $NO_2$ columns (in $10^{15}\ \mathrm{molec\ cm^{-2}}$) averaged over China (110–123°E, 30–40°N), Europe (10°W–30°E, 35–60°N), the United States (70–125°W, 28–50°N), South America (50–70°W, 20°S–Equator), North Africa (20°W–40°E, Equator–20°N), Central Africa (10–40°E, Equator–20°S), Southern Africa (25–34°E, 22–31°S), Southeast Asia (96–105°E, 10–20°N), Australia (113–155°E, 11–44°S), and India (68–89°E, 8–33°N) obtained from the satellite retrievals (black), model simulation (blue), and the data assimilation (red). The model simulation and data assimilation results were obtained at the local overpass time of the retrievals with applying the averaging kernel.

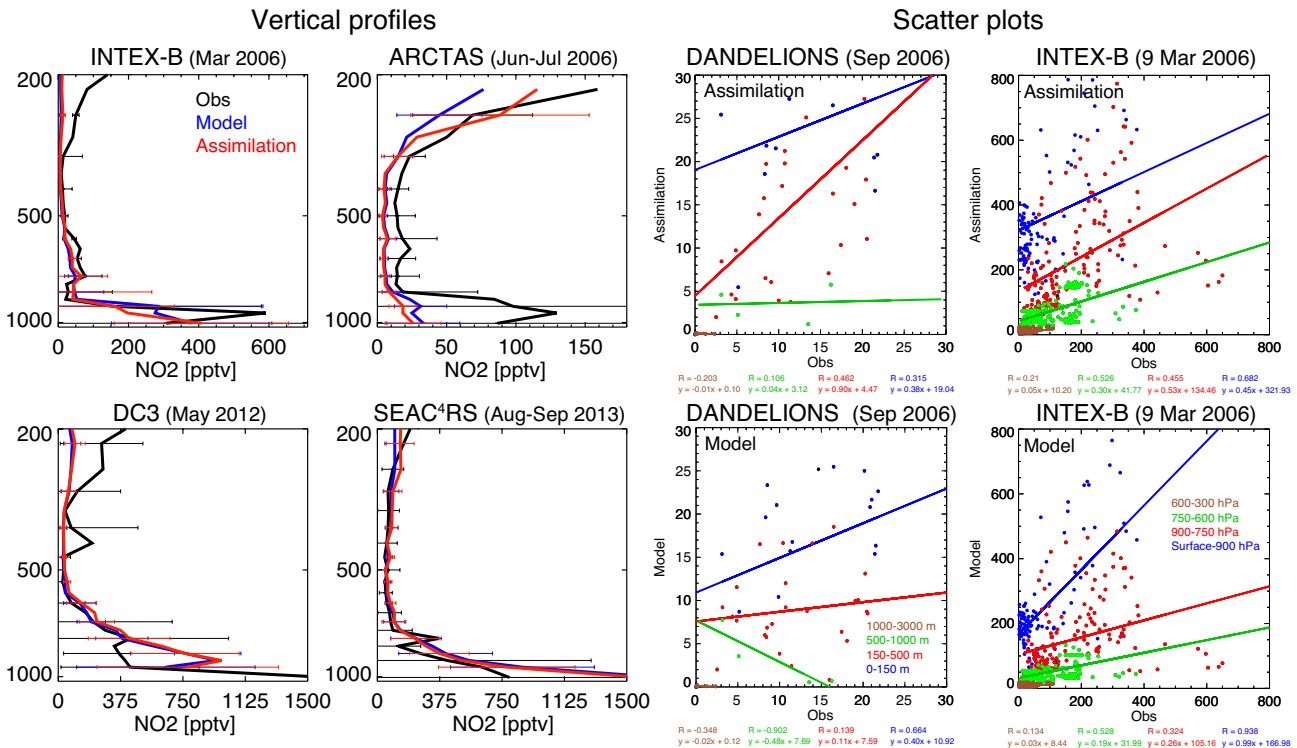

**Figure 4.** (Left panels) Mean vertical $NO_2$ profiles obtained during the ARCTAS campaign in June–July 2009; the ARCTAS campaign in June–July 2006; the DC3 campaign in May 2012; and the SEAC[4]RS campaign in August–September 2013. The black line represents the observation; the blue line represents the model simulation; and the red line represents the data assimilation. The error bars represent the standard deviation. (Right panels) Scatter plots of $NO_2$ concentrations for the data assimilation (top) and the model simulation (bottom) during the DANDELIONS campaign ($\mu g m^{-3}$) in September 2006 and during the INTEX-B campaign (in pptv) on March 9, 2006. The straight lines represent linear regression lines for each level. Each line represents a linear fit to the points of the same colour, and the colours represent the altitude level.

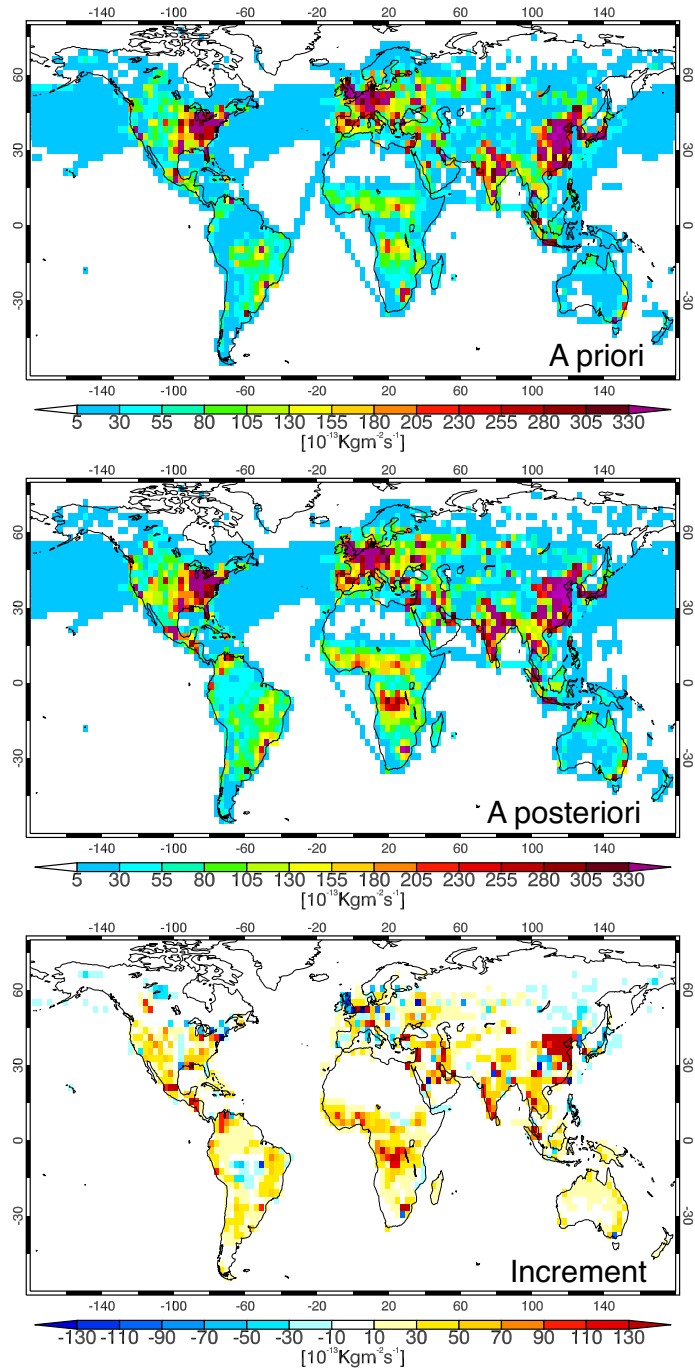

**Figure 5.** Global distributions of surface $NO_x$ emissions (in $10^{-13}$kgm$^{-2}$s$^{-1}$) averaged over 2005–2014. The a priori emissions (top), a posteriori emissions from the data assimilation run (middle), and analysis increment (bottom) are shown.

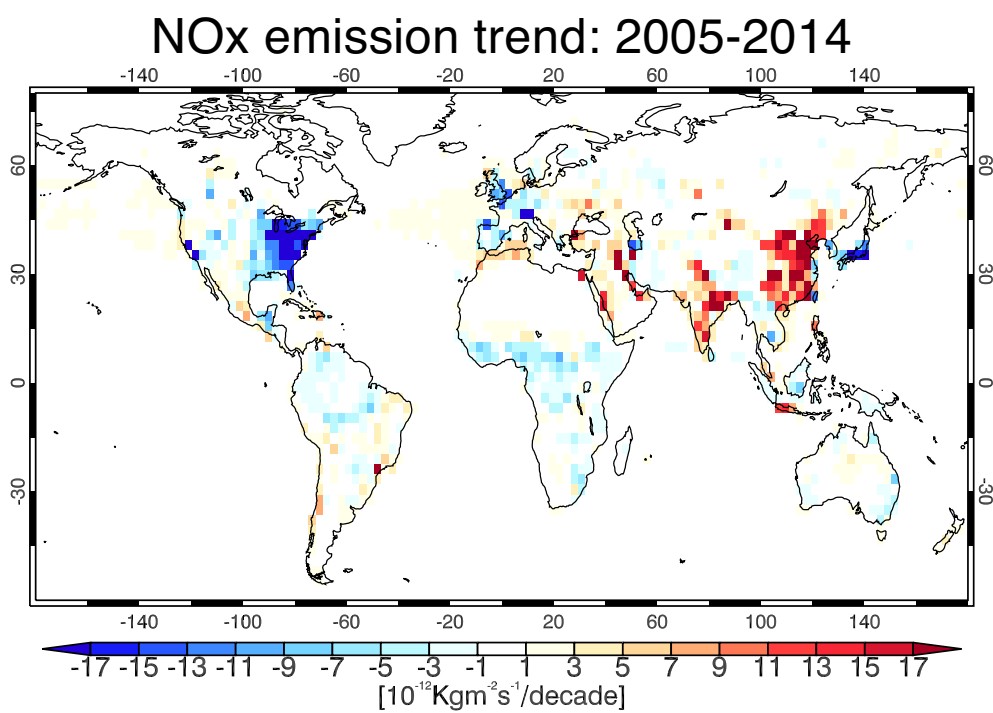

**Figure 6.** Global distribution of linear trend of the a posteriori surface $NO_x$ emissions (in $10^{-12}$kgm$^{-2}$s$^{-1}$ per decade) for the period 2005–2014. The red (blue) colour indicates positive (negative) trends.

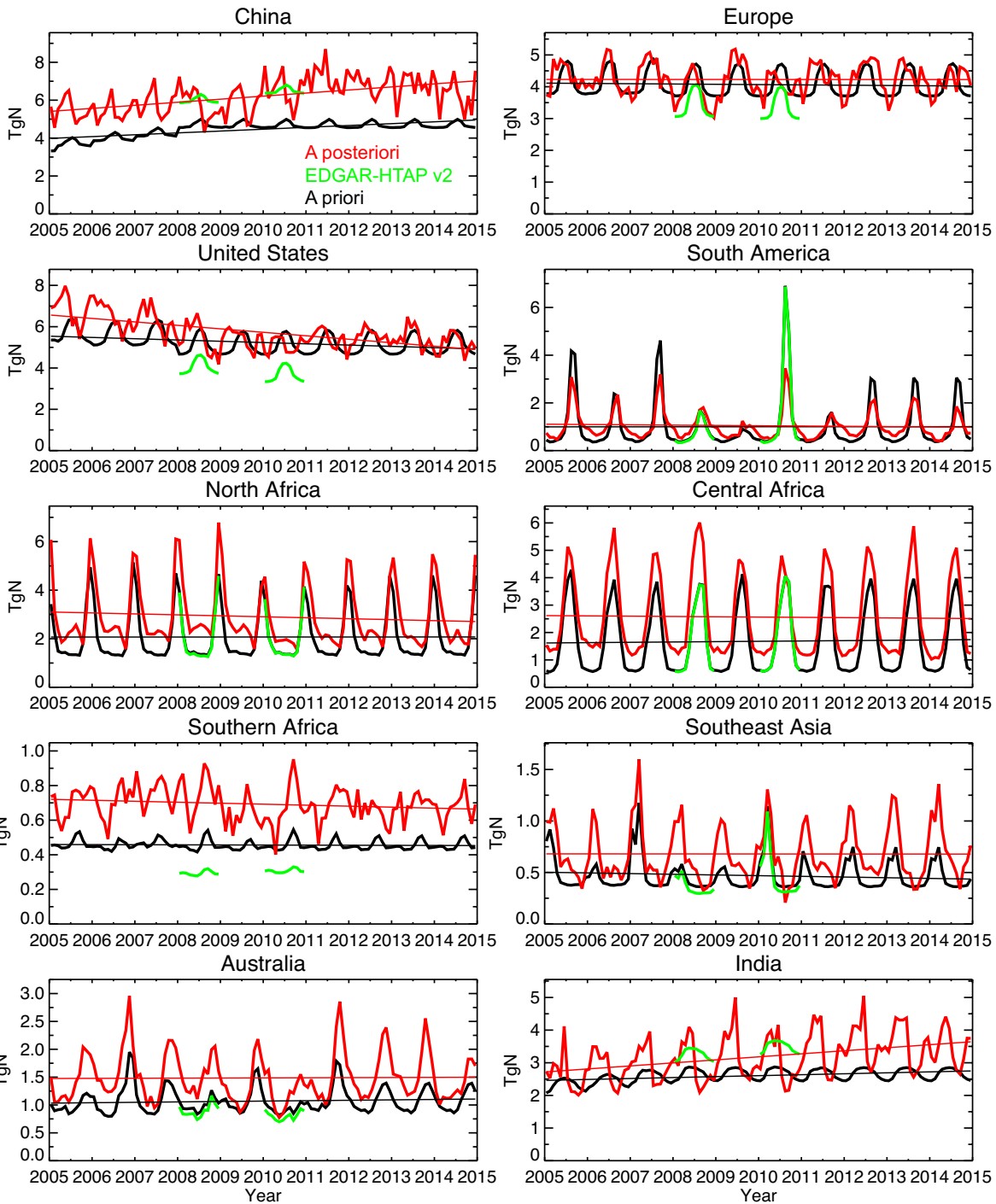

**Figure 7.** Time series of monthly total regional surface $NO_x$ emissions (in $Tg\,N\,yr^{-1}$) obtained from the a priori emissions (black lines) and the a posteriori emissions (red lines) for the period 2005–2014. The results are also shown for EDGAR-HTAP v2 emissions (green lines) for the years 2008 and 2010.

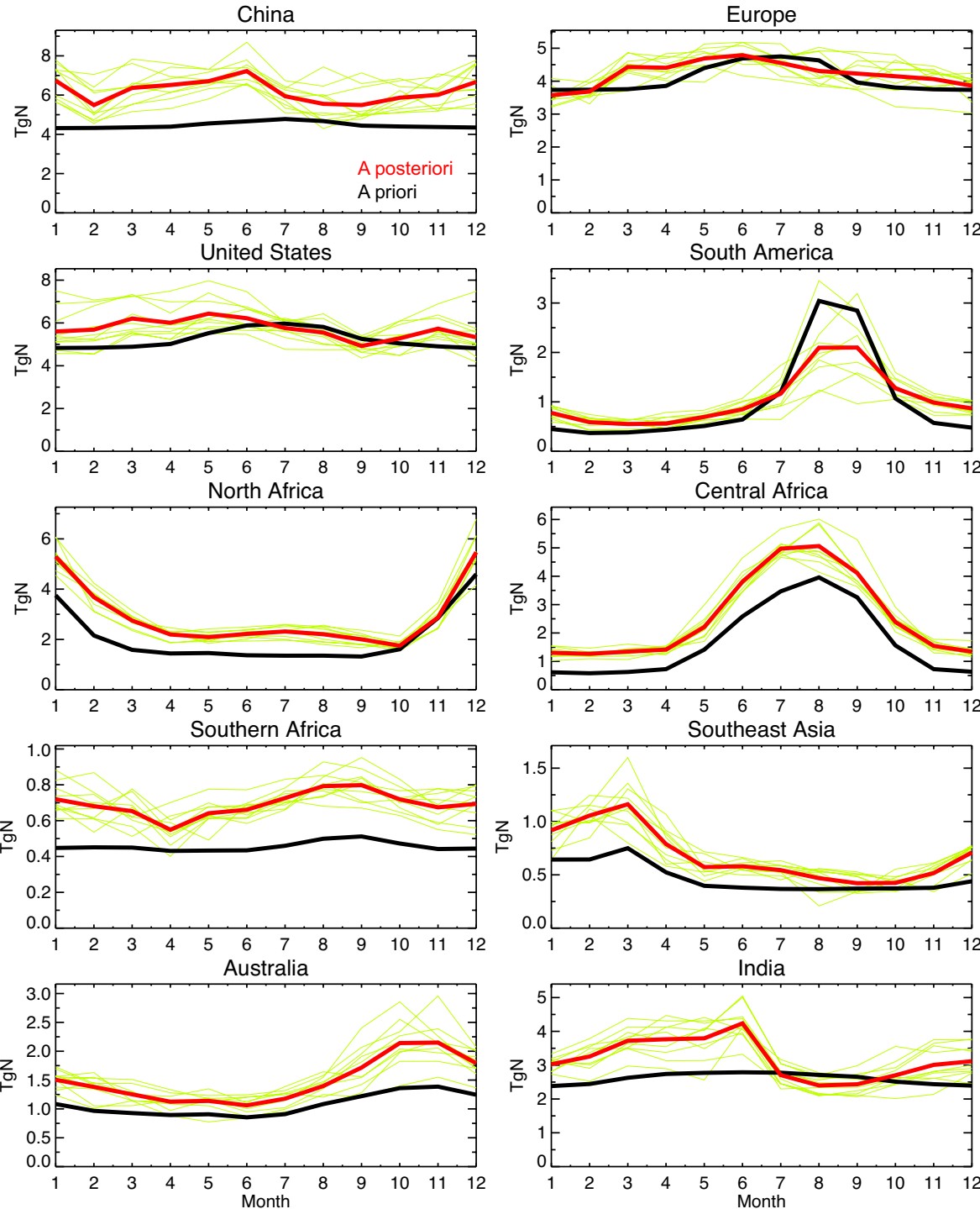

**Figure 8.** Seasonal variations of the regional surface $NO_x$ emissions (in $\mathrm{Tg\,N\,yr^{-1}}$) obtained from the a priori emissions (black line) and the a posteriori emissions (red line) averaged over the period 2005–2014. The results are also shown for the a posteriori emissions for individual years during 2005–2014 (yellow lines).

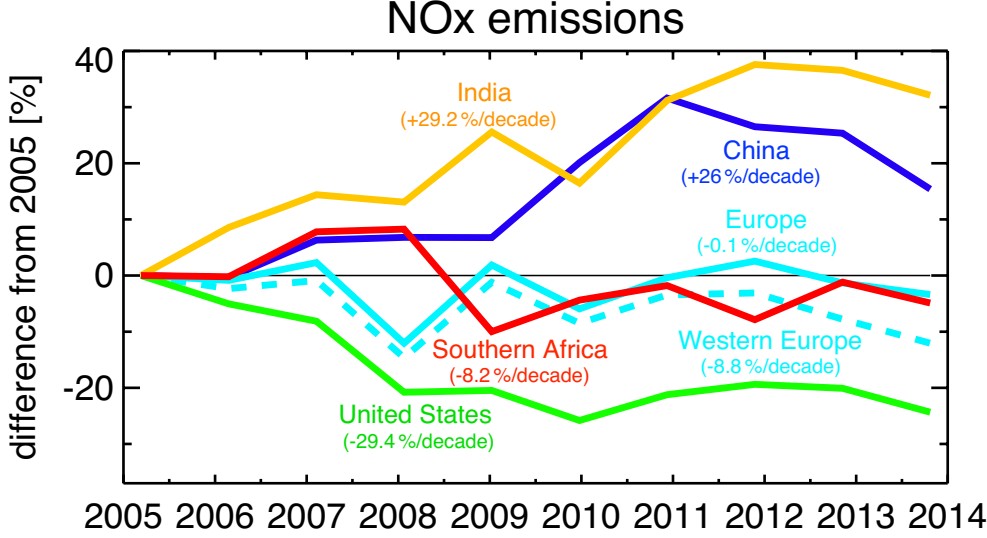

**Figure 9.** Time series of the difference (in %) of the annual mean a posteriori surface NO$_x$ emissions relative to the 2005 emissions in the period 2005–2014 for India (yellow), China (blue), Europe (light blue), western Europe (light blue dashed line), Southern Africa (red), and the United States (green).

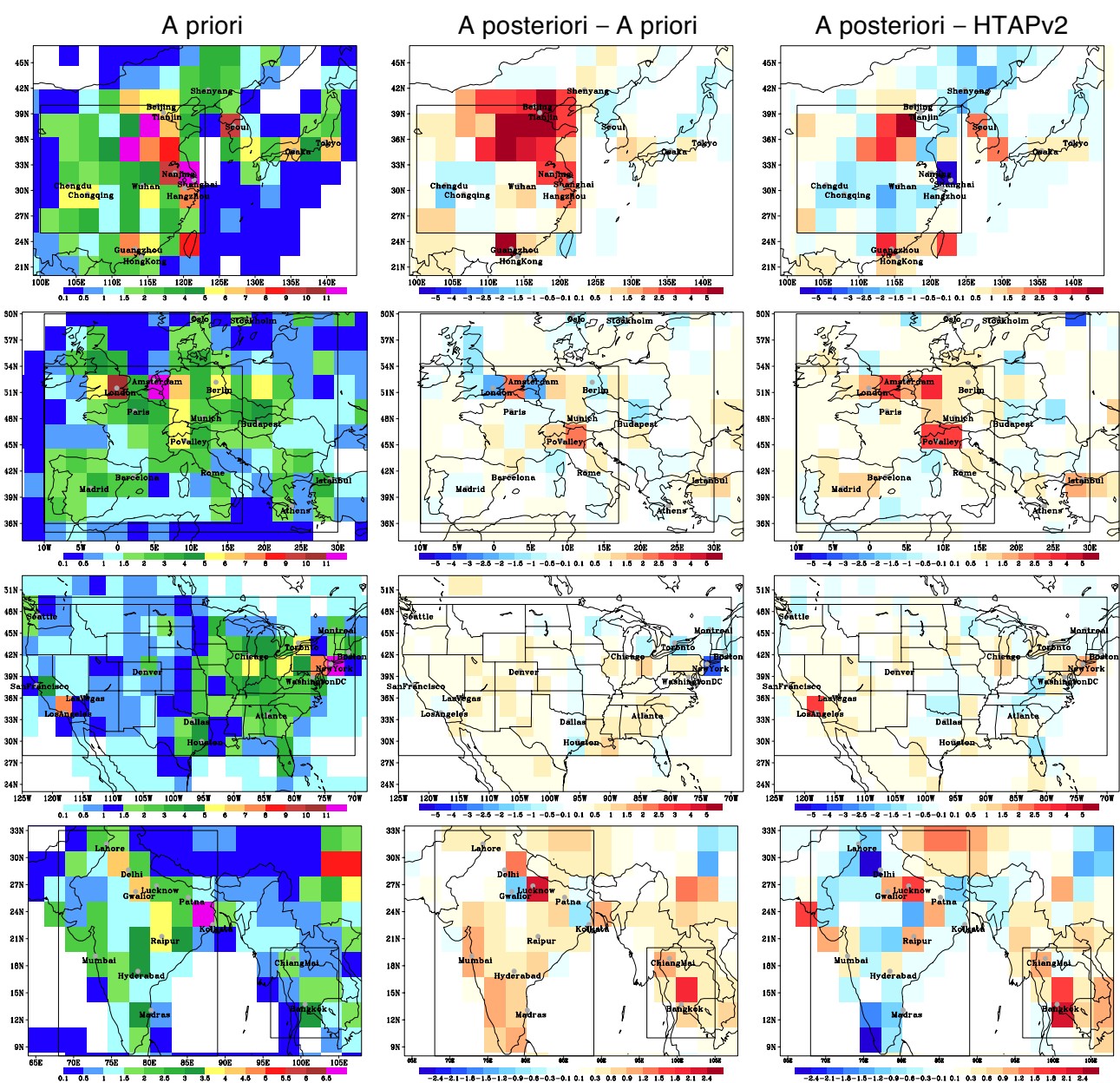

**Figure 10.** The regional distribution of ten-year mean surface $NO_x$ emissions (in $10^{-11} kg m^{-2} s^{-1}$) over East Asia (upper panels), Europe (upper middle panels), the United States (lower middle panels), and Southeast Asia (lower panels) obtained from the a posteriori emissions in the period 2005–2014 (left panels), and the difference between the a posteriori emissions and a priori emissions in the period 2005-2014 (centre panels), and between the a posteriori emissions and EDGAR-HTAP v2 emissions for the years 2008 and 2010 (right panels). The black square line represents the region used for the regional mean analysis.

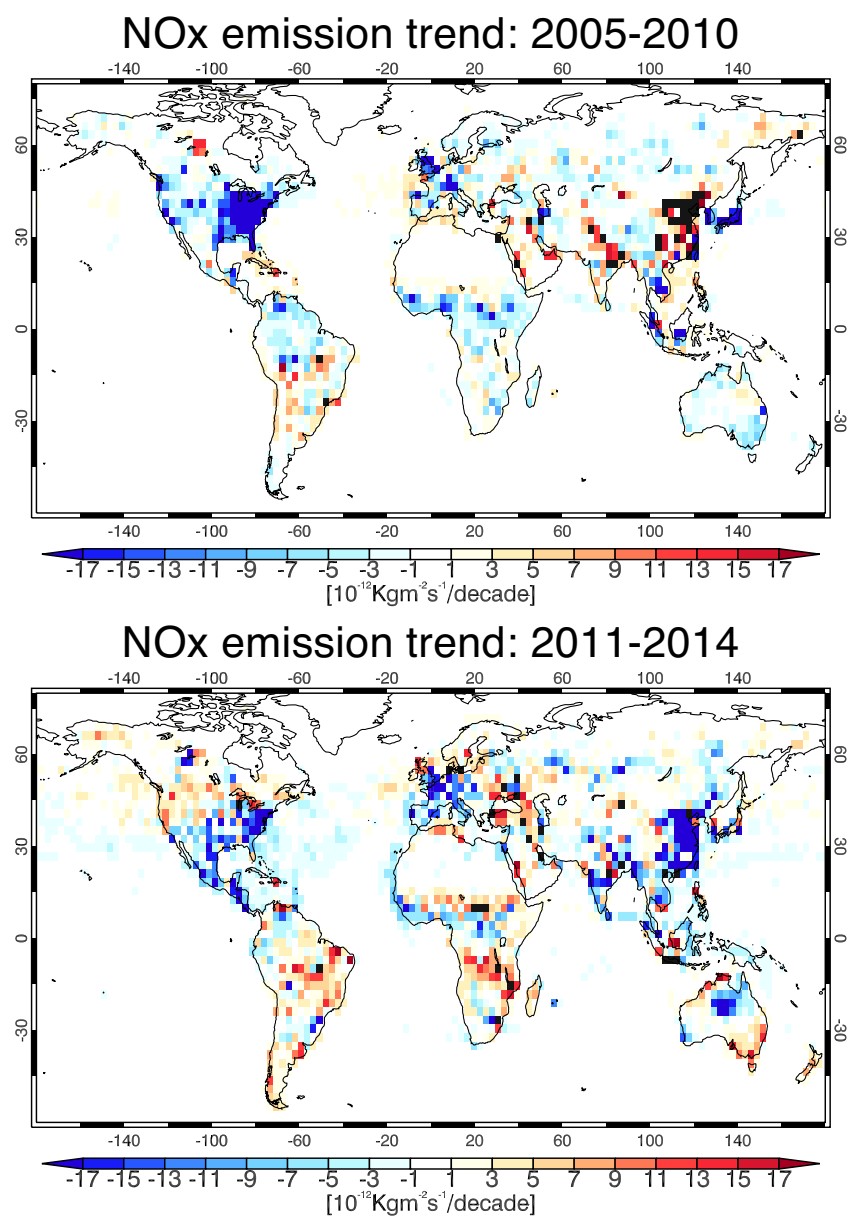

**Figure 11.** Global distribution of linear trend of the a posteriori surface $NO_x$ emissions for the period 2005–2010 (left) and 2011–2014 (right). The red (blue) colour indicates positive (negative) trends.

# NOx emission trend: 2005-2014

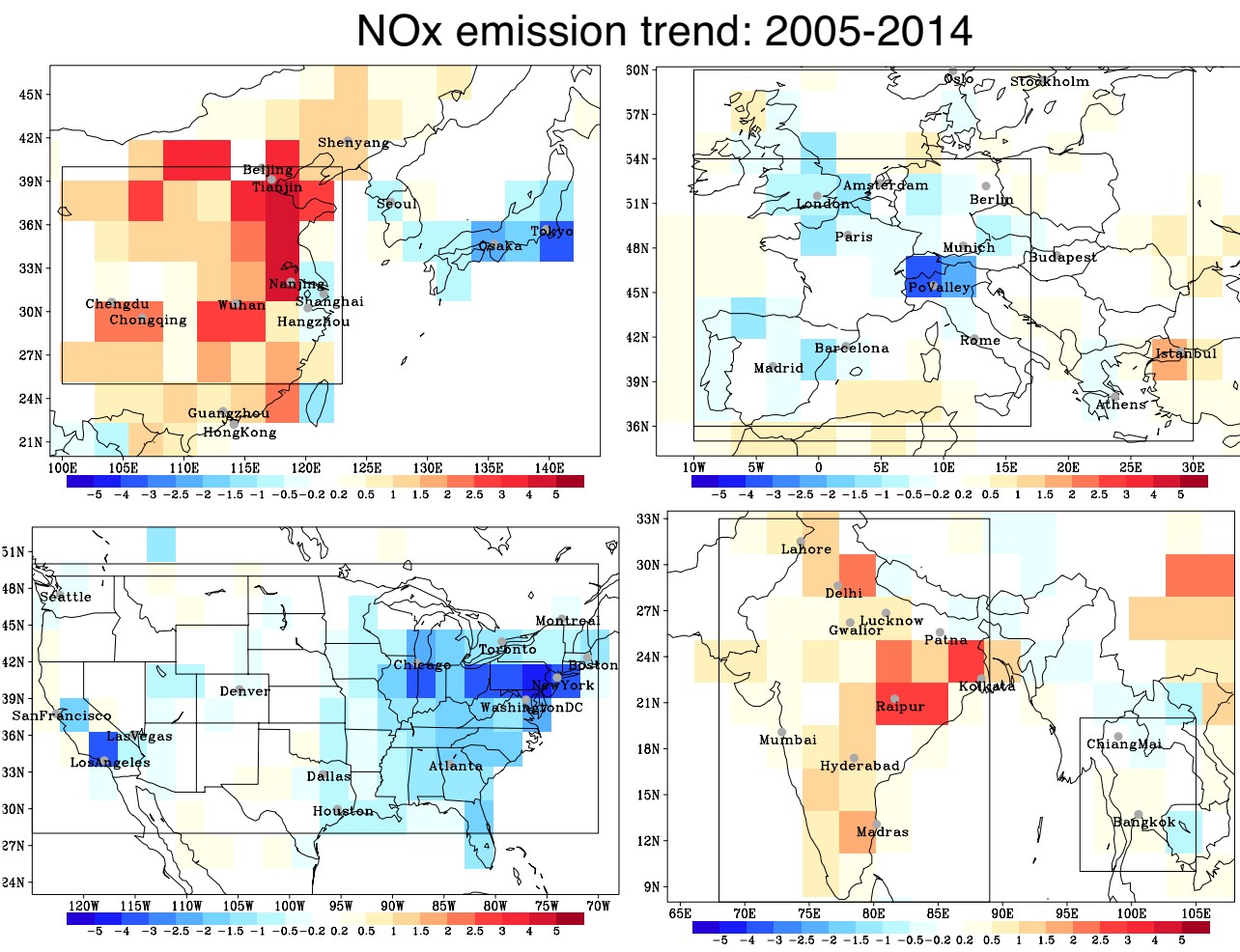

**Figure 12.** The regional distribution of the linear trend in surface $NO_x$ emissions (in $10^{-11} kgm^{-2}s^{-1}$ per decade) during 2005–2014 over East Asia (upper left), Europe (upper right), the United States (bottom left), and Southeast Asia (bottom right), obtained from the a posteriori emissions. The black square line represents the region used for the regional mean analysis.

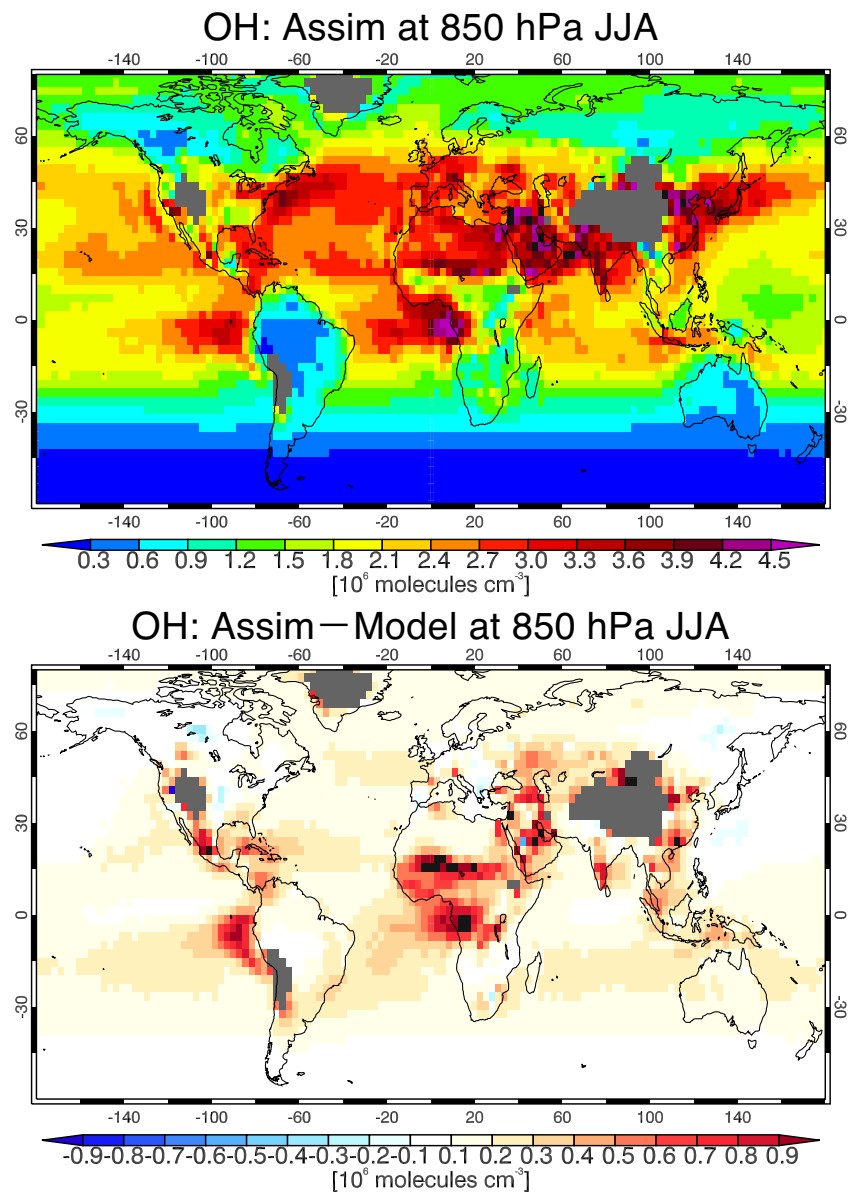

**Figure 13.** Global distribution of the ten-year mean OH concentration (in $10^6$ molecules cm$^{-3}$) in the data assimilation run (top) and its difference between the data assimilation run and the model simulation (bottom) averaged over June, July, and August over the 2005–2014 period at 850 hPa.

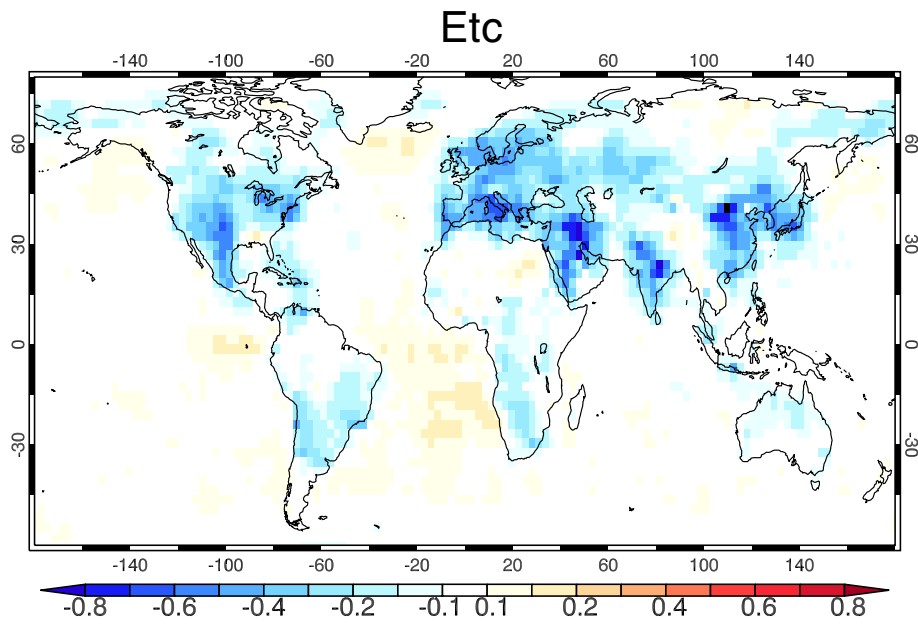

**Figure 14.** Global distribution of the annual mean correction factor for the emission diurnal variability ($Etc$) for the period 2005–2014.