# Peer review of "Decadal changes in global surface NOx emissions from multi-constituent satellite data assimilation"

_Atmospheric Chemistry and Physics, 2016_

## Referee Comment (RC1) · Anonymous Referee #1 · 30 Jun 2016

General comments

The paper analyses the changes in NOx polluting emissions, by assimilating different constituents from different satellite instruments in the chemistry-transport model. The manuscript is clear and it has nice logic flow. Because there are several works dealing with emission estimation using satellite-based observations, I would recommend stressing the added value of this approach (for example already in the abstract and in the trend calculation in Sect. 5.4), i.e. the assimilation of non-NO2 observations as compared to previous work where only NO2 is assimilated in the system. I recommend publication after addressing the following specific and technical comments.

Specific comments

[Figure]

P9 L17 The larger pixel size for GOME-2 and SCIAMACHY could indeed produce a dilution effect (lower NO2 level for larger pixel) compare to the smaller OMI pixel and thus, in principle, partially reduce the difference due to the different overpass time. Could you comment about that in the text?

Figure 2: It is quite difficult to distinguish the differences in these maps. It could be useful to show the differences compared to the observations in the second and third row, instead of the absolute tropospheric NO2 columns. It should help in highlighting the differences.

P11 L19 Are there any known/expected differences in the ways of reporting, that you could mention here between the a priori and the EDGAR-HTAP emission databases?

P11 L29-30 Again, is there an expected reason to explain the similarity between EDGAR_HTAP and a posteriori emissions, relative to the a priori?

P18 L18 You might want to refer here to this work about in plume chemistry effect: Vinken, G. C. M., Boersma, K. F., Jacob, D. J., and Meijer, E. W.: Accounting for non-linear chemistry of ship plumes in the GEOS-Chem global chemistry transport model, Atmos. Chem. Phys., 11, 11707-11722, doi:10.5194/acp-11-11707-2011, 2011.

P18 L32-33 It is unclear for me what do you mean for "overcorrect". Do you mean that NO2-only gives too high emission values? According Table 3, the NO2-only data assimilation almost always (except South America) gives smaller values than the full assimilation. Could you clarify?

Table 4 and section 5.4: Do these emission trends change when NO2-only assimilation is taken into account? I would include in Table 4 also the trends with NO2-only assimilation if the differences are sizeable.

How your results reported in Table 4 and Fig.8 compare with those reported as NO2 tropospheric columns (OMI Standard Product not DOMINO as in you study) by Krotkov et al. (2016) in their Fig. 8?

It could be interesting also to compare your results in China and US to the results by Liu et al. (2016) in Table S2 of their supplement. Those results are not based on data assimilation but are based on satellite data only. Liu, F., Beirle, S., Zhang, Q., Dörner, S., He, K., and Wagner, T.: NOx lifetimes and emissions of cities and power plants in polluted background estimated by satellite observations, Atmos. Chem. Phys., 16, 5283-5298, doi:10.5194/acp-16-5283-2016, 2016.

Technical corrections

P1 L6 biased -> biases

P2 L23 add reference Krotkov et al. 2016 here too

P2 L22 Kalam -> Kalman

P6 L32 GOME-II -> GOME-2

P7 L18-19 This needs reference

P7 L25 You might want to mention that those resolutions are valid in nadir direction only, but get bigger on the side of the swath and actually since 2008-2009 OMI row-anomaly doesn't allow complete daily global coverage.

Table 2 Australis -> Australia (and in the other tables too)

P14 L20 Los Angels -> Los Angeles

P20 L30 There are two dots at the end of the sentence

Table 4: Caption: OM ->OMI

Table 4 Is there a reason you put Table 4 before 5 and 6 but then you refer to Table 4 only in section 5.4, after mentioning 5 and 6? Please, clarify.
* * *

---

## Referee Comment (RC2) · Anonymous Referee #2 · 23 Aug 2016

This is a very interesting assimilation study on global NOx emissions. It is innovative in that it uses not only NO$_2$ data to constrain the emissions, but also data for other related chemical compounds (O$_3$, CO, HNO$_3$). Furthermore, it uses NO$_2$ data from 3 different nadir sensors (OMI, SCIAMACHY, GOME-2) which have different overpass times. In most inverse modeling and assimilation studies for NOx, the data from only one satellite sensor are used (and data from other sensors are possibly used for evaluation).

**General comments**

1. I certainly appreciate the effort made by the authors to incorporate more data.

[Figure]

There is logic to it: more data should be better than just one dataset. It is argued (maybe a bit too emphatically) that non-$NO_2$ datasets improve the NOx emission estimation because they should lead to better estimation of the NOx lifetime in the model. In general, that might be true, but I wouldn't be so sure that it is automatically the case. I find that adding more data from different species might contribute to obscure the interpretation of the results, because the additional data come with their own limitations and uncertainties (including biases) which are not all well characterized. I am not fully convinced that authors understand perfectly the role of the different datasets in the assimilation. I wonder in particular to what extent the NOx emission updates are driven by the non-$NO_2$ datasets. For example, ozone is apparently biased low in the model. Increasing NOx emissions is naturally found to improve ozone. But ozone could be biased low due to other reasons (transport, deposition, NMVOC chemistry and emissions). So, is ozone improved for the good reasons? Who knows? Many other CTMs overestimate surface ozone. I encourage the authors to moderate their claims regarding the advantages of additional data.

That being said, I concur that assimilating non-$NO_2$ dataset should contribute to improve (somewhat) the NOx lifetime in the model, which is a good thing. But I would expect the authors to provide a more quantitative and systematic analysis of how the non-$NO_2$ datasets influence the assimilation results. I also encourage the authors to be more cautious in their discussion, to reflect the possible limitations and complications associated with the use of additional, non-$NO_2$ measurements.

2. Regarding the use of 3 different $NO_2$ sensors, it is obvious (and I think the authors know) that the diurnal cycle alone cannot explain entirely the difference between NO2 columns from e.g. GOME-2 and OMI. And even if it would, it is also obvious that the diurnal cycle of NOx emissions is only one among many different processes affecting the diurnal cycle of NO2 columns. This article presents a smart

but crude procedure to improve the match with the 3 sensors simultaneously in spite of their inconsistencies: additional control parameters are introduced which allow modifying the diurnal cycle of emissions at every model pixel. Unfortunately, the result is not much credible as it would imply much stronger rush hour emission peaks even in regions where mobile emissions (cars) are not the main NOx source category. Power plans, industries, etc. do not have peak activity around 8 AM. The most negative values of the Etc parameter (Fig. 13) are found in Inner Mongolia, which has only few cars but does have power plants. Even though the diurnal cycle adjustment serves its purpose, it is clearly artificial. The authors should provide a better explanation of why they choose this procedure. Maybe it is the only one which works since we don't really understand the reasons for the inconsistency between morning and afternoon sensors. More discussion is warranted.

3. Although the paper is already quite long, I would expect at least some comparisons with independent $NO_2$ measurements. The reader has no clue regarding how the model performs for vertical profiles of $NO_2$. Also, given the focus of the paper on the diurnal cycle, comparisons with ground-based remote sensing data data could be useful. Therefore, although this article is clearly interesting and the methodology appears generally sound, I recommend that the authors try to address those main comments, as well as the other comments listed below, before it can be published in ACP.

**Other comments**

- Page 5, 1st full paragraph: I find odd to apply a unique diurnal cycle to all emissions in a given region, e.g. the anthropogenic-type cycle over Europe, eastern China, Japan, North America. This is strange. Why not make a weighted average based on the fractional a priori contribution of anthropogenic, biomass burning

and soil emissions? The diurnal cycle in New York doesn't have to be the same as in Wyoming.

- Page 5, 2nd paragraph: What is the vertical LNOx profile parameterization?

- Page 8, 1st full paragraph: Apparently the retrievals of Boersma et al. (2011, 2004) are used. But then the section goes on mentioning reduced errors based on Maasakkers (2013) even though the retrieval of Maasakkers is not used. This is difficult to understand.

- Page 10, lines 9-10: The assimilation effectively corrected the $NO_2$ columns at the different overpass times. The complete diurnal cycle of $NO_2$ concentration is an entirely different story. The model performance could be checked by comparisons with ground-based remote sensing data.

- Page 10, line 13: Here and elsewhere, the observation errors for highly polluted cases are considered very large. But the relative errors there are often of the order of 25-35% which is generally less than at more remote locations. Of course during the winter, things are different due to large zenith angles, clouds, and snow.

- Page 10, line 17: Some explanation on why the model lifetime of NOx would be too short would be useful.

- Page 12, on trends: The total number of observations changes during the 10-year period. Also the different satellites provided data during different periods. Some comments on possible consequences for trend estimation are needed.

- Page 12, lines 12-14: After assimilation, the emission is higher in January than in July over the U.S. Therefore, it is very unlikely that soil emissions can explain the changes in seasonality.

- Page 12, lines 15-17: The emission factors are indeed uncertain, but so are also the biomass burnt estimates.

- Page 12, lines 17-18: Note that the year-to-year variations over South America are very large.

- Page 12, lines 30-34: the increases aren't highest just over cities and lowest over remote areas. The entire N-E China and the Guangzhou area show large increases. The Chengdu/Chongqing area (with emission decreases) is certainly not "remote". Over N-E China, given the model resolution, it is not possible to distinguish urban from rural areas. Furthermore, Inner Mongolia shows large increases.

- Page 14, 1st full paragraph: The results regarding the trends in Europe are difficult to understand. Could you compare with previous studies for Europe, e.g. Curier et al. (Remote Sens. Environ. 2014, doi:10.1016/j.rse.2014.03.032)? From Figure 3, the OMI observations indicate a positive $NO_2$ trend, whereas GOME-2 shows the opposite trend. Such difference cannot be due the diurnal cycle of $NO_2$. Apparently those instruments have drifts which can be interpreted as emission trends. Please comment on this.

- Page 14, line 23 "The summertime peak enhancement is obvious over remote regions": Could you substantiate that claim?

- Page 15, line 10: Couldn't this be verified with e.g. MODIS fire counts or other biomass burning proxies?

- Page 15, line 27: Although temperature has some effect, the shorter NOx lifetimes at tropical latitudes such as India are primarily due to higher photolysis rates and specific humidity.

- Page 16, line 3: Why would high resolution analysis be required? This shouldn't be so complicated. For example, biomass burning has a distinct seasonality which can be probed at coarse resolution.

- Page 16, end of section 4.2.6. The high temporal correlation between N. Africa and Central Africa is interesting. Would this be related to biomass burning or to soil emissions (or both)? Examination of MODIS fire counts could help, also possibly temperature data. Is this correlation also found in the $NO_2$-only assimilation?

- Page 19, first paragraph: Basically, the improved ozone is due to the general increase in NOx emissions over all regions, whereas the a priori model seems to have a negative bias in surface ozone. In the study of Travis et al. (ACPD, 2016, doi:10.5194/acp-2016-110), NOx emissions over the U.S. are found to be largely overestimated in comparisons with aircraft data.

- Page 19, last full paragraph: Could you also provide the global tropospheric chemical lifetime of methane (or methyl chloroform) in the model?

- Page 20, first sentence: "The inverse lifetime is expected to be proportional to the ratio of NOx to $NO_2$". It's the other way around. Increase the NOx to $NO_2$ ratio should increase the fraction of NO (which does not react with OH) and therefore decrease the sink of NOx, i.e. the inverse lifetime. The main effect of a NOx emission increase is (most often) increased OH levels and therefore shorter NOx lifetime. The point which is made in this paragraph is unclear.

- Page 20, lines 20-25: The large adjustments are first said to suggest a change in diurnal evolution of the emissions. Then they are said to suggest other possible causes related to the model or the retrievals. Correct, but then the first suggestion is not necessary. Values of Etc as negative as -0.6 or -0.8 are found at some locations, which are impossibly large. Large Etc should be found only in areas

where traffic is the dominant source. This does not appear to be the case (Fig. 13).

- Page 22, line 4: Is the given observed $NO_2$ concentration trend for OMI or for all sensors? The trend appears very different between GOME-2 and OMI.

- Page 22, line 6: Why would $NO_2$ have become more long-lived? Does OH show a negative trend in this region? If so, what are the causes for this trend? Note that the fraction of $NO_2$ to NOx is determined mostly by ozone and the photolysis rate of $NO_2$. A shift in $NO_2$:NOx emission ratio does not matter much except directly over emission areas (titration effect). The paragraph seems to imply that the $NO_2$:NOx emission ratio in the model has changed over the 10-year period. Is this true?

**Technical corrections**

- Page 1: "Forkert" should be "Folkert"

- p. 1 l. 6: "biased" should be "biases"

- p.1 l. 8: "the development" : do you mean the evolution?

- p. 2 l. 2: "traffic rush hours, economic activity..." those are not "source categories". Sentence is confusing, please rephrase.

- p. 2 l. 25: Kalam should be Kalman

- p. 2 l. 33: insert a hyphen between "multi" and "constituent" same line: replace advancement by advance or progress

- p. 3, l. 21-22: The sentence "The OH magnitude and gradient is the primary chemical pathway for propagating observational information..." does not make much sense. Rephrase or delete.

- p. 3 l. 31: Replace maybe "an EnKF technique" by "a variant of the EnKF technique"

- p. 4 l. 25: Explain "background spread"

- p. 4 l. 30: Isn't Yienger and Levy (1995) the correct reference for GEIA NOx? Please check.

- p. 6 lines 12, 16, 22: I think "Ets" should be "Etc"

- p. 6 l. 31: GOME-2 (not GOME-II)

- p. 10 l. 24: Here and at other instances, replace "c.f." by more standard phrasing (e.g. "see")

- p. 11 l. 12: the sentence seems to imply that the chemical lifetime of NOx might be underestimated, which is not what you mean here. Rephrase.

- p. 11 l. 20: I suppose you mean GFED 3 here, not EDGAR 4.2 (see section 2.1)

- p. 12 l. 9: "southern parts of the Eurasian continent" : don't you simply mean China? The seasonal variation over Southeast Asia does not show a summer maximum, so it does not fit into the point made in this sentence.

- p. 12 l. 18 "assumptions applied for the a priori emissions" I think you could be more specific (use of climatology after 2011)

- p. 12 l. 27: "The EDGAR v4 emissions are too low": that statement is too blunt for several reasons. Replace "EDGAR v4" by "our a priori inventory" (since EDGAR

for 2008 is used after 2008, and since soil emissions are not from EDGAR). Furthermore, add something like "Our assimilation indicates that...".

- p. 12 l. 28: "too low by a factor of 0.6": awkward. Should be too low by a factor of 1/0.6 (i.e. about 1.7)

- p.12 l. 29: emissions are maximum in June, not July.

- p. 13 l. 34: "in the reported mobile emissions": why specifically in this source category?

- p. 14 l. 1: replace "reveal" by "show"

- p. 14 l. 2: replace "by" by "after"

- p. 14, l. 18: "around Atlanta (...) and Denver": this seems to indicate that increments are found mostly over cities, which is not true. Consider replacing by "Southeast US and most of Western US"

- p. 14 l. 19: delete "over" after "around"

- p. 14 l. 20: Los Angeles

- p. 15 l. 9: I think the word "boreal" is superfluous here (and at many other instances in the text)

- p. 15 l. 20-21: "particularly strong increase around Delhi" but the changes over Delhi are lower than the regional average!

- p. 16 l. 5: Replace "by data assimilation" by "due to data assimilation"

- p. 16 l. 33: replace "reflection" by "reflecting" same line: replace "when" by "whereas"

- p. 19 l. 17: LNOx (instead of LNO)

- p. 19 l. 34: here and elsewhere in the manuscript, insert hyphen between "multiple" and "species"

- p. 20 l. 17: Replace "rom the three. . ." by "constrained by the three. . ."

- p. 21 l. 8: "using either model after data assimilation" : awkward, the model is used for data assimilation

- p. 32: Table 2 and elsewhere: Replace "Australis" by "Australia"

- p. 41 Figure 6: it is impossible to distinguish black and dark blue on the "South America" plot. Consider using other colors.

- p. 44, Figure 9: the title of the middle panels should be "A posteriori – A priori". Same for the title of the right panels, the minus sign is missing.

---

## Author Response (AR1)

**Author's comments in reply to the anonymous referee for "Decadal changes in global surface NOx emissions from multi-constituent satellite data assimilation " by K. Miyazaki et al.**

We want to thank the referee for the helpful comments. We have revised the manuscript according to the comments, and hope that the revised version is now suitable for publication. Below are the referee comments in italics with our replies in normal font.

***Reply to Referee #1***

*The paper analyses the changes in NOx polluting emissions, by assimilating different constituents from different satellite instruments in the chemistry-transport model. The manuscript is clear and it has nice logic flow. Because there are several works dealing with emission estimation using satellite-based observations, I would recommend stressing the added value of this approach (for example already in the abstract and in the trend calculation in Sect. 5.4), i.e. the assimilation of non-NO2 observations as compared to previous work where only NO2 is assimilated in the system. I recommend publication after addressing the following specific and technical comments.*

The impact of non-NO2 measurements is discussed more carefully in the revised manuscript. Table 5 has been added to discuss the impact of non-NO2 measurements on the a posteriori emissions. Linear trend estimations from the NO2-only assimilation have been included in Table 4. The following discussions have been added to Section 5.1:

"Table 5 compares the estimated emissions between the multiple-species data assimilation and a NO2-only data assimilation. The estimated emissions differ in many regions if non-NO2 data assimilation is considered because the ratio of predicted NOx emission and NO2 column has been adjusted by non-NO2 observations. The assimilation of non-NO2 measurements leads to changes of up to about 70 % in the regional monthly-mean emissions. The estimated ten-year total regional emissions for South America and Australia are about 10 % lower in the multiple-species assimilation than in the NO2-only assimilation. The RMSE between the two estimates for the monthly total regional emissions is 15.5 % for central Africa, 16.5 % for Australia, and about 5-8 % for major polluted regions during the ten-year period. The estimated monthly mean emissions are mostly smaller in the multiple-species assimilation than in the NO2-only assimilation, especially over the tropical and southern subtropical regions such as South America, central Africa, and Australia, suggesting that NO2-only data assimilation tends to overcorrect the emissions from the a priori. The monthly total global emissions decrease by up to 6 TgN (in boreal summer) if non-NO2 data assimilation is considered. The ten-year linear trend is also different over most industrial areas (Table 4). For instance, the positive trend for India is 34.3 %/decade

in the NO2-only assimilation, which is larger than the 29.2 %/decade in the multiple-species assimilation. For the United States, the negative trend is larger in the multiple-species assimilation (-29.4 %/decade) than in the NO2-only assimilation (-23.9 %/decade). These results confirm that the assimilation of measurements for species other than NO2 provides additional constraints on the NOx emissions over both anthropogenic and biomass burning regions."

*P9 L17 The larger pixel size for GOME-2 and SCIAMACHY could indeed produce a dilution effect (lower NO2 level for larger pixel) compare to the smaller OMI pixel and thus, in principle, partially reduce the difference due to the different overpass time. Could you comment about that in the text?*

The concentration of individual observations over polluted regions can vary between sensors, corresponding to the pixel size. However, because we employed the super observation approach and averaged multiple observations within a large super observation grid (i.e., about 2.8 degrees) before data assimilation, the influence of different pixel size should have a small impact on the data assimilation result, assuming the super observation grid is well covered by observation pixels. To clarify this point, the sentence has been rewritten as:
"Therefore, the differences in overpass time and also in pixel size could be the principle cause of the differences between the three different satellite retrievals, although the use of super observations for all the sensors reduces the influence of different pixel sizes."

*Figure 2: It is quite difficult to distinguish the differences in these maps. It could be useful to show the differences compared to the observations in the second and third row, instead of the absolute tropospheric NO2 columns. It should help in highlighting the differences.*

Figure 2 has been revised to show the differences.

*P11 L19 Are there any known/expected differences in the ways of reporting, that you could mention here between the a priori and the EDGAR-HTAP emission databases?*
*P11 L29-30 Again, is there an expected reason to explain the similarity between EDGAR_HTAP and a posteriori emissions, relative to the a priori?*

To our best knowledge, no comprehensive comparison has been made between these inventories.

*P18 L18 You might want to refer here to this work about in plume chemistry effect: Vinken, G. C. M., Boersma, K. F., Jacob, D. J., and Meijer, E. W.: Accounting for non- linear chemistry of ship plumes in*

*the GEOS-Chem global chemistry transport model, Atmos. Chem. Phys., 11, 11707-11722, doi:10.5194/acp-11-11707-2011, 2011.*

Added.

*P18 L32-33 It is unclear for me what do you mean for "overcorrect". Do you mean that NO2-only gives too high emission values? According Table 3, the NO2-only data assimilation almost always (except South America) gives smaller values than the full assimilation. Could you clarify?*

We have extended the sensitivity calculation using NO2-measurements only. The results confirm that the estimated regional emissions are mostly higher in the NO2-only assimilation than in the multiple-species assimilation. To clarify this, the sentence has been rewritten as follows:

"The estimated monthly mean emissions are mostly smaller in the multiple-species assimilation than in the NO2-only assimilation, especially over the tropical and southern subtropical regions such as South America, central Africa, and Australia, suggesting that NO2-only data assimilation tends to overcorrect the emissions."

*Table 4 and section 5.4: Do these emission trends change when NO2-only assimilation is taken into account? I would include in Table 4 also the trends with NO2-only assimilation if the differences are sizeable.*

The linear trends from the NO2-only assimilation have been included in Table 4 and discussed in the revised manuscript.

*How your results reported in Table 4 and Fig.8 compare with those reported as NO2 tropospheric columns (OMI Standard Product not DOMINO as in you study) by Krotkov et al. (2016) in their Fig. 8?*

The following sentence has been added:
"These year-to-year variations in the observed NO2 concentrations have previously been reported by Duncan et al. (2016) and Krotkov et al. (2016)."

*It could be interesting also to compare your results in China and US to the results by Liu et al. (2016) in Table S2 of their supplement. Those results are not based on data assimilation but are based on satellite data only. Liu, F., Beirle, S., Zhang, Q., Dörner, S., He, K., and Wagner, T.: NOx lifetimes and emissions of cities and power plants in polluted background estimated by satellite observations, Atmos. Chem.*

*Phys., 16, 5283-5298, doi:10.5194/acp-16-5283-2016, 2016.*

Thank you for the information. However, Liu et al (2016) estimated emissions sources at a 40 km × 40 km scale for point source areas (power plants and cities), which is about seven time higher resolution than that of our estimates. As the estimated emissions may be sensitive to the resolution, direct comparison with their results is difficult.

*Technical corrections*
*P1 L6 biased -> biases*

Corrected.

*P2 L23 add reference Krotkov et al. 2016 here too*

Added.

*P2 L22 Kalam -> Kalman*
*P6 L32 GOME-II -> GOME-2*

Corrected.

*P7 L18-19 This needs reference*

Added.

*P7 L25 You might want to mention that those resolutions are valid in nadir direction only, but get bigger on the side of the swath and actually since 2008-2009 OMI row-anomaly doesn't allow complete daily global coverage.*

The following sentence has been added:
"Since December 2009, approximately half of the pixels have been compromised by the so-called row anomaly, which reduced the daily coverage of the instrument."

*Table 2 Australis -> Australia (and in the other tables too)*
*P14 L20 Los Angels -> Los Angeles*

*P20 L30 There are two dots at the end of the sentence*

Corrected.

*Table 4: Caption: OM ->OMI*

Corrected.

*Table 4 Is there a reason you put Table 4 before 5 and 6 but then you refer to Table 4 only in section 5.4, after mentioning 5 and 6? Please, clarify.*

Table 4 is referred before Table 5 and 6 in the revised paper,

**Author's comments in reply to the anonymous referee for "Decadal changes in global surface NOx emissions from multi-constituent satellite data assimilation " by K. Miyazaki et al.**

We want to thank the referee for the helpful comments. We have revised the manuscript according to the comments, and hope that the revised version is now suitable for publication. Below are the referee comments in italics with our replies in normal font.

*Reply to Referee #2*

*1. I certainly appreciate the effort made by the authors to incorporate more data. There is logic to it: more data should be better than just one dataset. It is argued (maybe a bit too emphatically) that non-NO2 datasets improve the NOx emission estimation because they should lead to better estimation of the NOx lifetime in the model. In general, that might be true, but I wouldn't be so sure that it is automatically the case. I find that adding more data from different species might contribute to obscure the interpretation of the results, because the additional data come with their own limitations and uncertainties (including biases) which are not all well characterized. I am not fully convinced that authors understand perfectly the role of the different datasets in the assimilation. I wonder in particular to what extent the NOx emission updates are driven by the non-NO2 datasets. For example, ozone is apparently biased low in the model. Increasing NOx emissions is naturally found to improve ozone. But ozone could be biased low due to other reasons (transport, deposition, NMVOC chemistry and emissions). So, is ozone improved for the good reasons? Who knows? Many other CTMs overestimate surface ozone. I encourage the authors to moderate their claims regarding the advantages of additional data.*
*That being said, I concur that assimilating non-NO2 dataset should contribute to improve (somewhat) the NOx lifetime in the model, which is a good thing. But I would expect the authors to provide a more quantitative and systematic analysis of how the non-NO2 datasets influence the assimilation results. I also encourage the authors to be more cautious in their discussion, to reflect the possible limitations and complications associated with the use of additional, non-NO2 measurements.*

We appreciate the author's constructive comments. First, the possible limitations associated with the use of non-NO2 measurements are more clearly discussed in the revised manuscript. The following sentences have been added to Section 5.3:

"Although the assimilation of multiple-species data influences the representation of the entire chemical system (Miyazaki et al., 2012b, 2015), the influence of model and observation errors remains a concern. In the multiple-species data assimilation, model performance is critical for the correct propagation of observational information between chemical species and to improve the emission estimation, whereas

biases in any of the measurement data sets (including non-NO2 measurements) may seriously degrade the emission estimation (Miyazaki et al., 2013). Improvements in the model, data assimilation scheme, and retrieved observations are essential to reduce the uncertainty on the emission estimates from the multiple-species data assimilation."

Secondly, the impact of non-NO2 measurements is more clearly discussed in the revised manuscript. Table 5 has been added to demonstrate the impact of non-NO2 measurements on the a posteriori emissions. Linear trend estimations from the NO2-only assimilation have been added to Table 4. The relevant discussions in Section 5.1 have been rewritten as follows:

"Table 5 compares the estimated emissions between the multiple-species data assimilation and a NO2-only data assimilation. The estimated emissions differ in many regions if non-NO2 data assimilation is considered because the ratio of predicted NOx emission and NO2 column has been adjusted by non-NO2 observations. The assimilation of non-NO2 measurements leads to changes of up to about 70 % in the regional monthly-mean emissions. The estimated ten-year total regional emissions for South America and Australia are about 10 % lower in the multiple-species assimilation than in the NO2-only assimilation. The RMSE between the two estimates for the monthly total regional emissions is 15.5 % for central Africa, 16.5 % for Australia, and about 5-8 % for major polluted regions during the ten-year period. The estimated monthly mean emissions are mostly smaller in the multiple-species assimilation than in the NO2-only assimilation, especially over the tropical and southern subtropical regions such as South America, central Africa, and Australia, suggesting that NO2-only data assimilation tends to overcorrect the emissions from the a priori. The monthly total global emissions decrease by up to 6 TgN (in boreal summer) if non-NO2 data assimilation is considered. The ten-year linear trend is also different over most industrial areas (Table 4). For instance, the positive trend for India is 34.3 %/decade in the NO2-only assimilation, which is larger than the 29.2 %/decade in the multiple-species assimilation. For the United States, the negative trend is larger in the multiple-species assimilation (-29.4 %/decade) than in the NO2-only assimilation (-23.9 %/decade). These results confirm that the assimilation of measurements for species other than NO2 provides additional constraints on the NOx emissions over both anthropogenic and biomass burning regions."

Thirdly, as suggested by the reviewer, the model ozone low bias could be introduced by errors other than those for NOx emissions. To discuss this, the following sentences have been added to Section 5.1:
"Note that the emissions of O3 precursors other than NOx, such as VOCs, and various model processes in atmospheric transport and chemistry influences the model performance. The optimization of additional precursors emissions and the improvement of the forecast model could be important for improving O3 simulations, as discussed in our previous studies (Miyazaki et al., 2012b; 2015)."

*2. Regarding the use of 3 different NO2 sensors, it is obvious (and I think the authors know) that the diurnal cycle alone cannot explain entirely the difference between NO2 columns from e.g. GOME-2 and OMI. And even if it would, it is also obvious that the diurnal cycle of NOx emissions is only one among many different processes affecting the diurnal cycle of NO2 columns. This article presents a smart but crude procedure to improve the match with the 3 sensors simultaneously in spite of their inconsistencies: additional control parameters are introduced which allow modifying the diurnal cycle of emissions at every model pixel. Unfortunately, the result is not much credible as it would imply much stronger rush hour emission peaks even in regions where mobile emissions (cars) are not the main NOx source category. Power plans, industries, etc. do not have peak activity around 8 AM. The most negative values of the Etc parameter (Fig. 13) are found in Inner Mongolia, which has only few cars but does have power plants. Even though the diurnal cycle adjustment serves its purpose, it is clearly artificial. The authors should provide a better explanation of why they choose this procedure. Maybe it is the only one which works since we don't really understand the reasons for the inconsistency between morning and afternoon sensors. More discussion is warranted.*

I agree that the diurnal emission variation does not solely explain the simulated tropospheric NO2 column differences between the OMI and GOME-2 (or SCIAMACHY) overpass time. Because of model errors and differences between the satellite instruments, the obtained diurnal cycle adjustment can be artificial for some cases, as mentioned by the reviewer. The following sentences have been added to note this point more clearly:

In Section 3.2.1:

"Multiple satellite NO2 retrievals obtained at different overpass times have the potential to constrain diurnal emission variability (e.g., Lin et al., 2010), although differences between the different NO2 retrievals and errors in model processes could introduce artificial corrections (see also Section 5.2). Note that the retrievals from different instruments used are all based on the same retrieval method (DOMINO v2, TM4NO2A v2) and largely consistent ancillary data, which limits the discrepancies between the data sets to large degree (Boersma et al., 2008) (see Section 2.3.1). We also acknowledge that differences between the surface reflectivity and cloud data used may lead to some structural uncertainty between the morning and afternoon sensors, although numerous validation studies pointed out that the three NO2 column retrievals agree well with independent reference data (e.g., Irie et al., 2011; Ma et al., 2013)."

In Section 5.2:

"Thus, the model errors could artificially affect the diurnal emission variability."

Over Mongolia, the soil-type emission diurnal variability function was applied to the a priori emissions

with a maximum in the afternoon. Thus, the negative values of Etc may imply too much afternoon emission because of the applied a priori diurnal variability. To more precisely describe the applied diurnal emission variability function, the model description in Section 2.1 has been rewritten as follows: "We applied anthropogenic-type diurnal variations for total emissions with maxima in morning and in evening with a factor of about 1.5 (black dotted line in Fig. 1, for which the daily mean hourly emission value is 1) in Europe, eastern China, South Korea, Japan, India, and North America; biomass burning-type variations with a rapid increase in morning and maximal emissions in the mid-day with a maximum factor of about 4 in North and central Africa, southeast Asia, and northern and central South America; and soil-type diurnal variations with maximal emissions in afternoon with a factor of about 1.2 in Australia, Sahara, western China, and Mongolia."

The relevant discussions in Section 5.2 have been rewritten as follows:
"The optimized Etc for biomass-burning and soil emission dominant regions are mostly slightly negative, which may suggest that the applied diurnal emission variability with an afternoon maximum (see Section 2.1) was misleading for some regions. In contrast, they are positive for most of the ocean. These results suggest the need to not only correct diurnal $NO_2$ variations, but also account for the differences in the sampling and bias between OMI and other instruments as well as the influences of model errors."

*3. Although the paper is already quite long, I would expect at least some comparisons with independent $NO_2$ measurements. The reader has no clue regarding how the model performs for vertical profiles of $NO_2$. Also, given the focus of the paper on the diurnal cycle, comparisons with ground-based remote sensing data could be useful. Therefore, although this article is clearly interesting and the methodology appears generally sound, I recommend that the authors try to address those main comments, as well as the other comments listed below, before it can be published in ACP.*

Comparisons with independent airborne and ground-based lidar measurements have been added. The data used are described in Section 2.4, and the validation results of the vertical profile are shown in Fig. 4 and discussed in Section 3.4 of the revised manuscript. We will not discuss detailed diurnal variations of the simulated $NO_2$ profiles, since this requires careful discussions and is beyond the scope of this paper.

*Page 5, 1st full paragraph: I find odd to apply a unique diurnal cycle to all emissions in a given region, e.g. the anthropogenic-type cycle over Europe, eastern China, Japan, North America. This is strange. Why not make a weighted average based on the fractional a priori contribution of anthropogenic, biomass burning and soil emissions? The diurnal cycle in New York doesn't have to be the same as in Wyoming.*

Considering the large uncertainty in the individual inventory, we decided to use the simple diurnal emission variability scheme based on the dominant emission category, as in other global model studies (e.g., Boersma et al., 2008; 2011). Nevertheless, we confirmed that this simple scheme leads to significant improvements on the model performance (Miyazaki et al., 2012a).

*Page 5, 2nd paragraph: What is the vertical LNOx profile parameterization?*

The following sentence has been added:
"The vertical profiles of the LNOx sources are determined on the basis of the C-shaped profile given by Pickering et al. (1998)."

*Page 8, 1st full paragraph: Apparently the retrievals of Boersma et al. (2011, 2004) are used. But then the section goes on mentioning reduced errors based on Maasakkers (2013) even though the retrieval of Maasakkers is not used. This is difficult to understand.*

The paragraph has been rewritten for easier understanding.

*Page 10, lines 9-10: The assimilation effectively corrected the NO2 columns at the different overpass times. The complete diurnal cycle of NO2 concentration is an entirely different story. The model performance could be checked by comparisons with ground-based remote sensing data.*

I agree with this comment. The sentence has been replaced by:
"Considering the short lifetime and rapid diurnal variation of biomass burning activity at low latitudes, these improvements suggest that the assimilation of multiple-species and multiple NO2 measurements effectively corrected the temporal changes in the tropospheric NO2 column between the different overpass times."
Comparisons with ground-based measurements have been added. Please see my reply above.

*Page 10, line 13: Here and elsewhere, the observation errors for highly polluted cases are considered very large. But the relative errors there are often of the order of 25-35% which is generally less than at more remote locations. Of course during the winter, things are different due to large zenith angles, clouds, and snow.*

We here discuss the absolute errors, which are larger over polluted areas than over remote areas.

*Page 10, line 17: Some explanation on why the model lifetime of NOx would be too short would be useful.*

The sentence has been rewritten as:

"The remaining errors may also result from model errors such as too short lifetime of NOx through the NO2+OH and NO+HO2 reactions and the reactive uptake of NO2 and N2O5 by aerosols (e.g., Lin et al., 2012b; Stavrakou et al. 2013)."

*Page 12, on trends: The total number of observations changes during the 10-year period. Also the different satellites provided data during different periods. Some comments on possible consequences for trend estimation are needed.*

The relevant sentence in Section 5.3 has been rewritten as follows:

"Discontinuities in the assimilated measurements (e.g., lack of most TES retrievals after 2010, OMI row anomaly since January 2009, and the limited data coverage of SCIAMACHY (before February 2012) and GOME-2 (after January 2007)) may also affect long-term emission estimates."

*Page 12, lines 15-17: The emission factors are indeed uncertain, but so are also the biomass burnt estimates.*

The sentence has been rewritten as:

"... large uncertainties in emission factors and biomass burnt estimates used in the inventories."

*Page 12, lines 17-18: Note that the year-to-year variations over South America are very large.*

To clarify the meaning and in response to a suggestion from another referee, the sentence has been rewritten as:

"The weak year-to-year variations in the a priori emissions are partly attributable to the use of climatology after 2011"

*Page 12, lines 30-34: the increases aren't highest just over cities and lowest over remote areas. The entire N-E China and the Guangzhou area show large increases. The Chengdu/Chongqing area (with emission decreases) is certainly not "remote". Over N-E China, given the model resolution, it is not possible to distinguish urban from rural areas. Furthermore, Inner Mongolia shows large increases.*

Thank you for this point. The sentence has been rewritten as:

"At the grid scale, the estimated emissions are higher than the a priori emissions over northern and eastern China, such as Beijing (+58 % at the nearest grid point), Tianjin (+97 %), Nanjing (+30 %), and around Guangzhou (+78 %), whereas they are lower around Chengdu and Chongqing (Fig. 10)."

*Page 14, 1st full paragraph: The results regarding the trends in Europe are difficult to understand. Could you compare with previous studies for Europe, e.g. Curier et al. (Remote Sens. Environ. 2014, doi:10.1016/j.rse.2014.03.032)?*

The following sentence has been added:

"Strong negative emission trends over these regions were similarly found by Curier et al. (2014) for 2005--2010."

*From Figure 3, the OMI observations indicate a positive NO2 trend, whereas GOME-2 shows the opposite trend. Such difference cannot be due the diurnal cycle of NO2. Apparently those instruments have drifts which can be interpreted as emission trends. Please comment on this.*

Firstly, as the estimated emission does not follow a simple linear trend, the comparison of linear trends between the three sensors estimated for different time periods is difficult. Secondly, the positive trend in OMI is partly attributed to very high concentrations in November 2011. We confirmed that the weather for November 2011 was unusual in Europe. For instance, in the Netherlands, it was the driest month in 100 years of measurements, with ten times less rainfall than the climatological mean. Strong high-pressure systems prevented rain from washing out the NO2 pollution. GOME-2 and SCIAMACHY do not reveal such high concentrations in November 2011, although the reason for this is unclear (the different sampling rates between the sensors could partly explain this). This difference can partly explain the different trend between the sensors. Thirdly, we point out that the trends (or year-to-year variations) found from the three sensors are mostly consistent, except for Europe. Further efforts are required to investigate the observed NO2 trends from the three sensors. However, this is beyond the scope of this paper.

*Page 14, line 23 "The summertime peak enhancement is obvious over remote regions": Could you substantiate that claim?*

The sentence has been rewritten as:

"The summertime peak enhancement is obvious over remote regions such as high-temperature agricultural land over the South Atlantic, the East South Central, and the Southwestern United States,..."

*Page 15, line 10: Couldn't this be verified with e.g. MODIS fire counts or other biomass burning proxies?*

The cited paper (Venkataraman et al., 2006) confirmed the fact using MODIS fire counts.

*Page 15, line 27: Although temperature has some effect, the shorter NOx life times at tropical latitudes such as India are primarily due to higher photolysis rates and specific humidity.*

The sentence has been rewritten as:
"In contrast, tropospheric NO2 columns over India are much lower compared to those in northern midlatitude polluted areas, as a result of the high values of temperature, photolysis rates, and specific humidity, leading to shorter NO2 lifetimes throughout the year (Beirle at al., 2011)."

*Page 16, line 3: Why would high resolution analysis be required? This shouldn't be so complicated. For example, biomass burning has a distinct seasonality which can be probed at coarse resolution.*

This sentence describes point sources (i.e., power plants) over Southern Africa. For such cases, high-resolution analysis is required to distinguish between different emissions sources, whereas coarse resolution models are known to underestimate NO2 concentration (Valin et al., 2011). To clarify these points, the sentence has been rewritten as:
"The various emission sources may have experienced different variations, and high resolution emission analysis is required to understand the detailed spatial variation in these emissions and to obtain unbiased emission estimates (Valin et al., 2011)."

*Page 16, end of section 4.2.6. The high temporal correlation between N. Africa and Central Africa is interesting. Would this be related to biomass burning or to soil emissions (or both)? Examination of MODIS fire counts could help, also possibly temperature data. Is this correlation also found in the NO2-only assimilation?*

A similar correlation is found in the NO2-only assimilation. This is indeed interesting. However, detailed analysis of the causal mechanisms for individual cases is beyond the scope of this study.

*Page 19, first paragraph: Basically, the improved ozone is due to the general increase in NOx emissions over all regions, whereas the a priori model seems to have a negative bias in surface ozone. In the study of Travis et al. (ACPD, 2016, doi:10.5194/acp-2016-110), NOx emissions over the U.S. are found to be largely overestimated in comparisons with aircraft data.*

We understand some models have a positive ozone bias, and these models may have NO2 biases that also differ from our results. The description in this paper is of the MIROC simulation results, and we do not attempt to generalize the implication of this to other models.

*Page 19, last full paragraph: Could you also provide the global tropospheric chemical lifetime of methane (or methyl chloroform) in the model?*

The methane lifetime was not calculated in the simulation.

*Page 20, first sentence: "The inverse lifetime is expected to be proportional to the ratio of NOx to NO2". It's the other way around. Increase the NOx to NO2 ratio should increase the fraction of NO (which does not react with OH) and therefore decrease the sink of NOx, i.e. the inverse lifetime. The main effect of a NOx emission increase is (most often) increased OH levels and therefore shorter NOx lifetime. The point which is made in this paragraph is unclear.*

To clarify the meaning, the paragraph has been rewritten as follows:
"To elucidate the changes in the NOx chemical lifetime, Table 6 compares the lower tropospheric OH concentration and the ratio of the regional mean surface NOx emissions and lower tropospheric NO2 concentrations (averaged from the surface to 790 hPa) between the multiple-species data assimilation and the model simulation (NOx-emi/NO2) in the boreal summer. It was confirmed that both the concentration assimilation (mainly TES O3 and MOPITT CO measurements) and the changes in surface NOx emissions lead to an increase in the OH concentration in the lower troposphere. Meanwhile, the increased ratio of NOx to NO2 (i.e. increased fraction of NO) in the multiple-species assimilation compared to the model simulation indicates that the HO2+NO reaction, which is the source of OH, is enhanced in the multiple-species assimilation. These results suggest that NOx chemical lifetime is decreased because of increased OH concentrations (through the NO2+OH reaction, which acts as the main sink of NOx) in the multiple-species data assimilation over most industrial and biomass burning areas, and demonstrate the utility of the multiple-species assimilation to constrain the tropospheric chemistry (i.e. chemical regime) controlling NOx variations and to improve surface NOx emission inversions."

*Page 20, lines 20-25: The large adjustments are first said to suggest a change in diurnal evolution of the emissions. Then they are said to suggest other possible causes related to the model or the retrievals. Correct, but then the first suggestion is not necessary. Values of Etc as negative as -0.6 or -0.8 are found at some locations, which are impossibly large. Large Etc should be found only in areas where traffic is the dominant source. This does not appear to be the case (Fig. 13).*

I understand that some results are difficult to understand based on a change in the diurnal evolution of the emissions, but both possibilities remain valid. We cannot ignore either of them, without any evidence. Therefore, we believe that the current statement is reasonable. Please also see our reply above.

*Page 22, line 4: Is the given observed NO2 concentration trend for OMI or for all sensors? The trend appears very different between GOME-2 and OMI.*

Yes, the statement is based on the OMI observations, which is described in the revised manuscript. As shown by the black straight line in Fig. 3, the trend is different between the sensors. To explain this, the following sentence has been added. Please also see my reply above.
"Note that the linear trend in the observed concentration is different between the instruments over Europe (c.f., Fig. 3)."

*Page 22, line 6: Why would NO2 have become more long-lived? Does OH show a negative trend in this region? If so, what are the causes for this trend? Note that the fraction of NO2 to NOx is determined mostly by ozone and the photolysis rate of NO2. A shift in NO2:NOx emission ratio does not matter much except directly over emission areas (titration effect). The paragraph seems to imply that the NO2:NOx emission ratio in the model has changed over the 10-year period. Is this true?*

This paragraph describes how the NO2 concentration/NOx emission ratio has changed over the ten-year period over Europe. OH trends and the causal mechanisms are not discussed in the manuscript, since there could be many factors affecting them, as suggested by the reviewer. Therefore, as described in the manuscript, further efforts will be required to explain these mechanisms (in a separate study).

***Technical corrections***

*Page 1: "Forkert" should be "Folkert"*

Corrected.

*p. 1 l. 6: "biased" should be "biases"*

Corrected.

*p.1 l. 8: "the development" : do you mean the evolution?*
Yes, replaced.

*p. 2 l. 2: "traffic rush hours, economic activity. . ." those are not "source categories". Sentence is confusing, please rephrase.*

The sentence has been replaced by:
"Examples include traffic rush hours, economic activity, biomass-burning activity, wintertime-heating of buildings, and rain-induced emission pulses of NOx".

*p. 2 l. 25: Kalam should be Kalman*

Corrected.

*p. 2 l. 33: insert a hyphen between "multi" and "constituent" same line: replace advancement by advance or progress*

Corrected.

*p. 3, l. 21-22: The sentence "The OH magnitude and gradient is the primary chemical pathway for propagating observational information..." does not make much sense. Rephrase or delete.*

The sentence has been rewritten as follow:
"The changes in OH are the important chemical pathway for propagating observational information between various species and for modulating the chemical lifetimes among these species."

*p. 3 l. 31: Replace maybe "an EnKF technique" by "a variant of the EnKF technique"*

Replaced.

*p. 4 l. 25: Explain "background spread"*

The sentence has been replaced by
"...of background error covariance in the stratosphere, as estimated from ensemble model simulations,.."

*p. 4 l. 30: Isn't Yienger and Levy (1995) the correct reference for GEIA NOx? Please check.*

Corrected.

*p. 6 lines 12, 16, 22: I think "Ets" should be "Etc"*

Corrected.

*p. 6 l. 31: GOME-2 (not GOME-II)*

Corrected.

*p. 10 l. 24: Here and at other instances, replace "c.f." by more standard phrasing (e.g. "see")*

Corrected.

*p. 11 l. 12: the sentence seems to imply that the chemical lifetime of NOx might be underestimated, which is not what you mean here. Rephrase.*

The words "(and/or chemical lifetime of NOx)" have been removed.

*p. 11 l. 20: I suppose you mean GFED 3 here, not EDGAR 4.2 (see section 2.1)*

Corrected.

*p. 12 l. 9: "southern parts of the Eurasian continent" : don't you simply mean China? The seasonal variation over Southeast Asia does not show a summer maximum, so it does not fit into the point made in this sentence.*

Corrected.

*p. 12 l. 18 "assumptions applied for the a priori emissions" I think you could be more specific (use of climatology after 2011)*

Corrected.

*p. 12 l. 27: "The EDGAR v4 emissions are too low": that statement is too blunt for several reasons. Replace "EDGAR v4" by "our a priori inventory" (since EDGAR for 2008 is used after 2008, and since soil emissions are not from EDGAR). Furthermore, add something like "Our assimilation indicates that. . .".*

Replaced by "Out a priori inventory is..."

*p. 12 l. 28: "too low by a factor of 0.6": awkward. Should be too low by a factor of 1/0.6 (i.e. about 1.7)*

Replaced by "by about 40 %".

*p.12 l. 29: emissions are maximum in June, not July.*

Replaced.

*p. 13 l. 34: "in the reported mobile emissions": why specifically in this source category?*

The sentence has been removed.

*p. 14 l. 1: replace "reveal" by "show"*

Replaced.

*p. 14 l. 2: replace "by" by "after"*

Replaced.

*p. 14, l. 18: "around Atlanta (...) and Denver": this seems to indicate that increments are found mostly over cities, which is not true. Consider replacing by "Southeast US and most of Western US"*

Replaced by "the Southeast United States (e.g., +23 % near Atlanta) and most of the Western United States (e.g., +26 % near Denver),"

*p. 14 l. 19: delete "over" after "around"*

Removed.

*p. 14 l. 20: Los Angeles*

Corrected.

*p. 15 l. 9: I think the word "boreal" is superfluous here (and at many other instances in the text)*

Removed.

*p. 15 l. 20-21: "particularly strong increase around Delhi" but the changes over Delhi are lower than the regional average!*

Delhi has been removed from this statement.

*p. 16 l. 5: Replace "by data assimilation" by "due to data assimilation"*

Replaced

*p. 16 l. 33: replace "reflection" by "reflecting" same line: replace "when" by "whereas"*

Replaced.

*p. 19 l. 17: LNOx (instead of LNO)*

Corrected.

*p. 19 l. 34: here and elsewhere in the manuscript, insert hyphen between "multiple" and "species"*

Corrected.

*p. 20 l. 17: Replace "rom the three. . ." by "constrained by the three. . ."*

Replaced.

*p. 21 l. 8: "using either model after data assimilation" : awkward, the model is used for data assimilation*

Replaced by "either model and data assimilation".

*p. 32: Table 2 and elsewhere: Replace "Australis" by "Australia"*

Corrected.

*p. 41 Figure 6: it is impossible to distinguish black and dark blue on the "South America" plot. Consider using other colors.*

Changed.

*p. 44, Figure 9: the title of the middle panels should be "A posteriori – A priori". Same for the title of the right panels, the minus sign is missing.*

Corrected.

1 Decadal changes in global surface NOx emissions from multi-constituent satellite data
2 assimilation

4 \begin{abstract}

6 Global surface emissions of nitrogen oxides (\chem{NO_x}) over a ten-year period (2005--
7 2014) are estimated from an assimilation of multiple satellite datasets: tropospheric
8 \chem{NO_2} columns from OMI, GOME-2, and SCIAMACHY; \chem{O_3} profiles from
9 TES; CO profiles from MOPITT; and \chem{O_3} and \chem{HNO_3} profiles from MLS
10 using an ensemble Kalman filter technique. Chemical concentrations of various species and
11 emission sources of several precursors are simultaneously optimized. This is expected to
12 improve the emission inversion because the emission estimates are influenced by
13 biases in the modelled tropospheric chemistry, which can be partly corrected by also
14 optimizing the concentrations. We present detailed distributions of the estimated emission
15 distributions for all major regions, the diurnal and seasonal variability, and the
16 evolution of these emissions over the ten-year period. The estimated regional total emissions
17 show a strong positive trend over India (+29 \%/decade), China (+26 \%/decade), and the
18 Middle East (+20 \%/decade), and a negative trend over the United States (-38 \%/decade),
19 Southern Africa (-8.2 \%/decade), and western Europe (-8.8 \%/decade). The negative trends
20 in the United States and western Europe are larger during 2005--2010 relative to 2011--2014,
21 whereas the trend in China becomes negative after 2011. The data assimilation also suggests a
22 large uncertainty in anthropogenic and fire-related emission factors and an important
23 underestimation of soil \chem{NO_x} sources in the emission inventories. Despite the large
24 trends observed for individual regions, the global total emission is almost constant between
25 2005 (47.9 \unit{Tg\,N\,yr^{-1}}) and 2014 (47.5 \unit{Tg\,N\,yr^{-1}}).

27 \end{abstract}

29 \introduction

[revised manuscript text omitted]
 \chem{NO_2}, from TES of \chem{O_3}, from MOPITT measurements of \chem{CO}, and from MLS of \chem{O_3} and \chem{HNO_3}. The retrieved concentration and observation error information were obtained for each retrieval, where the observation error included contributions from smoothing errors, model parameter errors, forward model errors, geophysical noise, and instrument errors. These combined errors, together with a representativeness error for super observations (Miyazaki et al., 2012a), were considered in the observation error matrix ($\mathbf{R}$) for data assimilation.

For the assimilation of the satellite retrievals, observation operators ($\mathrm{H}$) were developed, consisting of the spatial interpolation operator ($S$), a priori profile in the satellite retrievals ($\vec{x}_{\text{apriori}}$), and an averaging kernel ($\mathbf{A}$). This operator mapped the model fields ($\vec{x}_{i}^{\mathrm{b}}$) into retrieval space ($\vec{y}_i^{\mathrm{b}}$), as follows:

\begin{align}

&

\vec{y}_i^{\mathrm{b}}= H(

\vec{x}_{i}^{\mathrm{b}}

) = \vec{x}_{\text{apriori}} + \vec{A} (S(\vec{x}_i^{\mathrm{b}})-

\vec{x}_{\text{apriori}}),

\end{align}

[revised manuscript text omitted]

The improved representation of \chem{NO_x} emissions is confirmed by the better agreement of simulated \chem{O_3} concentrations with independent ozonesonde observations using \chem{NO_x} emissions from  multiple-species assimilation than those using \chem{NO_x} emissions from \chem{NO_2}-only data assimilation, which was also demonstrated by Miyazaki and Eskes (2013). After 2010, TES \chem{O_3} retrievals were not assimilated because of the lack of standard observations. Even so, the optimized surface \chem{NO_x} emissions from the  multiple-species assimilation improved agreements with TES \chem{O_3} ver. 6 special observations during 2011--2014 for most locations (Table S1). These results indicate that  multiple-species measurements provide important information for improving surface \chem{NO_x} source estimations and improve the chemical consistency including the relation between concentrations and the estimated emissions. Note that the emissions of \chem{O_3} precursors other than \chem{NO_x}, such as \chem{VOCs}, and various model processes in atmospheric transport

and chemistry influences the model performance. The optimization of additional precursors emissions and the improvement of the forecast model could be important for improving \chem{O_3} simulations, as discussed in our previous studies (Miyazaki et al., 2012b; 2015).

\chem{LNO_x} sources are important for a realistic representation of tropospheric \chem{NO_2} columns, which are optimized from data assimilation in our framework. Using the  multiple-species data assimilation, the ten-year mean global \chem{LNO_x} source amount was estimated at 5.8 \unit{Tg\,N\,yr^{-1}}, in contrast to 5.3 \unit{Tg\,N\,yr^{-1}} estimated from the model simulation and 6.3$\pm$1.4 \unit{Tg\,N\,yr^{-1}} in our previous data assimilation estimate (Miyazaki et al., 2014). The data assimilation increments for \chem{LNO_x} sources are large in the upper troposphere in the NH and the TR, in which non-\chem{NO_2} measurements with different vertical sensitivities provided important constraints. Through its influence on simulated tropospheric \chem{NO_2} columns, for instance, the inclusion of the \chem{LNO_x} source optimisation altered the surface \chem{NO_x} emission estimates over eastern China by up to 12\% in summer. Moreover, surface CO emissions increased by 10 \% in the NH by the assimilation of MOPITT CO measurements in our system. Both optimised \chem{LNO_x} sources and CO emissions reveal enhanced seasonal and interannual variations over many regions after data assimilation, providing important constraints on long-term estimates of surface \chem{NO_x} emissions, through their influence on OH and thus the \chem{NO_x} chemical lifetime.

[revised manuscript text omitted]

($\Delta$\chem{NO_x}-emi/\chem{NO_2}) between the data assimilation run and the model simulation in the boreal summer (averaged over June--August) over the 2005--2014 period.

Table 7: Regional and ten-year mean correction factor for the emission diurnal variability ($Etc$) for 2005--2014.

Figure 1: Schematic diagram of the correction scheme for the emission diurnal variation for a case with $Etc=-0.3$. The black dotted time represents the a priori emission diurnal variability function ($Et$) for anthropogenic emissions. The black solid line represents the a posteriori emission variation after applying the daily emission scaling factor ($Et \times Es$). The blue line represents the correction factor for the emission diurnal variability ($Etc$). The red line represents the a posteriori emission variation after applying the daily emission scaling factor and the correction factor for the emission diurnal variability ($Et \times Es - Etc$ for 07:30--10:30, and $Et \times Es + Etc$ for 10:30--13:30).

Figure 2: Global distributions of the tropospheric \chem{NO_2} columns (in $10^{15}$\,\unit{molec\,cm^{-2}}). The results are shown for OMI (left columns, sampling time $\approx$13:00 hrs) for 2005--2014, SCIAMACHY (middle columns, 10:00 hrs) for 2005--2011, and GOME-2 (right columns, 09:30 hrs) for 2007--2014. Upper rows show the tropospheric \chem{NO_2} columns obtained from the satellite retrievals (OBS); centre shows the difference between  the model simulation and the satellite retrievals (Model-OBS); and lower rows  show the difference between the data assimilation and the satellite retrievals (Assim-OBS).

Figure 3: Time series of regional monthly mean tropospheric \chem{NO_2} columns (in $10^{15}$\,\unit{molec\,cm^{-2}}) averaged over China (110--123$^\circ$E, 30--40$^\circ$N), Europe (10$^\circ$W--30$^\circ$E, 35--60$^\circ$N), the United States (70--125$^\circ$W, 28--50$^\circ$N), South America (50--70$^\circ$W, 20$^\circ$S--Equator), North Africa (20$^\circ$W--40$^\circ$E, Equator--20$^\circ$N), Central Africa (10--40$^\circ$E, Equator--20$^\circ$S), Southern Africa (25--34$^\circ$E, 22--31$^\circ$S), Southeast Asia (96--105$^\circ$E, 10--20$^\circ$N), Australia (113--155$^\circ$E, 11--

44$^\circ$S), and India (68--89$^\circ$E, 8--33$^\circ$N) obtained from the satellite retrievals (black), model simulation (blue), and the data assimilation (red). The model simulation and data assimilation results were obtained at the local overpass time of the retrievals with applying the averaging kernel.

Figure 4: (Left panel) Mean vertical \chem{NO_2} profiles obtained during the ARCTAS campaign in June--July 2009. The black represents the observation; the blue line represents the model simulation; the red line represents the data assimilation. The error bars represent the standard deviation. (Right six panels) Scatter plots of \chem{NO_2} concentrations for the data assimilation (top) and the model simulation (bottom) during the DANDELIONS campaign (in $\mu g m^{-3}$) in September 2006 (second left columns) and during the INTEX-B campaign (in \unit{pptv}) on March 9, 2006 (third left columns) and March 11, 2006 (right columns). The straight lines represent linear regression lines for each level. Each line represents a linear fit to the points of the same colour, and the colours represent the altitude (or pressure) level.

Figure 54: Global distributions of surface \chem{NO_x} emissions (in $10^{-13}\unit{kg m^{-2} s^{-1}}$) averaged over 2005--2014. The a priori emissions (top), a posteriori emissions from the data assimilation run (middle), and analysis increment (bottom) are shown.

Figure 65: Global distribution of linear trend of the a posteriori surface \chem{NO_x} emissions (in $10^{-13}\unit{kg m^{-2} s^{-1}}$ per decade) for the period 2005--2014. The red (blue) colour indicates positive (negative) trends.

Figure 76: Time series of monthly total regional surface \chem{NO_x} emissions (in \unit{Tg\,N\,yr^{-1}} ) obtained from the a priori emissions (black lines) and the a posteriori emissions (red lines) for the period 2005--2014. The results are also shown for EDGAR-HTAP v2 emissions (blue green lines) for the years 2008 and 2010.

Figure 87: Seasonal variations of the regional surface \chem{NO_x} emissions (in \unit{Tg\,N\,yr^{-1}}) obtained from the a priori emissions (black line) and the a posteriori emissions (red line) averaged over the period 2005--2014. The results are also shown for the a posteriori emissions for individual years during 2005--2014 (yellow lines).

Figure 98: Time series of the difference (in \%) of the annual mean a posteriori surface \chem{NO_x} emissions relative to the 2005 emissions in the period 2005--2014 for India (yellow), China (blue), Europe (light blue), western Europe (light blue dashed line), Southern Africa (red), and the United States (green).

Figure 109: The regional distribution of ten-year mean surface \chem{NO_x} emissions (in $10^{-13}\unit{kg m^{-2} s^{-1}}$) over East Asia (upper panels), Europe (upper middle panels), the United States (lower middle panels), and Southeast Asia (lower panels) obtained from the a posteriori emissions in the period 2005--2014 (left panels), and the difference between the a posteriori emissions and a priori emissions in the period 2005-2014 (centre panels), and between the a posteriori emissions and EDGAR-HTAP v2 emissions for the years 2008 and 2010 (right panels). The black square line represents the region used for the regional mean analysis.

Figure 11: Global distribution of linear trend of the a posteriori surface \chem{NO_x} emissions for the period 2005--2010 (left) and 2011--2014 (right). The red (blue) colour indicates positive (negative) trends.

Figure 120: The regional distribution of the linear trend in surface \chem{NO_x} emissions (in $10^{-13}\unit{kg m^{-2} s^{-1}}$ per decade) during 2005--2014 over East Asia (upper left), Europe (upper right), the United States (bottom left), and Southeast Asia (bottom right), obtained from the a posteriori emissions. The black square line represents the region used for the regional mean analysis.

Figure 13: Global distribution of the ten-year mean \chem{OH} concentration (in $10^6$ \unit{molecules cm^{-3}}) in the data assimilation run (top) and its difference between the data assimilation run and the model simulation (bottom) averaged over June, July, and August over the 2005--2014 period at 850 hPa.

Figure 14: Global distribution of the annual mean correction factor for the emission diurnal variability ($Etc$) for the period 2005--2014.

---

## Author Response (AR2)

**Author's comments in reply to the anonymous referee for "Decadal changes in global surface NOx emissions from multi-constituent satellite data assimilation" by K. Miyazaki et al.**

We want to thank the referee for the helpful comments. We have revised the manuscript according to the comments, and hope that the revised version is now suitable for publication. Below are the referee comments in italics with our replies in normal font.

*Reply to Referee #2*

*The authors did some effort to address my main concerns. In particular the possible various limitations associated with the method are better explained in the manuscript. A section on validation using independent NO2 data has been added, although I found it mostly unconvincing due to several issues (see further below). As in my first review, my primary concern is the fact that the authors fail to explain the role of the different datasets in the assimilation. It is said repeatedly that non-NO2 observations have a large impact on the optimization of the emissions. And this is indeed shown in the new Table 4. But what is, more precisely, the role of each dataset? We are left almost clueless on that matter. It is not enough to claim that the lifetime of NOx is better represented by the model when those observations are used. I would like a discussion explaining, qualitatively and quantitatively how the measurements of ozone and CO influence the assimilation. It is therefore necessary to, first, present the model biases for CO and O3, and secondly, discuss how the correction of those biases by the assimilation impacts the optimization of NOx emissions.*

The role of each measurement on the estimated emissions is intensively discussed in Section 5.1 of the revised manuscript. This revision is based on new results from Observing System Experiments (OSEs) over several months. Note that conducting OSEs for a longer time period (i.e., the entire reanalysis period) would involve a huge computational cost, making it very difficult. Meanwhile, understanding the OSE results for each measurement associated with detailed model errors is not always straightforward because of the complex chemical processes. Since we demonstrate the impact of non-NO2 measurements on the estimated NOx emissions and OH and intensively discuss these results in the revised manuscript, we think that no further discussion is required to present the value of the multiple-species assimilation in this study.

*Regarding the diurnal cycle (my second major comment in my previous review), the authors have not addressed the main issue, which is that the modification to the diurnal cycle of emissions deduced from satellite measurements is not credible as it implies much stronger rush hour emission peaks even in*

*regions where mobile emissions (cars) are not the main source category. The most negative values of the Etc parameter are found (not in Mongolia but) in Inner Mongolia, i.e. in Northern China, around 110 W, 41 N, in a region with very strong emission trends (see Figure 12) due to anthropogenic emissions, i.e. power plants and industries (not cars). This should be mentioned and shortly discussed.*

To highlight the large Etc values and their potential problems, the following sentence has been added in the revised manuscript:

"Large negative values of Etc are found over northern China, northern India, and the Middle East, where various emission sources (not only mobile sources with morning peaks) could be important."

Note that the limitation of the estimated diurnal emission variability is already discussed in Section 5.2 as follows:

"These results also suggest a larger negative bias in simulated tropospheric NO2 column in the morning, associated with errors in the chemical lifetime and atmospheric transports (e.g., boundary layer development) and also associated with biases between the different NO2 retrievals. Thus, the model errors could artificially affect the diurnal emission variability."

*Comparisons with airborne and lidar NO2 measurements (from ARCTAS, INTEX-B and DANDELIONS) have been added (Figure 4). But the ARCTAS profile is almost useless as the assimilation does not change the NO2 tropospheric profile, except in the upper troposphere (UT). The mechanism by which NO2 is increased by the assimilation in the UT is not explained, except for the fact that it is related to HNO3 MLS observations. The text mentions the effect of inter-species correlations. But it is the first (and only) time that such correlations are mentioned. More details would be needed to explain how such correlations are set up in the system. Furthermore, the ARCTAS NO2 measured during ARCTAS in the UT is known to be too high. Browne et al. (ACP 11, 4209-4219, 2011) showed that at low temperature, a large fraction of the measured XNO2 is due to dissociation of CH3O2NO2 and HO2NO2 in the inlet prior to detection, leading to a large overestimation of NO2 in the UT. Also for DANDELIONS, only morning measurements are used for validation, whereas afternoon observations are rejected without a good reason. Only two INTEX-B flights are used, and I find the presentation awkward. For the March 9 flight, I find very weird that the observed values are so low in the layer closer to the surface (most measurements are well below 100 ppt). I checked the INTEX-B files (from www-air.larc.nasa.gov/missions/merges/) and I estimate that the average NO2 in that layer for that day was 228 ppt, even when excluding the Mexico and Houston areas. Please verify the data selection for that flight. Furthermore, the authors claim to see an improvement in the slopes of the linear regressions, but what meaning is there in such slopes when the correlations are so low? It would be useful to show*

*vertical profiles for INTEX-B (as for ARCTAS) and possibly for another campaign like INTEX-A or the more recent ones (SEAC4RS).*

Concerning the improvement in the upper tropospheric NO2 for the ARCTAS profile, the relevant sentences have been rewritten as follows:

"In contrast, the data assimilation mostly removed the model negative bias in the upper troposphere and lower stratosphere, mainly because of the MLS O3 and HNO3 data assimilation and through the use of the inter-species correlation that was determined using background error covariances estimated from ensemble model simulations (c.f., Section 2.2). An estimated inter-species correlation is demonstrated in Miyazaki et al. (2012b) in Fig. 3, which shows a strong positive correlation between the concentrations of NO2 with those of O3 and HNO3, reflecting complex tropospheric chemical processes. The data assimilation widely influences the NOx and NOy species in both analysis and forecast steps. This improvement cannot be achieved using the NO2 measurements only."

The possible overestimation in ARCTAS measurement is noted in the revised manuscript as follows:
"Note that Browne et al. (2011) investigated that the observed NO2 concentrations could be too high in the upper troposphere."

A clear explanation about the use of morning measurements for the DANDELIONS profile is included in the revised manuscript as follows:
"The model grid points used for the interpolation around Cabauw are located in Belgium, northeastern Netherlands, western Germany, and on the North Sea. Boundary layer conditions are different among the grid points, especially between land and ocean. To avoid a possibly large error of representativeness in the validation, particularly under the different boundary layer condition, the profiles obtained in the morning (before 12:00 p.m.) were used because the differences between land and sea mixing layer depths are then still relatively small, following Miyazaki et al. (2012a)."

The comparisons of NO2 profiles have been expanded in the revised manuscript, including those from all flights of INTEX-B, DC3, and SEAC4RS campaigns. INTEX-A was conducted in 2004, which is not covered by the calculation in this study. Section 2.4 has been expanded to describe the additional aircraft data used.

The discussions on the vertical NO2 profiles in Section 3.4 have been rewritten as follows:
"Compared with the INTEX-B and DC3 profiles, both the model and assimilation are too low in the middle/upper troposphere, whereas in the lower troposphere these are too high compared with the DC3

profile and too low compared with the INTEX-B profile. Compared with the SEAC4RS profile, both the model and assimilation are too high in the lower troposphere. Because of the coarse model resolution (approximately 2.8°), the model has difficulty in representing the spatial footprint of the measurement, and this could cause large differences near the surface for comparisons at urban sites. The near-surface concentration will be sensitive to the model resolution owing to fine-scale emission distribution and transport, as well as non-linear chemical processes, as discussed in Valin et al (2011) and Miyazaki et al (2012a). The coarse model resolution may also make the improvements by data assimilation obscure."

We did not remove the scatter plots for the ARCTAS and DANDELIONS measurements from Fig. 4 because these clearly demonstrate improvements in the variability. Although the correlations are not very large, it is clear that the data assimilation improves both the slope and correlation.

*Minor comments:*

*The authors response suggested that the HNO3-forming channel of the NO+HO2 reaction is taken into account in the model. I'm very surprised by that. Some words should be provided in the model decription, including the references for the rates (is the effect of water vapor also considered?)*

The model did not consider the HNO3-forming channel of the NO+HO2 reaction, but it did consider NO+HO2->NO2+OH and OH+NO2+M->HNO3+M reactions. To avoid any confusion, the relevant sentences have been revised as follows:
"The remaining errors may also result from model errors such as too short lifetime of NOx through processes such as the NO2+OH reactions and the reactive uptake of NO2 and N2O5 by aerosols (e.g., Lin et al., 2012b; Stavrakou et al. 2013)."
"... indicates that the HO2+NO to NO2+OH reaction, which is the source of OH, is..."

*Also in their response, the authors added a sentence "The summertime peak enhancement is obvious over remote regions such as high-temerature agricultural land over the South Atlantic (...)". Agricultural land over the Atlantic? Please rephrase.*

The sentence has been replaced by "... over the East South Central and the Southwestern United States"

*Regarding the improved ozone due to higher NOx emissions: that this increase would actually deteriorate ozone in other models (e.g. Geos-Chem, cf. Travis et al. ACPD 2016) calls into question the reality of the NOx emission increase. This should be mentioned and possibly discussed in the manuscript.*

To discuss this point briefly, the sentence has been revised as follows:

"Note that the emissions of O3 precursors other than NOx, such as VOCs, and various model processes in atmospheric transport and chemistry influence the model performance. The impact of using the optimized NOx emissions may vary with models (e.g., given different forecast errors of NO2 and O3)."

*I find very weird that the authors cannot provide any indication regarding the lifetime of methane in their model. This is an essential and very standard metric of any global atmospheric model. It would be also very useful for the discussion of the assimilation results.*

Because the system used has been optimized for data assimilation calculations, several configurations, including model diagnostics and outputs, have been changed from the original model setup. We hope to include these diagnostics in future analyses.

*The discussion on the changes of the NOx lifetime states that "both the concentration assimilation (mainly TES O3 and MOPITT CO measurements) (...) lead to an increased in the OH concentrations". This is probably true but needs to be demonstrated. Also, rephrase, e.g. "both the assimilation of non-NO2 compounds (mainly TES O3 and MOPITT CO measurements) (...)".*

Fig. S2 has been added to demonstrate OH changes by the non-NO2 data assimilation. The discussions have been revised based on the OSE results. Please also see our reply above.

*In the next sentence, it is stated that HO2+NO is enhanced, and that the NOx lifetime is decreased due to higher OH in the multiple-species assimilation (compared to the model simulation). This is very probably correct, but I don't see any proof that the non-NO2 observations are essential here for that respect. Therefore, the last sentence "demonstrate the utility of multiple-species assimilation..." is unsubstantiated.*

As discussed in the revised manuscript and shown in Fig. S2, the non-NO2 measurements provided important constraints on the OH concentrations. Thus, the sentence provides a reliable statement in the revised manuscript. Please also see my reply above.

*The authors also did not answer my questions on the trends of NO2 concentrations and NOx emissions over Europe. The manuscript suggests that NO2 has become more long-lived. Surely you can check in your model outputs whether e.g. OH concentrations show a trend. The other explanation "a shift in*

*NO2:NOx emission ratios related to the increasing share of European diesel cars could have occurred"
is very strange, it is like if the authors cannot verify what they have in their model.*

The sentences have been revised to describe the OH trend over Europe as follows:

"This suggests that NO2 may have become longer-lived or has become a larger fraction of NOx over Europe over the past decade. In fact, the lower tropospheric OH concentrations show slight negative trends (by up to -5 %/decade) over most of Western Europe over the past decade (figure not shown). Another possible explanation is that a shift in NO2:NOx emission ratios related to the increasing share of European diesel cars could have occurred."

Concerting the sentence "a shift in NO2:NOx emission…", a further study is clearly needed to verify this, but providing this possibility in the manuscript is still valid.

*Other corrections*

*Page 13 line 7 "the possibility FOR improving"*
*Page 13 line 14 delete "and" before "used in data assimilation"*

Corrected.

*Page 15 line 12-13 explain why representativeness errors would be smaller in the morning compared to the afternoon.*

Please see my reply above.

*Page 15 line 24 replace "corrected" by "sampled"*
*Page 16 line 5 replace "at polar region" by "in polar regions"*
*Page 16 line 28 replace "principle" by "main"*
*Page 20 line 13 insert "only" before "a small effect"*

Corrected.

*Page 20 line 26 what is meant by "commonly"? Rephrase.*

Replaced by "also".

*Page 21 1st full paragraph: the difficulty to represent the measurements would disappear when using a large number of measurements, because the errors on the averages will cancel out. The solution is therefore to use larger datasets than used here.*

Sampling biases can be systematic because of model processes including non-linear chemistry. This is true even when we use larger datasets for fixed-point measurements and a coarse resolution model.

*Page 25, line 25 The temporal shift is actually larger than 1 month.*

The timing of peak emission for the regional emission for Europe (Fig. 8) occurred earlier by 1 month from July to June.

*Page 33 line 21 Replace "adjusted" by "modified"*

Corrected.

*Page 34 line 4: "The monthly total global emissions decrease by up to 6 TgN": that value (6TgN) is impossibly high, please check. Remember that the total global NOx source if of the order of 40-50 TgN per year.*

We confirmed that this occurred in October 2008.

*Page 35 line 1: "influences" --> "influence"*

Corrected.

*Page 36 line 15 "It was confirmed" --> "It is found" (??)*

Corrected.

*Page 38 line 4 "misleading" --> "inappropriate"*
*Page 46 line 7 Add "line" after "black"*
*Page 46 line 9 Delete "six"*

Corrected.

**Decadal changes in global surface NOx emissions from multi-constituent satellite data assimilation**

\begin{abstract}

Global surface emissions of nitrogen oxides (\chem{NO_x}) over a ten-year period (2005--2014) are estimated from an assimilation of multiple satellite datasets: tropospheric \chem{NO_2} columns from OMI, GOME-2, and SCIAMACHY; \chem{O_3} profiles from TES; CO profiles from MOPITT; and \chem{O_3} and \chem{HNO_3} profiles from MLS using an ensemble Kalman filter technique. Chemical concentrations of various species and emission sources of several precursors are simultaneously optimized. This is expected to improve the emission inversion because the emission estimates are influenced by biases in the modelled tropospheric chemistry, which can be partly corrected by also optimizing the concentrations. We present detailed distributions of the estimated emission distributions for all major regions, the diurnal and seasonal variability, and the evolution of these emissions over the ten-year period. The estimated regional total emissions show a strong positive trend over India (+29 \%/decade), China (+26 \%/decade), and the Middle East (+20 \%/decade), and a negative trend over the United States (-38 \%/decade), Southern Africa (-8.2 \%/decade), and western Europe (-8.8 \%/decade). The negative trends in the United States and western Europe are larger during 2005--2010 relative to 2011--2014, whereas the trend in China becomes negative after 2011. The data assimilation also suggests a large uncertainty in anthropogenic and fire-related emission factors and an important underestimation of soil \chem{NO_x} sources in the emission inventories. Despite the large trends observed for individual regions, the global total emission is almost constant between 2005 (47.9 \unit{Tg\,N\,yr^{-1}}) and 2014 (47.5 \unit{Tg\,N\,yr^{-1}}).

\end{abstract}

\introduction

[revised manuscript text omitted]
 \chem{NO_2}, from TES of \chem{O_3}, from MOPITT measurements of \chem{CO}, and from MLS of \chem{O_3} and \chem{HNO_3}. The retrieved concentration and observation error information were obtained for each retrieval, where the observation error included contributions from smoothing errors, model parameter errors, forward model errors, geophysical noise, and instrument errors. These combined errors, together with a representativeness error for super observations (Miyazaki et al., 2012a), were considered in the observation error matrix ($\mathbf{R}$) for data assimilation.

For the assimilation of the satellite retrievals, observation operators ($\mathrm{H}$) were developed, consisting of the spatial interpolation operator ($S$), a priori profile in the satellite retrievals ($\vec{x}_{\text{apriori}} $), and an averaging kernel ($\mathbf{A}$). This operator mapped the model fields ($\vec{x}_{i}^{\mathrm{b}}$) into retrieval space ($\vec{y}_i^{\mathrm{b}}$), as follows:

\begin{align}

 &

\vec{y}_i^{\mathrm{b}}= H(

\vec{x}_{i}^{\mathrm{b}}

) = \vec{x}_{\text{apriori}} + \vec{A} (S(\vec{x}_i^{\mathrm{b}})-

\vec{x}_{\text{apriori}}),

\end{align}

[revised manuscript text omitted]

\chem{NO_2} and TES \chem{O_3} assimilation, in contrast to 3.16 TgN in the multi-species assimilation and 2.45 TgN in the a priori emissions), whereas other non-\chem{NO_2} measurements (i.e., MOPITT and MLS) have less impact. Similar important contributions of TES \chem{O_3} measurements are found for South America in January 2008 (1.09 TgN in the \chem{NO_2}-only assimilation and 0.91 TgN in the \chem{NO_2} and TES \chem{O_3} assimilation, in contrast to 0.90 TgN in the multi-species assimilation and 0.46 TgN in the a priori emissions). These changes in \chem{NO_x} emissions are associated with negative adjustments of \chem{O_3} by the TES assimilation over South America throughout the troposphere and positive adjustments of \chem{O_3} over India in the middle troposphere, and their influence on \chem{NO_x}-\chem{OH}-\chem{O_3} chemical reactions and the \chem{LNO_x} source optimization, as discussed below.

The ten-year linear trend is also different over most industrial areas (Table 4). For instance, the positive trend for India is 34.3 \%/decade in the \chem{NO_2}-only assimilation, which is larger than the 29.2 \%/decade in the multiple-species assimilation. For the United States, the negative trend is larger in the multiple-species assimilation (-29.4 \%/decade) than in the \chem{NO_2}-only assimilation (-23.9 \%/decade). These results confirm that the assimilation of measurements for species other than \chem{NO_2} provides additional constraints on the \chem{NO_x} emissions over both anthropogenic and biomass burning regions.

The improved representation of \chem{NO_x} emissions is confirmed by the better agreement of simulated \chem{O_3} concentrations with independent ozonesonde observations using \chem{NO_x} emissions from multiple-species assimilation than those using \chem{NO_x} emissions from \chem{NO_2}-only data assimilation, which was also demonstrated by Miyazaki and Eskes (2013). After 2010, TES \chem{O_3} retrievals were not assimilated because of the lack of standard observations. Even so, the optimized surface \chem{NO_x} emissions from the multiple-species assimilation improved agreements with TES \chem{O_3} ver. 6 special observations during 2011--2014 for most locations (Table S1). These results indicate that multiple-species measurements provide important information for improving surface \chem{NO_x} source estimations and improve the chemical consistency including the relation between concentrations and the estimated emissions. Note that the emissions of \chem{O_3} precursors other than \chem{NO_x}, such as \chem{VOCs}, and various model processes in atmospheric transport and chemistry influences the model performance—. The impact of using the optimized \chem{NO_x} emissions may vary with models (e.g., given different forecast errors of \chem{NO_2} and \chem{O_3}). The optimization of additional precursors emissions and the improvement of the forecast model could be important for improving \chem{O_3} simulations, as discussed in our previous studies (Miyazaki et al., 2012b; 2015).

\chem{LNO_x} sources are important for a realistic representation of tropospheric \chem{NO_2} columns, which are optimized from data assimilation in our framework. Using the multiple-species data assimilation, the ten-year mean global \chem{LNO_x} source amount was estimated at 5.8 \unit{Tg\,N\,yr^{-1}}, in contrast to 5.3 \unit{Tg\,N\,yr^{-1}} estimated from the model simulation and 6.3$\pm$1.4 \unit{Tg\,N\,yr^{-1}} in our previous data assimilation estimate (Miyazaki et al., 2014). The data assimilation increments for \chem{LNO_x} sources are large and mostly positive in the middle and upper troposphere in the NH and the TR, in which non-\chem{NO_2} measurements with different vertical sensitivities provided important constraints. Through its influence on simulated tropospheric \chem{NO_2} columns, for instance, the inclusion of the \chem{LNO_x} source optimisation altered the surface \chem{NO_x} emission estimates over eastern China by up to 12\% in summer. Moreover, surface CO emissions increased by 10 \% in the NH by the assimilation of MOPITT CO measurements in our system. Both optimised \chem{LNO_x} sources and CO emissions reveal enhanced seasonal and interannual variations over many regions after data assimilation, providing important constraints on long-term estimates of surface \chem{NO_x} emissions, through their influence on OH and thus the \chem{NO_x} chemical lifetime.

Figure 13 shows changes in OH concentrations ($\Delta$\chem{OH}) in the lower troposphere in the boreal summer (averaged over June--August) due to data assimilation. The multiple-species assimilation changes the global OH distribution, increasing OH over most globally. As summarised in Table 6, the regional impact is large (greater than +20 \%) in tropical regions such as over the Middle East, Southeast Asia, and Central and North Africa, and over industrial areas (greater than +10 \%), such as over China, the United States, and India. These changes in OH concentrations are influenced by changes in \chem{NO_x} emissions through the assimilation of \chem{NO_2} measurements ,  but  the assimilation of non-\chem{NO_2} measurements is also important. Fig. S2 demonstrates that the assimilation of non-\chem{NO_2} measurements acts to decrease the \chem{OH} concentration in the lower and middle troposphere for most regions in June 2008. The TES assimilation mostly reduces the \chem{O_3} concentration in the tropics, which leads to a decrease of \chem{OH} concentrations. In contrast, the TES assimilation acts to increase the \chem{OH} concentration in the NH extratropics in the lower and middle troposphere. The assimilation of MOPITT \chem{CO} acts to decrease the \chem{OH} concentration in the NH, because of the increased surface \chem{CO} emissions. In contrast to the increased ~~surface CO emissions in the NH, the ten-year mean regional total CO emissions are reduced by 12\% in the tropics, leading to an increase of OH concentrations in the tropics. The assimilation of TES \chem{O_3} retrievals also significantly changes OH concentrations, which results in a significant increase in OH concentration in the extratropics by up to 15 \% in the NH extratropics in summer, as demonstrated by Miyazaki et al. (2012b). Note that tdata assimilation runwith~~ the help of methyl chloroform observations (a proxy for OH concentrations) by Patra et al. (2014).

To elucidate the changes in the \chem{NO_x} chemical lifetime, Table 6 compares the lower tropospheric \chem{OH} concentration and the ratio of the regional mean surface \chem{NO_x} emissions and lower tropospheric \chem{NO_2} concentrations (averaged from the surface to 790 hPa) between the multiple-species data assimilation and the model simulation ($\Delta$\chem{NO_x}-emi/\chem{NO_2}) in the boreal summer. The multiple-species assimilation leads to an increase in the \chem{OH} concentration in the troposphere.

 Meanwhile, the increased ratio of \chem{NO_x} to \chem{NO_2} (i.e. increased fraction of \chem{NO}) in the multiple-species assimilation compared to the model simulation indicates that the $\chem{HO_2}+\chem{NO} \to \chem{NO_2}+\chem{OH}$ reaction, which is the source of \chem{OH}, is enhanced in the multiple-species assimilation. It is also found that the assimilation of non-\chem{NO_2} measurements suppress these changes for most regions in both the \chem{OH} concentration (c.f., Fig. S2) and the \chem{NO_x}-emi/\chem{NO_2} ratio. For instance, the ten-year mean ratio over Central Africa is increased by 16.5 \% in the multiple-species assimilation, in contrast to the 19.3 \% increase in the \chem{NO_2}-only assimilation.

These results suggest that \chem{NO_x} chemical lifetime is decreased because of increased \chem{OH} concentrations (through the \chem{NO_2}+\chem{OH} reaction, which acts as the main sink of \chem{NO_x}) in the multiple-species data assimilation (and also in the \chem{NO_2}-only assimilation) than in the model simulation over most industrial and biomass burning areas. It is also suggested for many regions \chem{NO_x} chemical lifetime is longer in the multiple-species assimilation than in the \chem{NO_2}-only assimilation, because of decreased \chem{OH} concentrations by the assimilation of non-\chem{NO2} measurements. These changes, together with the increased \chem{LNO_x} sources, could explain the smaller \chem{NO_x} emissions in the multiple-species assimilation than in the \chem{NO_2}-only assimilation in many cases (c.f., Table 5). These results  demonstrate the utility of the multiple-species assimilation to constrain the tropospheric chemistry (i.e., chemical regime) controlling \chem{NO_x} variations and to improve surface \chem{NO_x} emission inversions.

[revised manuscript text omitted]

\chem{NO_2} columns from OMI, GOME-2, and SCIAMACHY; \chem{O_3} profiles from

TES; \chem{CO} profiles from MOPITT; and \chem{O_3} and \chem{HNO_3} profiles from MLS. The daily emission inversion is performed based on the ensemble Kalman filter data assimilation, which simultaneously optimises chemical concentrations of various species and emission sources of several precursors. Within the simultaneous emission and concentration optimisation framework, the analysis increment directly produced via chemical concentrations plays an important role in reducing model--observation mismatches arising from model errors unrelated to emissions, which can be expected to improve emission inversion. The assimilation of measurements for species other than \chem{NO_2} provides additional constraints on the \chem{NO_x} emissions over both anthropogenic and biomass burning regions, leading to changes in the regional monthly-mean emissions of up to 70 \%.

The impact of non-\chem{NO_2} measurements varied largely with season, year, and region.

In addition to daily emission factors, the diurnal emission variability function was optimised using multiple \chem{NO_2} retrievals, obtained in the morning (SCIAMACHY and GOME-

2) and afternoon (OMI). The emission correction largely improved the agreement with observed tropospheric \chem{NO_2} columns, at both the seasonal and interannual time scales.

The ten-year mean global total surface \chem{NO_x} emissions after data assimilation is 48.4

\unit{Tg\,N\,yr^{-1}}, which is 26 \% higher than a priori emissions based on bottom-up inventories. The optimised ten-year mean emissions are higher over most industrialised areas.

The data assimilation corrected the timing and strength of emissions from biomass burning, such as over central Africa (the ten-year mean regional emission is 1.68 \unit{Tg\,N\,yr^{-1}}

in the a priori emissions and 2.57 \unit{Tg\,N\,yr^{-1}} in the a posteriori emissions), North

Africa (2.07 \unit{Tg\,N\,yr^{-1}} v.s 2.90 \unit{Tg\,N\,yr^{-1}}), Southeast Asia (0.47 \unit{Tg\,N\,yr^{-1}} v.s 0.68 \unit{Tg\,N\,yr^{-1}}), and South America (1.00 \unit{Tg\,N\,yr^{-1}} v.s 1.04 \unit{Tg\,N\,yr^{-1}}), suggesting a large uncertainty in fire-related emission factors in the emission inventories. At northern mid-latitudes and over Australia, the emissions are largely enhanced during summer, suggesting an important underestimation of soil sources in the a priori inventory. Using the emission ratio between different categories in the a priori emission inventories, the global total soil \chem{NO_x} emission for the 2005--2014 period is estimated at 7.9 \unit{Tg\,N\,yr^{-1}} yr$^{-1}$, which is much higher than the a priori estimate of 5.4 \unit{Tg\,N\,yr^{-1}} yr$^{-1}$. This soil \chem{NO_x} emission estimate may nevertheless be conservative, because the ratio between the source categories is kept fixed in our approach.

[revised manuscript text omitted]

20$^\circ$S--20$^\circ$N), the Southern Hemisphere (SH, 90--20$^\circ$S), and the globe (GL, 90$^\circ$S--90$^\circ$N).

Table 4: Linear trend (in \% per decade) of the regional a posteriori \chem{NO_x} emissions from the multiple-species assimilation (left column) and \chem{NO_2}-only assimilation (central column), and of the regional mean tropospheric \chem{NO_2} columns from OMI

(right column) for the period 2005-2014.

Table 5: Difference between the a posteriori emissions from the multiple-species assimilation and \chem{NO_2}-only assimilation. Relative difference for the regional ten-year mean emissions (left column), RMSE for the monthly regional emissions (central column), and range of relative difference for the monthly regional emissions (right column) are shown.

Table 6: Regional and ten-year mean difference in lower tropospheric OH concentration averaged below 790 hPa ($\Delta$ \chem{OH}) and the ratio of surface \chem{NO_x}

emission and lower tropospheric \chem{NO_2} concentration averaged below 790 hPa ($\Delta$\chem{NO_x}-emi/\chem{NO_2}) between the data assimilation run and the model simulation in the boreal summer (averaged over June--August) over the 2005--2014 period.

Table 7: Regional and ten-year mean correction factor for the emission diurnal variability ($Etc$) for 2005--2014.

Figure 1: Schematic diagram of the correction scheme for the emission diurnal variation for a case with $Etc=-0.3$. The black dotted time represents the a priori emission diurnal variability function ($Et$) for anthropogenic emissions. The black solid line represents the a posteriori emission variation after applying the daily emission scaling factor ($Et \times Es$).

The blue line represents the correction factor for the emission diurnal variability ($Etc$). The red line represents the a posteriori emission variation after applying the daily emission scaling factor and the correction factor for the emission diurnal variability ($Et \times Es - Etc$ for

07:30--10:30, and $Et \times Es + Etc$ for 10:30--13:30).

Figure 2: Global distributions of the tropospheric \chem{NO_2} columns (in

$10^{15}$\,\unit{molec\,cm^{-2}}). The results are shown for OMI (left columns, sampling time $\approx$13:00 hrs) for 2005--2014, SCIAMACHY (middle columns, 10:00 hrs) for

2005--2011, and GOME-2 (right columns, 09:30 hrs) for 2007--2014. Upper rows show the tropospheric \chem{NO_2} columns obtained from the satellite retrievals (OBS); centre shows the difference between the model simulation and the satellite retrievals (Model-OBS); and lower rows show the difference between the data assimilation and the satellite retrievals (Assim-OBS).

Figure 3: Time series of regional monthly mean tropospheric \chem{NO_2} columns (in

$10^{15}$\,\unit{molec\,cm^{-2}}) averaged over China (110--123$^\circ$E, 30--

40$^\circ$N), Europe (10$^\circ$W--30$^\circ$E, 35--60$^\circ$N), the United States (70--

125$^\circ$W, 28--50$^\circ$N), South America (50--70$^\circ$W, 20$^\circ$S--Equator),

North Africa (20$^\circ$W--40$^\circ$E, Equator--20$^\circ$N), Central Africa (10--

40$^\circ$E, Equator--20$^\circ$S), Southern Africa (25--34$^\circ$E, 22--31$^\circ$S),

Southeast Asia (96--105$^\circ$E, 10--20$^\circ$N), Australia (113--155$^\circ$E, 11--44$^\circ$S), and India (68--89$^\circ$E, 8--33$^\circ$N) obtained from the satellite retrievals (black), model simulation (blue), and the data assimilation (red). The model simulation and data assimilation results were obtained at the local overpass time of the retrievals with applying the averaging kernel.

Figure 4: (Left panels) Mean vertical \chem{NO_2} profiles obtained during the ARCTAS campaign in June--July 2009; the ARCTAS campaign in June--July 2006; the DC3 campaign in May 2012; and the SEAC$^4$RS campaign in August--September 2013. The black line represents the observation; the blue line represents the model simulation; and the red line represents the data assimilation. The error bars represent the standard deviation. (Right six panels) Scatter plots of \chem{NO_2} concentrations for the data assimilation (top) and the model simulation (bottom) during the DANDELIONS campaign (in $\mu g m^{-3}$) in September 2006 (second left columns) and during the INTEX-B campaign (in \unit{pptv}) on March 9, 2006 (third left columns) and March 11, 2006 (right columns). The straight lines represent linear regression lines for each level. Each line represents a linear fit to the points of the same colour, and the colours represent the altitude (or pressure) level.

Figure 5: Global distributions of surface \chem{NO_x} emissions (in $10^{-13}\unit{kg m^{-2} s^{-1}}$) averaged over 2005--2014. The a priori emissions (top), a posteriori emissions from the data assimilation run (middle), and analysis increment (bottom) are shown.

Figure 6: Global distribution of linear trend of the a posteriori surface \chem{NO_x} emissions (in $10^{-13}\unit{kg m^{-2} s^{-1}}$ per decade) for the period 2005--2014. The red (blue) colour indicates positive (negative) trends.

Figure 7: Time series of monthly total regional surface \chem{NO_x} emissions (in \unit{Tg\,N\,yr^{-1}} ) obtained from the a priori emissions (black lines) and the a posteriori emissions (red lines) for the period 2005--2014. The results are also shown for EDGAR-HTAP v2 emissions (green lines) for the years 2008 and 2010.

Figure 8: Seasonal variations of the regional surface \chem{NO_x} emissions (in \unit{Tg\,N\,yr^{-1}}) obtained from the a priori emissions (black line) and the a posteriori emissions (red line) averaged over the period 2005--2014. The results are also shown for the a posteriori emissions for individual years during 2005--2014 (yellow lines).

Figure 9: Time series of the difference (in \%) of the annual mean a posteriori surface \chem{NO_x} emissions relative to the 2005 emissions in the period 2005--2014 for India (yellow), China (blue), Europe (light blue), western Europe (light blue dashed line), Southern Africa (red), and the United States (green).

Figure 10: The regional distribution of ten-year mean surface \chem{NO_x} emissions (in $10^{-13}\unit{kg m^{-2} s^{-1}}$) over East Asia (upper panels), Europe (upper middle panels), the United States (lower middle panels), and Southeast Asia (lower panels) obtained from the a posteriori emissions in the period 2005--2014 (left panels), and the difference between the a posteriori emissions and a priori emissions in the period 2005-2014 (centre panels), and between the a posteriori emissions and EDGAR-HTAP v2 emissions for the years 2008 and 2010 (right panels). The black square line represents the region used for the regional mean analysis.

Figure 11: Global distribution of linear trend of the a posteriori surface \chem{NO_x} emissions for the period 2005--2010 (left) and 2011--2014 (right). The red (blue) colour indicates positive (negative) trends.

Figure 12: The regional distribution of the linear trend in surface \chem{NO_x} emissions (in $10^{-13}\unit{kg m^{-2} s^{-1}}$ per decade) during 2005--2014 over East Asia (upper left), Europe (upper right), the United States (bottom left), and Southeast Asia (bottom right), obtained from the a posteriori emissions. The black square line represents the region used for the regional mean analysis.

Figure 13: Global distribution of the ten-year mean \chem{OH} concentration (in $10^6$ \unit{molecules cm^{-3}}) in the data assimilation run (top) and its difference between the data assimilation run and the model simulation (bottom) averaged over June, July, and August over the 2005--2014 period at 850 hPa.

Figure 14: Global distribution of the annual mean correction factor for the emission diurnal variability ($Etc$) for the period 2005--2014.